# Design of an optimal combination therapy with broadly neutralizing antibodies to suppress HIV-1

Colin LaMont[1], Jakub Otwinowski[1†], Kanika Vanshylla[2], Henning Gruell[2], Florian Klein[2], Armita Nourmohammad[1,3,4]*

[1]Max Planck Institute for Dynamics and Self-Organization, Göttingen, Germany; [2]Laboratory of Experimental Immunology, Institute of Virology Faculty of Medicine and University Hospital Cologne, University of Cologne, Cologne, Germany; [3]Department of Physics, University of Washington, Seattle, United States; [4]Fred Hutchinson Cancer Research Center, Seattle, United States

**Abstract** Infusion of broadly neutralizing antibodies (bNAbs) has shown promise as an alternative to anti-retroviral therapy against HIV. A key challenge is to suppress viral escape, which is more effectively achieved with a combination of bNAbs. Here, we propose a computational approach to predict the efficacy of a bNAb therapy based on the population genetics of HIV escape, which we parametrize using high-throughput HIV sequence data from bNAb-naive patients. By quantifying the mutational target size and the fitness cost of HIV-1 escape from bNAbs, we predict the distribution of rebound times in three clinical trials. We show that a cocktail of three bNAbs is necessary to effectively suppress viral escape, and predict the optimal composition of such bNAb cocktail. Our results offer a rational therapy design for HIV, and show how genetic data can be used to predict treatment outcomes and design new approaches to pathogenic control.

*For correspondence: armita@uw.edu

Present address: †Dyno Therapeutics, Cambridge, United States

## Editor's evaluation

This paper will be of interest to scientists within the fields of statistical and biological physics, immunology, and vaccinology. The mathematical/statistical framework is rigorously constructed based on key concepts from population genetics and high-throughput viral genetic sequence data. The results provide important insights into the failures of past treatment regimens with broadly neutralizing antibodies to suppress viral escape in clinical trial participants. The results also present exciting and highly testable predictions of improved treatment strategies for combatting HIV through passive bnAb immunization.

## Introduction

Recent discoveries of highly potent broadly neutralizing antibodies (bNAbs) provide new opportunities to successfully prevent, treat, and potentially cure infections from evolving viruses such as HIV-1 (*Walker et al., 2011*; *Walker et al., 2009*; *Liao et al., 2013*; *Mouquet and Nussenzweig, 2013*; *Klein et al., 2013*; *Kwong et al., 2013*; *Caskey et al., 2015*; *Caskey et al., 2017*; *Baron et al., 2018*; *Sok and Burton, 2018*; *Zwick et al., 2001*; *Burton et al., 2012*, influenza *Sparrow et al., 2016*, and the Dengue virus *Ekiert and Wilson, 2012*; *Durham et al., 2019*). bNAbs target vulnerable regions of a virus, such as the CD4 binding site of HIV *env* protein, where escape mutations can be costly for the virus (*Walker et al., 2009*; *Chen et al., 2009*; *Zhou et al., 2010*; *Walker et al., 2011*; *Liao et al., 2013*; *West et al., 2014*; *Burton and Hangartner, 2016*). As a result, eliciting bNAbs is the goal of

a universal vaccine against the otherwise rapidly evolving HIV-1. Apart from vaccination, bNAbs can also offer significant advances in therapy against both HIV-1 and influenza (*West et al., 2014*; *Caskey et al., 2016*; *Gruell and Klein, 2018*; *Durham et al., 2019*). Specifically, augmenting current anti-retroviral therapy (ART) drugs with bNAbs may provide the next generation of HIV therapies (*Horwitz et al., 2013*; *Gruell and Klein, 2018*).

Recent studies have used bNAb therapies to curb infections by the Simian immunodeficiency virus (SHIV) in non-human primates (*Shingai et al., 2013*; *Barouch et al., 2013*; *Julg et al., 2017*), and HIV-1 infections in human clinical trials (*Caskey et al., 2015*; *Bar et al., 2016*; *Caskey et al., 2017*; *Baron et al., 2018*). Monotherapy trials with potent bNAbs, including 3BNC117 (*Caskey et al., 2015*, VRC01 *Bar et al., 2016*, and 10–1074 *Caskey et al., 2017*) indicate that administering bNAbs is safe and can suppress viral load in patients. Nonetheless, in each trial, escape mutants emerge resulting in a viral rebound after about 20 days past infusion of the bNAb. However, in trials that administered a combination of 10–1074 and 3BNC117, viral rebound was substantially suppressed (*Shingai et al., 2013*; *Baron et al., 2018*). The success of combination therapy is not surprising. For example, combinations of drugs has been repeatedly used against infectious agents, including current HIV ART cocktails and combination antibiotic treatments against Tuberculosis (*Lienhardt et al., 2012*).

The principle behind combination therapy with either drugs or antibodies is clear: It is harder for a pathogen population to acquire resistance against multiple treatment targets simultaneously than to acquiring resistance against each target separately. But deciding on combination therapy means navigating an enormous number of possible treatment options.

Experimental data from neutralization assays against pseudo-viruses together with modeling and machine learning techniques have been used to statistically characterize the efficacy of bNAbs and their combinations against different variants of HIV (*Wagh et al., 2016*; *Yu et al., 2019*). Using these neutralization models, an optimal combination therapy was proposed based on their breadth, potency of neutralization, and other relevant measures (*Wagh et al., 2016*; *Yu et al., 2019*). In another study, pharmacokinetic dynamics was coupled with drug-interaction models to determine an optimal dosing strategies (*Mayer et al., 2022*). These modeling approaches shed light on how combinations of bNAbs that can neutralize a panel of viruses. However, the key obstacle in bNAb therapy is the possibility of viral escape.

To characterize viral escape, mechanistic models, partly inspired by previous work on HIV escape from the anti-retro viral therapy (ART), have been developed to explain the dynamics of viremia in patients, following passive infusion of bNAbs. By making a fit to the trial data, these models are used to infer parameters related to the efficacy of a bNAb in clearing virions, reducing viral load and infectivity, and also to infer the characteristics of the HIV and the T-cell populations, such as the initial viral population size, the death rates of uninfected and infected T-cells, and the number of virions released by an infected T-cell (*Perelson et al., 1996*; *Ribeiro and Bonhoeffer, 2000*; *Rong et al., 2010*; *Rong et al., 2007*; *Tomaras et al., 2008*; *Lu et al., 2016*; *Reeves et al., 2020*; *Saha and Dixit, 2020*; *Cardozo-Ojeda and Perelson, 2021*; *Stephenson et al., 2021*). These detailed mechanistic models cannot easily generalize from one trial to another in order to predict the efficacy of a new bNAb mono- or combination therapy.

Evolution of the HIV-1 population is another key factor to consider in modeling the dynamics and escape of viruses in response to therapy. Studies on population genetics of HIV-1 have found rapid intra-patient evolution and turnover of the virus (*Lemey et al., 2006*; *Zanini et al., 2015*) and have indicated that the efficacy of drugs in ART can severely impact the mode of viral evolution and escape (*Feder et al., 2016*). Despite the complex evolutionary dynamics of HIV-1 within patients due to individualized immune pressure (*Nourmohammad et al., 2019*), genetic linkage (*Zanini et al., 2015*), recombination (*Neher and Leitner, 2010*; *Zanini et al., 2015*), and epistasis between loci (*Bonhoeffer et al., 2004*; *Zhang et al., 2020*), the genetic composition of a population can still provide valuable information about the evolutionary significance of specific mutations, especially in highly vulnerable regions of the virus. For example, analysis of genomic covariation in the Gag protein of HIV-1 has been successful in predicting fitness effect of mutations in relatively conserved regions of the virus, which could inform the design of rational T-cell therapies that target these vulnerable regions (*Ferguson et al., 2013*). In the context of bNAb therapy trials, an evolutionary model accounting for the intrinsic fitness cost associated with escape variants against a specific bNAb has been used to characterize HIV-1 dynamics and escape following bNAb infusion (*Meijers et al., 2021*).

In this study, we present a coarse-grained evolutionary model of viral response to bNAb infusion that uses genetic data of HIV-1 in untreated patient to predict bNAb therapy outcome by characterizing the chances of viral escape from a given bNAb in patients. Specifically, we develop a statistical inference framework that uses the high throughput longitudinal survey of viral sequences collected from 11 ART-naive patients over about 10 years of infection (*Zanini et al., 2015*) to characterize the evolutionary fate of escape mutations and to predict patient outcomes in recent mono- and combination therapy trials with 10–1074 and 3BNC117 bNAbs (*Caskey et al., 2015*; *Caskey et al., 2017*; *Baron et al., 2018*), and a trial with PGT121 bNAb (*Stephenson et al., 2021*). Using the accumulated intra-patient genetic variation from deep sequencing of HIV-1 populations in ART–naive patients (*Zanini et al., 2015*), we can estimate the diversity and the fitness effects of mutations at sites mediating escape. These variables parametrize our individual-based model for viral dynamics and characterize the expected path for a potential escape of HIV-1 populations in response to bNAb therapies in patients enrolled in the clinical trials. Although our coarse-grained model does not accurately reproduce the detailed dynamics of viremia in each patient and lacks the mechanistic insight of richer models proposed by *Perelson et al., 1996*; *Rong et al., 2010*; *Rong et al., 2007*; *Tomaras et al., 2008*; *Cardozo-Ojeda and Perelson, 2021*; *Stephenson et al., 2021*, it still accurately predicts the distribution of viral rebound times in response to passive bNAb infusions — a key measure the efficacy for a bNAb clinical trial.

Our prediction for the viral rebound time in response to a bNAb relies on only a few patient-specific parameters (i.e. the genetic diversity of patients prior to treatment), and is primarily done based on the inferred genetic parameters from the deep sequencing of HIV-1 populations in a separate cohort of ART-naive patient. Therefore, our model could be used to guide therapy trial design with bNAbs against which viral escape variants are previously characterized. To this end, we use our approach to assess a broader panel of nine bNAbs, for which escape sites can be identified from prior deep mutational scanning experiments (*Dingens et al., 2019*), to characterize the therapeutic efficacy of each of these bNAbs and to propose optimal combination therapies that can efficiently curb an HIV-1 infection. Our results showcase how the wealth of genetic data can be leveraged to guide rational therapy approaches against HIV. Importantly, this approach is potentially applicable to therapy designs against other evolving pathogens, such as chronic viruses like HCV, resistant bacteria, or cancer tumor cells.

## Model

### HIV-1 response to therapy

After infusion of bNAbs in a patient, the antibodies bind and neutralize the susceptible strains of HIV. The neutralized subpopulation of HIV-1 no longer infects T-cells, and the plasma RNA copy-number associated with this neutralized population decays. The dynamics of viremia in HIV-1 patients off ART following a bNAb therapy with 3BNC117 (*Caskey et al., 2015*), 10–1074 (*Caskey et al., 2017*), and their combination (*Baron et al., 2018*) are shown in *Figure 3—figure supplements 1–3*. With competition of the neutralized strains removed, the resistant subpopulation grows until the viral load typically recovers to a level close to the pretreatment state (i.e. the carrying capacity); see *Figure 1A*. The time it takes for the viral load to recover is the *rebound time*—a key quantity that characterizes treatment efficacy within a patient. Although the details of the viremia dynamics, especially at beginning and at the end of the therapy, may be complex (*Lu et al., 2016*; *Reeves et al., 2020*; *Saha and Dixit, 2020*; *Meijers et al., 2021*), the rebound time can be approximately modeled using a logistic growth after bNAb infusion (t > 0),

$$
N(t) = \begin{cases} N_k & t \leq 0 \\ (1-x)N_k e^{-rt} + \dfrac{N_k}{1 + \frac{1-x}{x}e^{-\gamma t}} & t > 0 \end{cases}
\tag{1}
$$

with the initial condition set for pre-treatment fraction of resistant subpopulation $x = N_r(0)/(N_r(0) + N_s(0))$, where $N_r(0)$ and $N_s(0)$ denote the size of resistant and susceptible subpopulations at time $t = 0$, respectively. Here, $\gamma$ is the growth rate of the resistant population, $r$ is the neutralization rate impacting the susceptible subpopulation, and $N_k$ is the carrying capacity (*Figure 1A*, Methods). In our analysis, we set $\gamma = 1/3 \text{ days}^{-1}$ or a doubling time of $\sim 2$ days, the known HIV-1 growth rate in patients (*Perelson et al., 1996*). We infer the neutralization rate $r$ as a global parameter

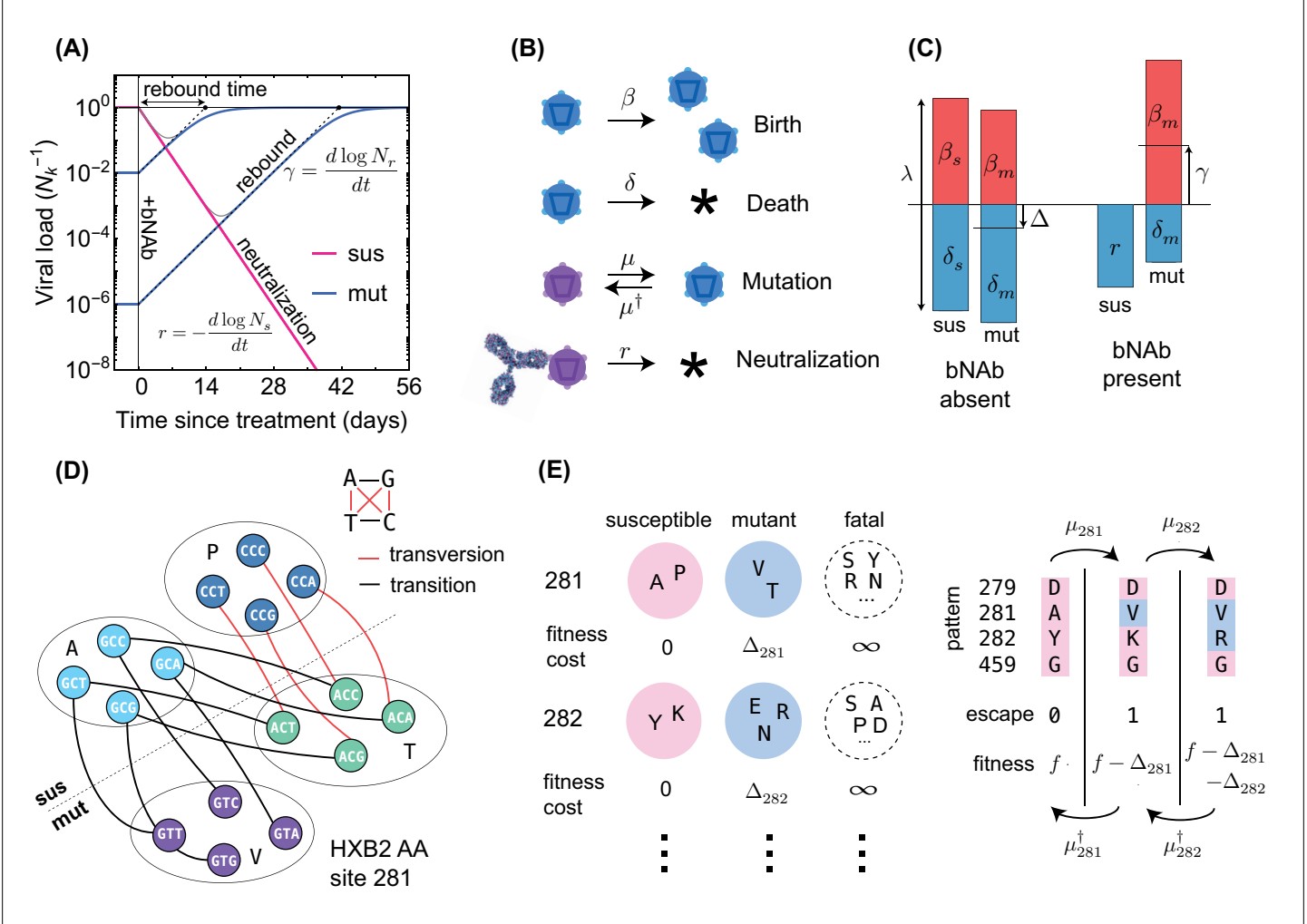

**Figure 1.** Schematics for the evolutionary dynamics of viral rebound. (**A**) The viral dynamics after the initiation of a treatment with bNAb infusion ($t = 0$) is determined by two competing processes. Susceptible strains (sus) undergo exponential decay (red line) with decay rate given by $r$, while the resistant mutants (mut) undergo logistic growth back up to the carrying capacity ($N_k$) of the patient. In the deterministic limit (***Equation 1***), the rebound time is linearly related to the log-frequency of the mutant fraction. (**B**) The schematic shows the four stochastic processes of birth, death, mutation, and neutralization with their respective rates for susceptible (purple) and resistant (blue) variants. These processes define the evolution of a viral population. Note that both the susceptible and the resistant variants are subject to birth and death with their respective rates. (**C**) The birth and death rates can be visualized as a region of size $\lambda = \beta + \delta$ which is partitioned into birth and death events. In the absence of antibodies, the susceptible population has balanced birth and death rates, $\beta_s = \delta_s$, while the resistant population has a negative net birth rate equal to the fitness difference $\Delta = \delta_m - \beta_m$. After introduction of the antibody, the susceptible population decays at rate $r$, and without competition from the susceptible population, the resistant population grows at the free growth-rate $\gamma$. (**D**) Mutational target size is inferred a priori from the genotype-phenotype mapping, which can be visualized as a bipartite graph. The nodes correspond to codons, while the edges are the mutations which link one codon to another, weighted according to the respective mutation rates. The average edge weight from codons of susceptible variants to the escape mutants determines the rate of escape mutations $\mu$. Mutations can be divided into two types: transitions (black) are within-class, and transversions (red) are out of class nucleotide changes. Transitions occur at about 8 times higher rate than transversions (***Figure 2***). (**E**) A coarse grained fitness and mutation model for two of the escape sites (281 and 282) against antibody 3BNC117 are shown. Left: At each escape mediating site, amino acids fall into one of three groups: (i) susceptible (wild-type), (ii) escape mutant, and (iii) fatal. For an escape-class amino acid at site the virus incurs a fitness cost $\Delta_i$, and these costs are additive across sites. Right: Mutations at a given site occur with (independent) forward $\mu_i$ and backward $\mu_i^\dagger$ rates which govern the substitution events between amino acid classes.

for each trial, since it depends on the neutralization efficacy of a bNAb at the concentration used in the trial. We will infer the patient-specific pre-treatment fraction of resistant subpopulation $x$, using a population genetics based approach based on which we characterize the mutational target size and selection cost of escape in the absence of a bNAb (see below). The maximum-likelihood fits of $N(t)$ to

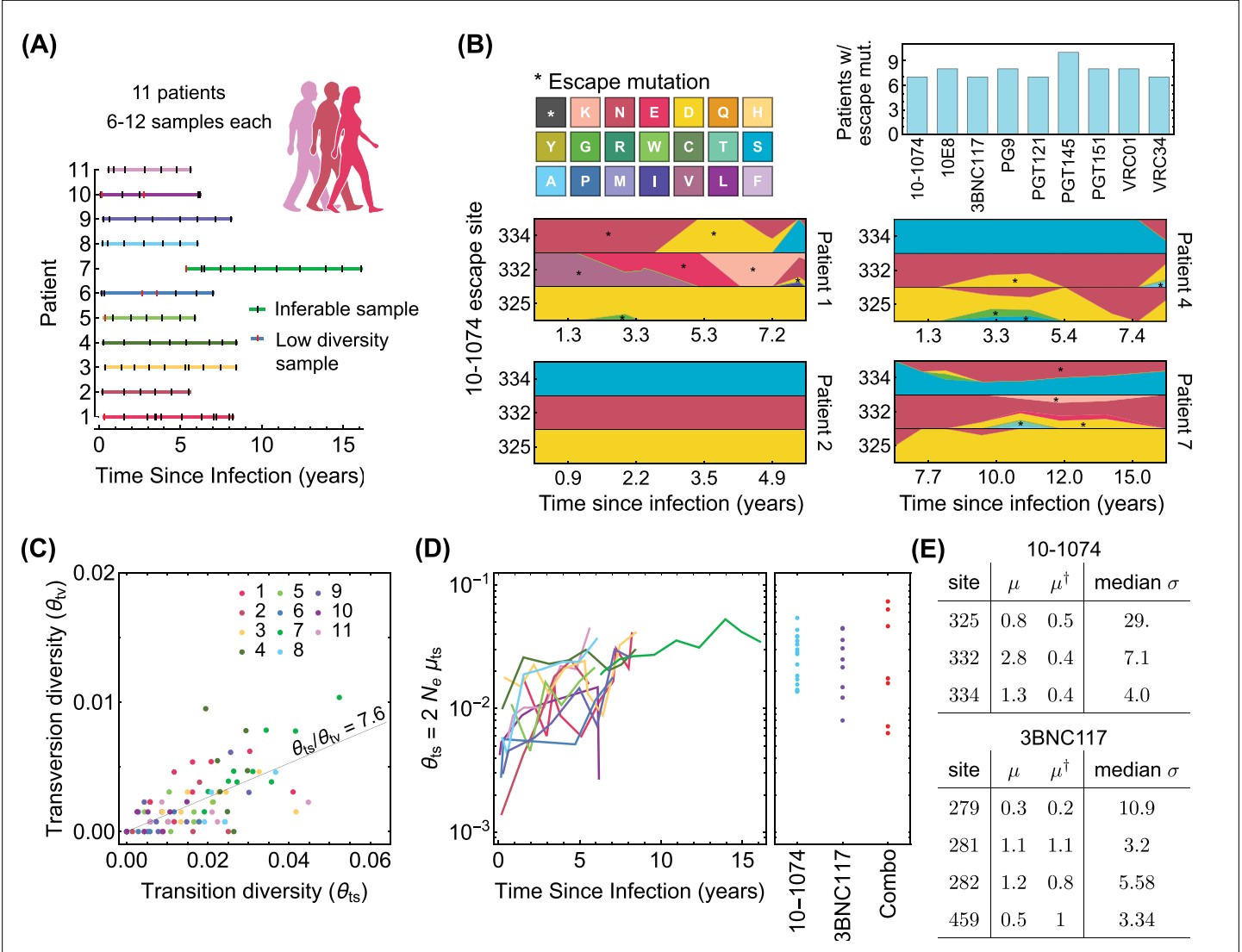

**Figure 2.** Statistics of viral genome sequences from bNAb-naive HIV-1 patients. (**A**) Statistics of the high-throughput longitudinal data collected from HIV-1 populations in 11 ART-naive patients from *Zanini et al., 2015*, is shown. Some of the have low diversity (vertical red lines) and were not usable for our study. Usable samples (vertical black lines) amount to 4–10 samples per patient, collected over 5–10 years of infection. (**B**) Lower panels show the relative frequencies (cube-root transformed for legibility) of different amino acids in four patients at the 3 escape sites against the 10–1074 bNAb, estimated from the polymorphism data at the nucleotide level in each patient over time. Despite 10–1074 being a broadly neutralizing antibody, mutations associated with escape (indicated by a *) are commonly observed in untreated patients. The upper right panel shows the number of individuals (out of the cohort of 11 bNAb-naive HIV-1 patients) that carry mutations associated with escape against the indicated bNAbs. (**C**) The nucleotide diversity associated with transversion $\theta_{tv} = 2N_e\mu_{tv}$ is shown against the transition diversity $\theta_{ts} = 2N_e\mu_{ts}$ for all patients (colors) and all time points. The covariance of these two diversity measurements yields an estimate for the transition/transversion ratio $\theta_{ts}/\theta_{tv} = 7.6$. (**D**) Left: The transition diversity is shown to grow as a function of time since infection in all the 11 patients (colors according to (**A**)). Right: The neutral diversity of viral populations in patients (points) from the three different clinical trials (*Caskey et al., 2015*; *Caskey et al., 2017*; *Baron et al., 2018*) analyzed in this study resemble the larger diversities of long-established viral populations in untreated patients. (**E**) The inferred forward and backward mutation rates ($\mu$, $\mu^\dagger$), relative to the transition rate, and the median selection strength $\sigma = 2N_e(f_{sus} - f_{res})$ at each escape site against the two bNAbs (10–1074, and 3BNC117) from the trial data used in this study are shown. Compared to the 10–1074 bNAb, escape from the 3BNC117 bNAb appears to be less costly, and is associated with a smaller mutational target.

The online version of this article includes the following figure supplement(s) for figure 2:

**Figure supplement 1.** Inference of neutral diversity from genetic data.

the viremia measurements in *Figures 2 and 3*, *Figure 3—figure supplements 1–3* specifies the initial resistant fraction $x_p$ and the rebound time $T_p$ in each patient $p$, which in this simple model, is given by $T_p = -\gamma^{-1} \log x_p$ (Methods).

The rebound times following passive infusion of 3BNC117 (*Caskey et al., 2015* and 10–1077 *Caskey et al., 2017*) bNAbs range from 1 to 4 weeks, with a small fraction of patients exceeding the monitoring time window in the studies (late rebounds past 56 days). The distribution of rebound times summarizes the escape response of the virus to a therapy and directly relates to the distribution for the pre-treatment fraction of resistant variants $P(x)$ across patients $P(T) \sim x^{-1} P(x)$.

## Stochastic evolutionary dynamics of HIV-1 subject to bNAb therapy

The fate of an HIV-1 population subject to bNAb therapy depends on the composition of the pre-treatment population with resistant and susceptible variants, and the establishment of resistant variants following the treatment. To capture these effects, we construct an individual-based stochastic model for viral rebound (*Figure 1B*). We specify a coarse-grained phenotypic model, where a viral strain of type $a$ is defined by a binary state vector $\vec{\rho}^a = [\rho_1^a, \ldots, \rho_\ell^a]$, with $\ell$ entries for potentially escape-mediating epitope sites; the binary entry of the state vector at the epitope site represents the presence ($\rho_i^a = 1$) or absence ($\rho_i^a = 0$) of an escape mediating mutation against a specified bNAb at this site of variant $a$. We assume that a variant is resistant to a given antibody if at least one of the entries of its corresponding state vector is non-zero. For multivalent treatment, a virus must be resistant to all antibodies comprising the treatment to have a positive growth after infusion.

At each generation, a virus with phenotype $a$ can undergo one of three processes: birth, death and mutation to another type $b$, with rates $\beta_a$, $\delta_a$, and $\mu_{a \to b}$, respectively (*Figure 1B*). The net growth rate of the viral subpopulation with phenotype $a$ is the birth rate minus the death rate, $\gamma_a = \beta_a - \delta_a$ (*Figure 1C*). The total rate of events (birth and death) per virion $\lambda = \beta_i + \delta_i$ modulates the amount of stochasticity in this birth-death process (Methods), which we assume to be constant across phenotypic variants. The continuous limit for this birth-death process results in a stochastic evolutionary dynamics for the sub-population size $N_a$,

$$\frac{dN_a}{dt} = \begin{cases} \text{absence of bNAb } \textbf{or} \text{ if } a \text{ is resistant :} \\ N_a \left( f_a - \phi \right) + \sum_b \left( N_b \mu_{b \to a} - N_a \mu_{a \to b} \right) \\ \quad + \sqrt{N_a \lambda} \, \eta(t) \\ \\ \text{presence of bNAb } \textbf{and} \text{ if } a \text{ is susceptible :} \\ -r N_a + \sum_b \left( N_b \mu_{b \to a} - N_a \mu_{a \to b} \right) + \sqrt{N_a \lambda} \, \eta(t) \end{cases} \tag{2}$$

where $\eta(t)$ is a Gaussian random variable with mean $\langle \eta(t) \rangle = 0$ and correlation $\langle \eta(t) \eta(t') \rangle = \delta(t - t')$ (Methods). Here, $f_a$ denotes the intrinsic fitness of variant $a$ and its net growth rate $\gamma$ is mediated by a competitive pressure $\phi = \frac{1}{N_k} \sum_b N_b f_b$ with the rest of the population constrained by the carrying capacity $N_k$, such that $\gamma_a = f_a - \phi$. In the presence of a bNAb, birth is effectively halted for susceptible variants and their death rate is set by the neutralization rate of the antibody, resulting in a net growth rate, $\gamma_{\text{sus.}} = -r$. At the carrying capacity, the competitive pressure is the mean population fitness $\phi = \bar{f}$, making the net growth rate of the whole population zero (*Figure 1C*). When susceptible variants are neutralized by a bNAb, the competitive pressure $\phi$ drops, and as a result, the resistant variants can rebound to carrying capacity, at growth rates near their intrinsic fitness.

To connect the birth-death model (*Equation 2*) with data, we should relate the simulation parameters of a birth-death process to molecular observables. We have already made a connection between the birth and death rates of a variant and its intrinsic fitness in *Equation 2*. In addition, the neutral diversity $\theta$ of a population at steady-state can be expressed as $\theta = 2N_k \mu / \lambda$, where $\mu \approx 10^{-5}$/ day is the per-nucleotide mutation rate, which we infer from intra-patient longitudinal HIV-1 sequence data (*Zanini et al., 2015*) (Methods). For consistency, we set the total rate of events $\lambda$ to be at least as large as the fastest process in the dynamics, which in this case is the growth rate of resistant viruses $\gamma \approx (3 \text{ days})^{-1}$, we choose $\lambda = (0.5 \text{ days})^{-1}$ (Methods). Therefore, the key parameters of the

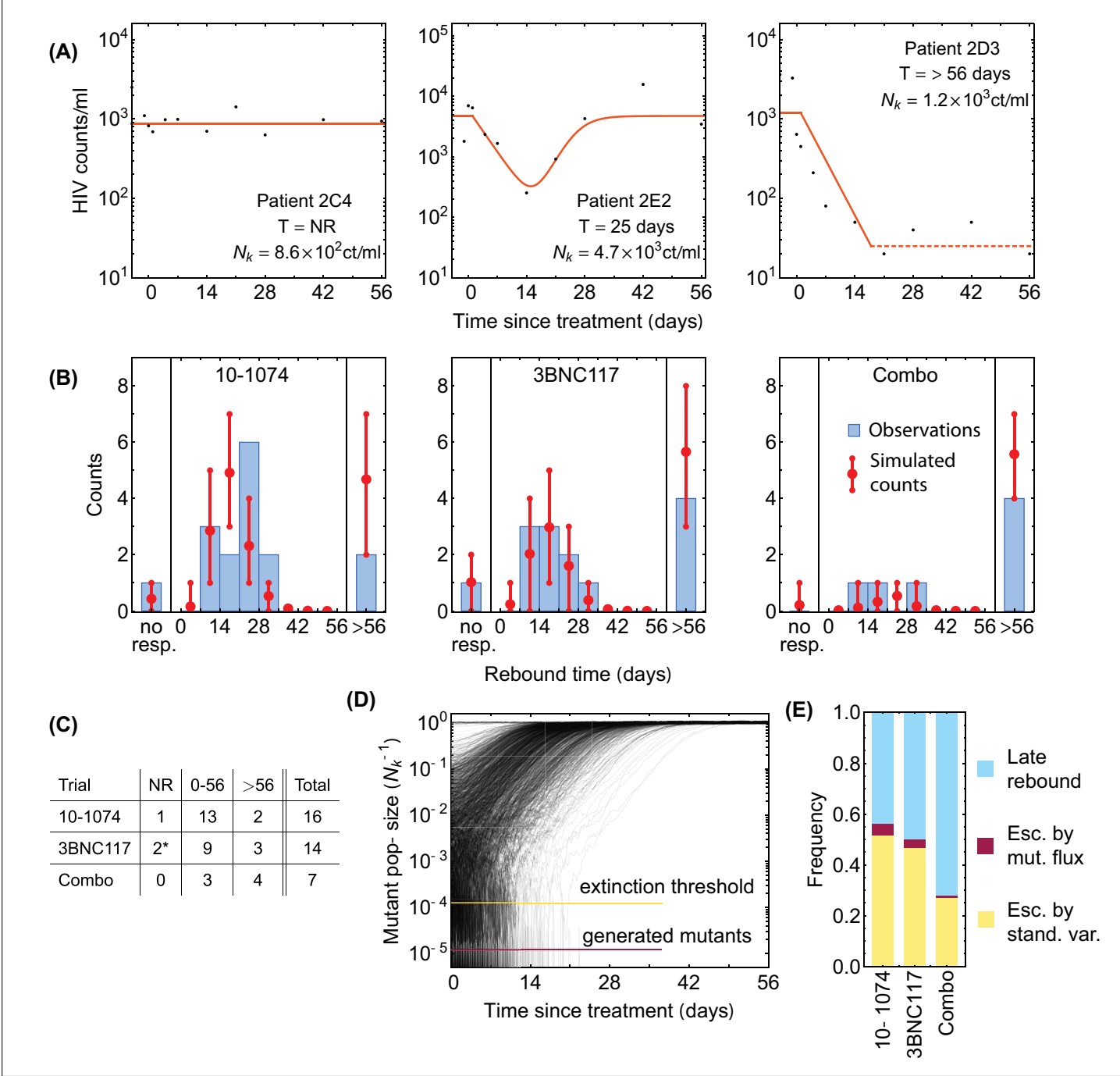

**Figure 3.** Statistics of viral rebound in clinical trials with bNAbs. (**A**) Panels show viremia of three patients from the 3BNC117 trial over time (black circles) and the fitted model of the viral decay and rebound processes from **Equation 1** (orange line). The viral rebound time $T$ and the fitted carrying capacity $N_k$ is shown in each panel. Shown are examples of a non-responder (NR; left), a rebound occurring during the trial window ($0 < T < 56$ days; center), and a late rebound ($T > 56$ days; right). (**B**) We compare the distribution of rebound times in patients from the three clinical trials with 10–1074 **Caskey et al., 2017**, 3BNC117 **Scheid et al., 2016**, and the combination of the two bNAbs **Baron et al., 2018** to the predictions from the simulations based on our evolutionary model (**Figure 1**, and Methods). The error bars show the inter decile range (0.1–0.9 quantiles) generated by the simulations for the corresponding trial. (**C**) The summary table shows the number of patients for whom the infecting HIV-1 population shows no response (NR), rebound during the trial window $0 < T < 56$, and a late rebound ($T > 56$ days) in each trial. Note that three patients were excluded from the 3BNC117 trial (*) because of insufficient dosage leading to weak viral response: 1mg/kg compared to the $3 - 30$mg/kg in the other treatment groups. (**D**) Plotted are 1,200 trajectories of the mutant viral population simulated using our individual based model. Due to the individual birth-death events, fluctuations are larger when the population size is smaller. At a critical threshold, $x_{ext}$, fluctuations are large enough to lead to almost certain extinction in the existing

*Figure 3 continued on next page*

*Figure 3 continued*

viral population. The critical threshold (yellow line) is an order of magnitude larger than the post-treatment spontaneously-generated mutant fraction (red line). (**E**) The predicted fraction of escape events associated with post-treatment spontaneous mutations (red) and the pre-treatment standing variation (yellow) are shown for the three trials. Late rebound events are indicated in blue. Because the spontaneously-generated mutant fraction is smaller than the extinction threshold, these mutations contribute to less than 4% of escape events (red), and escape is likely primarily driven driven by standing variation (yellow), that is, pre-existing escape variants.

The online version of this article includes the following figure supplement(s) for figure 3:

**Figure supplement 1.** Dynamics of viremia in patients from the 10–1074 trial.

**Figure supplement 2.** Dynamics of viremia in patients from the 3BNC117 trial.

**Figure supplement 3.** Dynamics of viremia in patients from the combination therapy trial.

birth-death model, that is, $\beta$, $\delta$, and $N_k$ can be expressed in terms of the intrinsic fitness of the variants $f_a$ and the neutral diversity $\theta$, which we will infer from data.

## Results

### Population genetics of HIV-1 escape from bNAbs

HIV-1 escape from different bNAbs has been a subject of interest for vaccine and therapy design, and a number of escape variants against different bNAbs have been identified in clinical trials or in infected individuals (*Lynch et al., 2015*; *Caskey et al., 2015*; *Scheid et al., 2016*; *Caskey et al., 2017*; *Baron et al., 2018*). This in-vivo data is often complemented with information from co-crystallized structures of bNAbs with the HIV-1 envelope protein (*Pancera et al., 2017*), and in-vitro deep mutational scanning (DMS) experiments, in which the relative change in the growth rate of tens of thousands of viral mutants are measured in the presence of different bNAbs (*Dingens et al., 2019*; *Dingens et al., 2017*; *Schommers et al., 2020*). We identify escape mutations against each of the bNAbs in this study by using information from clinical trials, the characterized binding sites, and the DMS assays (Methods); the list of escape mutations against each bNAb is given in *Appendix 1—table 1*.

The rise and establishment of an escape variant against a specific bNAb depend on three key factors, (i) neutral genetic diversity of the viral population, (ii) the mutational target size for escape from the bNAb (i.e. the number of paths leading to escape, weighted by their respective probabilities), and (iii) the intrinsic fitness associated with such mutations. Although viremia traces in clinical trials and growth experiments can be used to model the escape dynamics (*Haddox et al., 2018*; *Lynch et al., 2015*; *Lu et al., 2016*; *Reeves et al., 2020*; *Saha and Dixit, 2020*; *Meijers et al., 2021*), they do not offer a comprehensive statistical description for HIV-1 escape as they are limited by the number of enrolled individuals. Alternatively, mutation and fitness characteristics of such escape-mediating variants can be inferred from a broader cohort of untreated and bNAb-naive patients (*Illingworth et al., 2020*; *Louie et al., 2018*). We will infer statistical parameters for our coarse-grained fitness model (*Figure 1E*) from the large amount of high-throughput HIV-1 sequence data from *Zanini et al., 2015* (see *Figure 2A and B* for details) and use them to parameterize the birth-death model (*Figure 1B*).

### Diversity of the viral population

The neutral genetic diversity $\theta = 2N_k\mu/\lambda$ (i.e. the number of segregating alleles) is an observable that relates to key population genetics parameters, that is, the per-nucleotide mutation rate $\mu$, the population carrying capacity $N_k$, and the total number of events per virus in the birth-death process $\lambda$, which determines the noise amplitude in the evolutionary dynamics (Methods). $N_k$ and $\lambda$ together determine the effective population size $N_e = N_k/\lambda$. We use synonymous changes as a proxy for diversity associated with the neutral variation in an HIV-1 population at a given time point within a patient. By developing a maximum-likelihood approach based on the multiplicities of different synonymous variants, we can accurately infer the neutral diversity of a population from the large survey of synonymous sites in the HIV-1 genome (Methods and *Figure 2—figure supplement 1*). Importantly, we infer the neutral diversity of transition $\theta_{ts}$ and transversion $\theta_{tv}$ mutations separately, and consistent with previous work (*Feder et al., 2016*; *Zanini et al., 2017*), find that transitions occur with a rate of about 8 times larger than transversions (*Figure 2C*, *Figure 2—figure supplement 1B-E*).

Our inference indicates that the neutral diversity grows over the course of an infection in untreated HIV-1 patients from *Zanini et al., 2015* (*Figure 2D*). The patients enrolled in the three bNAb trials (*Caskey et al., 2015*; *Caskey et al., 2017*; *Baron et al., 2018*) show a broad range of neutral diversity prior to bNAb therapy (*Figure 2D*). In addition to the circulating viruses in a patient's sera, the viral reservoir, which consists of replication-competent HIV-1 in latently infected cells or un-sampled tissue, can also contribute to a bNAb escape in a patient. Evidence that the latent reservoir can contribute to HIV escape from bnAbs is directly visible in trials as the failure of pre-trial sequencing to exclude patients who do not harbor escape variants. We model the effect of the reservoir as augmenting the neutral diversity by a constant multiplicative factor $r_{\text{resv.}}$, so that patients with more diverse sera, representing usually longer infections, are also expected to have correspondingly more diverse reservoir populations. By fitting the observed rebound data, we infer the reservoir factor $r_{\text{resv.}} \simeq 2.07$ (Methods, *Figure 4—figure supplement 1*). We use the augmented genetic diversity of HIV-1 prior to the bNAb therapy in each trial to generate the rebound time and the probability of HIV-1 escape in patients.

## Mutational target size for escape

We define the mutational target size for escape from a bNAb as the number of trajectories that connect the susceptible codon to codons associated with escape variants, weighed by their probability of occurrences (Methods). The connecting paths with only single nucleotide transitions or transversions dominate the escape and can be represented as connected graphs shown in *Figure 1D*. To characterize the target size of escape for each bNAb, we determine the forward mutation rate $\mu \equiv \mu_{\text{sus.}\rightarrow\text{res.}}$ from the susceptible codons to the resistant (escape) codons, and the reverse mutation rate $\mu^{\dagger} \equiv \mu_{\text{sus.}\leftarrow\text{res.}}$ back to the susceptible variant (*Figure 1D*, Methods). The mutational target sizes vary across bNAbs, with HIV-1 escape being most restricted from 10E8 ($\mu/\mu_{\text{ts}} = 1.8$ where $\mu_{\text{ts}}$ is the single-nucleotide transition substitution rate) and most accessible in the presence of 10–1074 ($\mu/\mu_{\text{ts}} = 4.9$); see *Figure 4C* and *Appendix 1—table 1* for the list of mutational target size for escape against all bNAbs in this study.

## Fitness effect of escape mutations

Since bNAbs target highly conserved regions of the virus, we expect HIV-1 escape mutations to be intrinsically deleterious for the virus (*Ferguson et al., 2013*; *Meijers et al., 2021*), and incur a fitness cost relative to pre-treatment baseline $f_0$. We assume that fitness cost associated with escape mutations are additive and background-independent so the fitness of a variant $a$ in the absence of bNAb follows, $f_a = f_0 - \sum_i \Delta_i \sigma_i^a$, where $\Delta_i$ is the cost associated with the presence of a escape mutation at site (i.e., for $\sigma_i^a = 1$); see *Figure 1D*.

Interestingly, we observe the escape variants against different bNAbs to be circulating in the HIV-1 populations from the cohort of ART- and bNAb-naive patients (*Zanini et al., 2015*, *Figure 2B*). We use this data (*Zanini et al., 2015*) and extract the multiplicity of susceptible and escape variants in HIV-1 populations at each sampled time point from a given patient. We use a single locus approximation under strong selection to represent the stationary distribution of the underlying frequency of escape alleles $x$ in each patient from (*Zanini et al., 2015*), $P(x; \sigma, \theta, \theta^{\dagger}) \sim x^{-1+\theta}(1-x)^{-1+\theta^{\dagger}} \exp[-\sigma x]$, given the (scaled) fitness difference between the susceptible and the escape variants $\sigma = 2N_e(f_{\text{sus}} - f_{\text{mut}})$; see Methods.

Based on the statistics of escape and susceptible variants in all patients, we define a likelihood function that determines a Bayesian posterior for selection $\sigma$ associated with escape at each site (Methods). We found that it is statistically more robust to infer the strength of selection relative to a reference diversity measure $\sigma/\theta_{\text{ts}} = (f_{\text{sus.}} - f_{\text{res.}})/\mu_{\text{ts}}$, for which we choose the transition rate (Methods). This approach generates unbiased selection estimates in simulations and is robust to effects of linkage and recombination (Methods and Model robustness *Figure 4—figure supplement 1*, *Figure 4—figure supplement 2*). The inferred values of the scaled fitness costs $\sigma/\theta_{\text{ts}}$ are shown for the escape-mediating sites of the trial bNAbs in *Figure 2E*, and are reported in *Appendix 1—table 1*.

## Predicting the efficacy of bNAb therapy in clinical trials

Monotherapy trials with 10–1074 (*Caskey et al., 2017*, 3BNC117 *Caskey et al., 2015*, PGT121 *Stephenson et al., 2021*), and the combination therapy with 10–1074+3BNC117 in *Baron et al., 2018* have shown variable outcomes. In some patients, bNAb therapy did not suppress the viral load,

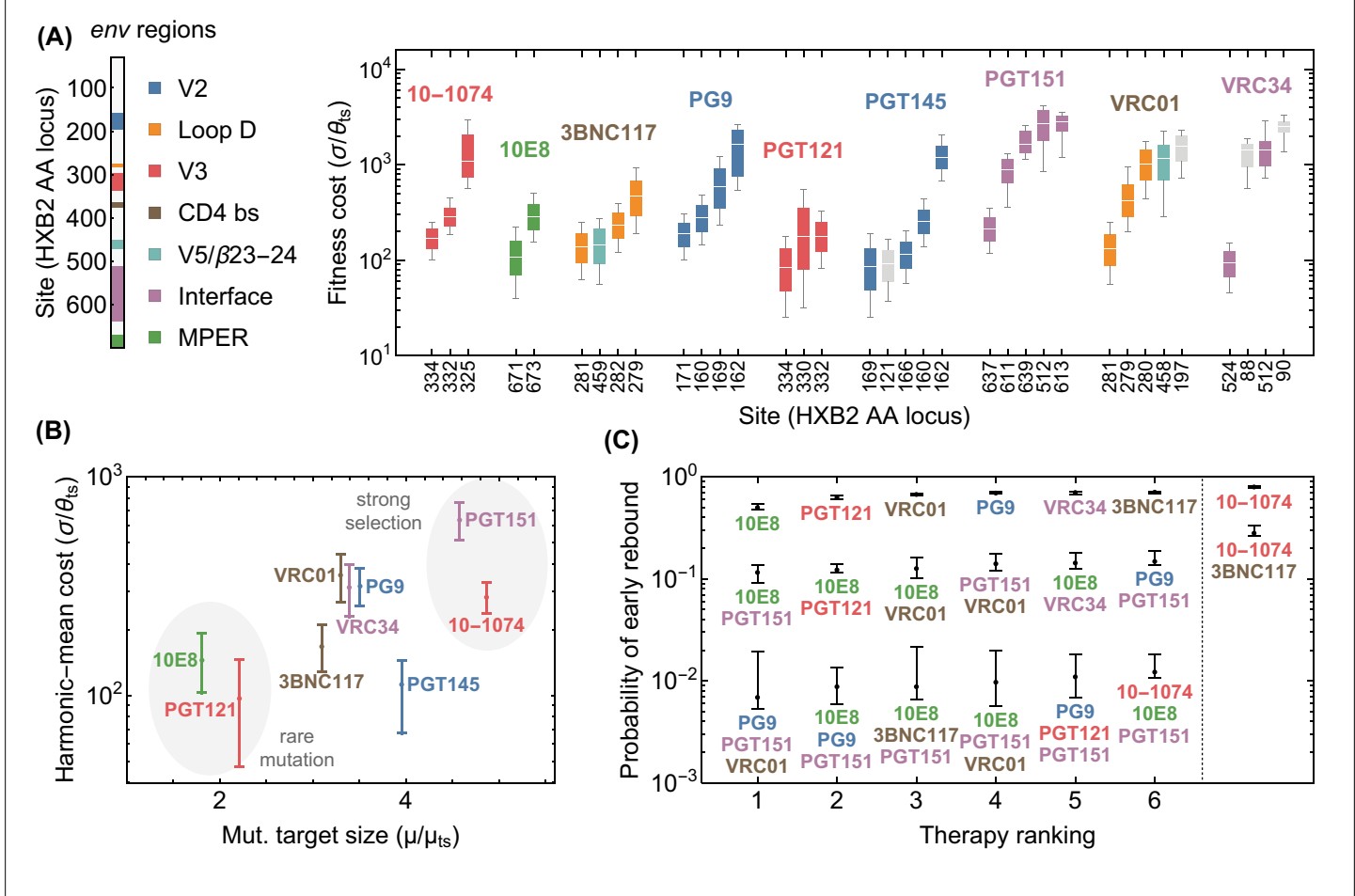

**Figure 4.** Statistics of viral escape for optimal combination therapy with bNAbs. (**A**) The posterior distribution for inferred selection strength on the escape-mediating sites associated with each of the 9 bNAbs in this study is shown (right); white line: median, box: 50% around the median, bar: 80% around median. Each escape site is color coded by its location on the *env* gene (left) and each antibody by its associated epitope location. (**B**) The harmonic mean of the selection strength $\sigma$ associated with cost of escape (scaled by transition diversity $\theta_{ts}$) is shown against the mutational target size for each bNAb; error bars indicate 50% around the median. For antibodies to be broadly neutralizing, it is sufficient that viral escape from them to be associated with a small mutational target size or a large fitness cost. The mutational target size is found to be weakly correlated with the average cost of escape from a given bNAb. We identify two distinct strategies for antibody breadth—selection limited and mutational-target-size limited escape pathways each highlighted in gray. (**C**) bNAb therapies with 1, 2, and 3 antibodies are ranked based on the predicted probability of early viral rebound, and in each case, six therapies with highest efficacies are shown; best ranked therapy is associated with the lowest probability of early rebound ($lt_{56}$ days); indicate 50% around the median. Also for reference, the probability of early viral rebound two therapies from the trials in this study (10–1074 and 10–1074+3BNC117) are shown.

The online version of this article includes the following figure supplement(s) for figure 4:

**Figure supplement 1.** Robustness of selection inference to intra-patient temporal correlations of HIV-1 alleles and clade-specific sampling.

**Figure supplement 2.** Robustness of selection inference to genetic linkage and hitchhiking.

**Figure supplement 3.** Minimum disparity estimation for adjustment of diversity.

**Figure supplement 4.** Disparity analysis for robustness of selection inference.

whereas in others suppression was efficient and no rebound was observed up to 56 days after infusion (end of surveillance in these trials); see *Figure 3A* for examples of patients with different rebound times, *Figure 3—figure supplements 1–3* for the viremia traces in all patients, and *Figure 3B and C* for the distributions and the summary statistics of the rebound times in patients in different trials.

Although we infer a large intrinsic fitness cost for a virus to harbor an escape allele (*Figure 2E*), these variants can emerge or already be present due to the large intra-patient diversity of HIV-1 populations (*Figure 2C*), or a larger mutational target size for these escape variants. Deep sequencing

data in untreated (likely bNAb-naive) patients shows circulation of resistant variants against a panel of bNABs in the majority of patients (**Figure 2B**). Our goal is to predict the efficacy of a bNAb trial, using the fitness effect and the mutational target size for escape from a given bNAb, both of which we infer from the high-throughput HIV-1 sequence data collected from bNAb-naive patients in **Zanini et al., 2015** (**Figure 2E**, **Appendix 1—table 1**). In addition, we modulate these measures with the patient-specific neutral diversity $\theta$ inferred from whole genome sequencing of HIV-1 populations in each patient prior to bNAb therapy (**Figure 2D**). These quantities parametrize the birth-death process for viral escape in a bNAb therapy (**Figure 1A**), which we use to characterize the distribution of rebound times in a given trial (Methods).

For both the 3BNC117 and the 10–1074 trials (**Caskey et al., 2015**; **Caskey et al., 2017**), we see an excellent agreement between our predictions of the rebound time distribution and data; see **Figure 3B**, and Methods and **Figure 4—figure supplement 3** for statistical accuracy of this comparison. For the PGT121 trial, the genomic data from patients' HIV-1 populations are insufficient and therefore, we used the neutral diversity $\theta_{\mathrm{ts}}$ estimated from the other three trials to predict the associated rebound time distribution (Appendix 4). Still, we see a good agreement between our predictions of the rebound time distribution and data; see **Appendix 4—figure 1**.

By assuming an additive fitness effect for escape from 10 to 1074 and 3BNC117, we also accurately predict the distribution of rebound times in the combination therapy (**Baron et al., 2018**, **Figure 3B**). The agreement of our results with data for combination therapy is consistent with the fact that the escape mediating sites from 10 to 1074 and 3BNC117 are spaced farther apart on the genome than 100 bp, beyond which linkage disequilibrium diminishes due to frequent recombination in HIV (**Zanini et al., 2015**). Importantly, in all the trials, our evolutionary model accurately predicts the fraction of participants for whom we should expect a late viral rebound (more than 56 days past bNAb infusion)—the quantity that determines the efficacy of a treatment.

Apart from the overall statistics of the rebound times, our stochastic model also enables us to characterize the relative contributions of the pre-treatment standing variation of the HIV-1 population versus the spontaneous mutations emerging during a trial to viral escape from a given bNAb. Given the large population size of HIV-1 and a high mutation rate ($\mu = 10^{-5}$ per generation), spontaneous mutations generate a fraction $x^{(\mu)}$ of resistant variants during a trial, which we can express as,

$$x^{(\mu)} = \int_0^{56\,\mathrm{wks}} (1 - x(0))e^{-rt}\mu\gamma\mathrm{d}t \tag{3}$$

In the best case scenario, there are no resistant virions prior to treatment i.e., $x(0) = 0$. Since the neutralization rate $r$ and the growth rate $\gamma$ are comparable, this deterministic approach predicts that mutations can generate a resistant fraction of $x^{(\mu)} \approx 10^{-5}$ during a trial. However, stochastic effects from random birth and death events play an important role in the fate and establishment of these resistant variants. The probability of extinction for a variant at frequency $x$ can be approximated as $p(\mathrm{extinct}) \approx 1 - e^{-x/x_{\mathrm{ext}}}$ (see Extinction Probability); here $x_{\mathrm{ext}} = \frac{\mu_{ts}}{\gamma\theta_{ts}} \approx 10^{-4}$ and a variant with fraction $x$ that falls below this critical value is likely to go extinct (**Figure 3D**). Since the total integrated mutational flux fraction during a trial is $x^{(\mu)} \sim 10^{-5}$, mutational flux rarely decides the outcome of patient treatment. Indeed, we infer that spontaneous mutations contribute to less than 4% of escape events in all the three trials and escape is primarily attributed to the standing variation from the serum or the reservoirs prior to therapy (**Figure 3E**). A similar conclusion was previously drawn based on a mechanistic model of escape in VRC01 therapy trials (**Saha and Dixit, 2020**).

## Devising optimal bNAb therapy cocktails

Clinical trials with bNAbs have been instrumental in demonstrating the potential role of bNAbs as therapy agents and in measuring the efficacy of each bNAb to suppress HIV. Still, these clinical trials can only test a small fraction of the potential therapies that can be devised. It is therefore important that trials test therapies that have been optimized based on surrogate estimates of treatment efficacy. The accuracy of our predictions for the rebound time of a HIV-1 population subject to bNAb therapy suggests a promising approach to the rational design of therapies based on genetic data of HIV populations collected from bNAb-naive patients.

Here, we use viral genome sequences to infer the efficacy of therapies with bNAbs, for which clinal trials are not yet performed. To do so, we first need to identify the routes of HIV escape from these bNAbs. We use deep mutational scanning data (DMS) on HIV-1 subject to 9 different bNAbs

from *Dingens et al., 2019* together with information from literature to identify the escape mediating variants from each of these bNAbs (Methods and Table S1). We then determine the mutational target size and the fitness effect of these escape variants using high-throughput sequences of HIV-1 in bNAb-naive patients from *Zanini et al., 2015*; these inferred values are reported in *Figure 4A and B* and Table S1. Using the inferred fitness and mutational parameters and by setting the pre-treatment neutral diversity $\theta$ to be comparable to that of the patients in the previous three trials with 10–1074 and 3BNC117 (*Figure 2C*), we simulate treatment outcomes for these 9 bNAbs and their twofold and threefold combinations.

Interestingly, we infer that escape from mono-therapies is almost certain and a combination of at least three antibodies is necessary to limit the probability of early rebound (<56 days) to below 1% (*Figure 4C*). When considering each of the nine antibodies, interesting patterns emerge. We find that the mutational target size and the fitness cost of escape, estimated as the harmonic-mean selection cost of individual sites, obey a roughly linear relationship (*Figure 4B*). As all these bNAbs have similar overall breadth (i.e. they neutralize over 70% of panel strains), this result suggests that for an antibody to be broad, its escape mediating variants should either be rare (i.e. small mutational target size) or intrinsically costly (i.e. incurring a high fitness cost), but it is not necessary to satisfy both of these requirements. For instance, we find that resistant variants against 10E8 weaker negative selection but escape target size is small, while PGT151 has a larger escape target size but makes up for it by having resistant variants with unavoidably high fitness cost.

The fitness-limited versus the mutation-limited strategies have different implications for the design of combination cocktails. The small mutational target size of 10E8 makes it the best candidate antibody for mono-therapy among the antibodies we consider because the escape variants against this antibody are less likely to circulate in a patient's serum prior to treatment. However, in combination, 10E8 appears less often in top ranked therapies than PGT151. PGT151 is unremarkable on its own because of a relatively large target size, but the high cost of escape makes it especially promising in combination therapies. Overall, fitness-limited bNAbs like PGT151 are more effective against high diversity viral populations, while mutation-limited bNAbs such as 10E8 are more effective against low diversity viral populations. Indeed, the best ranked therapy, namely the combination of PG9, PGT151, and VRC01, combines antibodies that target different regions of the virus and also have both types of fitness- and mutation-limited strategies for coverage against the full variability of viral diversities found in pre-treatment individuals (*Figure 4C*) participating in the clinical trials.

Escape from bNAbs in multivalent therapy is also influenced by the non-independence of escape pathways. Several of the bNAbs (e.g. 10–1074 and PG121) in this study target the same (or structurally adjacent) epitopes and escape from them is mediated by the same mutations. The sharing of mutational pathways invalidates the assumption of independence and may make our predictions too optimistic. However, the best performing antibody combinations in *Figure 4C* target multiple epitopes to reduce the chances of collective escape. Therefore, the assumptions of independent fitness effects and independent mutational pathways in this case are not consequential for the main predictions of our model (i.e., the choice of bNAb combinations). In vitro data shown in *Kong et al., 2015* suggest that additive and independent effects are the norm, but that small but consistent synergistic effects may imply that our assumption of site-wise independence is conservative. More data on HIV escape from combinations of bNAbs would be informative for further modeling efforts and relevant for long-term therapy design.

## Discussion

HIV therapy with passive bNAb infusion has become a promising alternative to anti-retroviral drugs for suppressing and preventing the disease in patients without a need for daily administration. The current obstacle is the frequent escape of the virus seen in mono- and even combination bNAb therapy trials (*Caskey et al., 2015*; *Bar et al., 2016*; *Caskey et al., 2017*; *Baron et al., 2018*). The key is to identify bNAb cocktails that can target multiple vulnerable regions on the virus in order to reduce the likelihood for the rise of resistant variants with escape-mediating mutations in all these regions. Identifying an optimal bNAb cocktail can be a combinatorially difficult problem, and designing patient trials for all the potential combinations is a costly pursuit.

Here, we have proposed a computational approach to predict the efficacy of a bNAb therapy trial based on population genetics of HIV escape, which we parametrize using high-throughput HIV-1

sequence data collected from a separate cohort of bNAb-naive patients (*Zanini et al., 2015*). Specifically, we infer the mutational target size for escape and the fitness cost associated with escape-mediating mutations in the absence of a given bNAb. These quantities together with the neutral diversity of HIV-1 within a patient parametrize our stochastic model for HIV dynamics subject to bNAb infusion, based on which we can accurately predict the distribution of rebound times for HIV in therapy trials with 10–1074, 3BNC117 and their combination, as well as a trial with PGT121 bNAb. Consistent with previous work on VRC01 (*Saha and Dixit, 2020*), we found that viral rebounds in bNAb trials are primarily mediated by the escape variants present either in the patients' sera or their latent reservoirs prior to treatment, and that the escape is not likely to be driven by the emergence of spontaneous mutations that establish during the therapy.

One key measure of success for a bNAb trial is the suppression of early viral rebound. Our model can accurately predict the rebound times of HIV-1 subject to three distinct therapies (*Caskey et al., 2015*; *Caskey et al., 2017*; *Baron et al., 2018*), based on the fitness and the mutational characteristics of escape variants inferred from high-throughput HIV-1 sequence data. This approach enables us to characterize routes of HIV-1 escape from other bNAbs, for which therapy trials are not available, and to design optimal therapies. We used deep mutational scanning data *Dingens et al., 2019* to identify escape-mediating variants against 9 different bNAbs for HIV. Our genetic analysis shows that bNAbs gain breadth and limit viral escape either due to their small mutational target size for escape or because of the large intrinsic fitness cost incurred by escape mutations. bNAbs with mutation-limited strategy are more effective at preventing escape in patients with low viral genetic diversity, while bNAbs with selection-limited strategy are more effective at high viral diversity.

Combination therapy with more than two bnAbs (or drugs in ART) has long been shown to be more effective in suppressing early viral rebound, both in theory and practice (*Perelson et al., 1996*; *Feder et al., 2016*; *Wagh et al., 2016*; *Klein et al., 2012*; *Mendoza et al., 2018*; *Yu et al., 2019*). In addition to corroborating this conclusion quantitatively, we provide a method for assessing new bnAbs for which escape mutations are known. Our method can be understood as a tool to navigate the combinatorial explosion of higher order cocktails for which we cannot possibly test all combinations. By assessing the evolvability of resistance against different combinations we can identify the best therapies to target for clinical trial. Specifically, we show that to suppress the chance of viral rebound to below 1%, we show that a combo-therapy with 3 bNAbs with a mixture of mutation- and selection-limited strategies that target different regions of the viral envelope is necessary. Such combination can counter the full variation of viral diversity observed in patients. We found that PG9, PG151, and VRC01, which respectively target V2 loop, Interface, and CD4 binding site of HIV envelope, form an optimal combination for a 3-bNAb therapy to limit HIV escape in patients infected with clade B of the virus.

The statistical agreement between our coarse-grained model and the observed distribution of the viral rebound times in trials (*Figure 3B*) implies that many of the mechanistic details are of secondary importance in predicting viral escape. Nonetheless, our approach falls short of reproducing the detailed characteristics of viremia traces in patients, especially at very short or very long times, during which the dynamics of T-cell response or the decay of bNAbs could play a role (*Lu et al., 2016*; *Reeves et al., 2020*; *Saha and Dixit, 2020*). The relationship between the short-term suppression of the virus, which is the focus of this analysis, and the long-term treatment success is complicated by reestablishment of HIV from the latent reservoirs and different modes of intra-host HIV evolution (*Liu et al., 2019*; *Margolis et al., 2017*).

One strategy to achieve a longer term treatment success is by combining bNAb therapy with ART. One main advantage of bNAb therapy is the fact that it can be administered once every few months, in contrast to ART, which should be taken daily and missing a dose could lead to viral rebound. Although multivalent bNAb therapy reduces the chances of short-term viral escape, viral escape remains a real obstacle for longer term success of a treatment with bNAbs. Alternatively, (fewer) bNAbs can be administered in combination with ART (*Horwitz et al., 2013*; *Gruell and Klein, 2018*), whereby ART could lower the replication rate of the HIV population, reducing the viral diversity and the chances of viral escape. Specifically, we can expect that emergence and establishment of rare (i.e., strongly deleterious) escape variants against bNAbs to be less likely in ART+ patients, which suggests that fitness-limited bNAbs should be more effective in conjunction with ART. More data would be necessary to understand the long-term efficacy of such augmented therapy, and specifically the role of viral

reservoirs in this context. A modeling approach could then shed light on how ART administration and bNAb therapy could be combined to efficiently achieve viral suppression.

## Limitations

We rest our analysis primarily on the predictive power of the observed variant frequencies in the untreated patients. Our model weighs these frequencies with respect to the viral diversity in a mathematically and biologically consistent way. However, we ignore the dynamics of antibody concentration and $IC_{50}$ neutralization during treatments, the details of T-cell dynamics during infection (*Perelson, 2002*), and also the evolutionary features of the genetic data, such as epistasis between loci (*Bonhoeffer et al., 2004*; *Zhang et al., 2020*, genetic linkage *Zanini et al., 2015*, and codon usage bias *Meintjes and Rodrigo, 2005*).

In our model of viral escape, we neglect the possibility of incomplete escape of the virus due to the reduced neutralization efficacy of bNAbs as their concentrations decay during trials. In Appendix 3, we show that this simplifying assumption is valid as long as the $IC_{50}$ is not the same order of magnitude as the initial dosage concentration of the infused bNAb. Notably, the data from therapy trials used in this study fall into the regime for which we can neglect the impact of incomplete neutralization (*Appendix 3—figure 2*). However, taking into account the dependence of viral fitness on bNAb concentration and its neutralization efficacy, as in the model proposed by *Meijers et al., 2021*, could improve the long-term predictive power of our approach.

Our predictions are limited by our ability to identify the escape variants for each bNAb, either based on the in vivo trial data and patient surveillance, or in-vitro assays such as DMS experiments. In Appendix 4, we compare the accuracy of our predictions for the rebound time distributions of HIV using DMS-inferred versus trial-inferred escape variants against the 10–1074 and the PGT121 bNAbs, and we find good agreements between the two approaches in both cases. However, it should be noted that identifying escape variants from DMS experiments lead to a more optimistic prediction for treatment success during the first 8 weeks of trials (i.e. they suggest a later rebound). One reason for this discrepancy may be related to the distinct genetic composition of viruses in the two approaches. DMS experiments characterize HIV escape by introducing mutations on a single genetic background (*Dingens et al., 2019*; *Dingens et al., 2017*; *Schommers et al., 2020*) (e.g. the HIV-1 strain BF520.W14M.C2 in *Dingens et al., 2017*), whereas clinical data contain diverse populations of viruses between and within individuals. It is more likely for diverse viral populations to contain variants in which positive epistasis between escape mutations and the background genome is present. Therefore, it is reasonable to expect that genetic data originating in clinical trials show more pathways of escape, resulting in a faster viral rebound following bNAb infusion. Nevertheless, DMS experiments are less costly for identifying escape variants compared to trials, and they provide a baseline to assess the efficacy of different bNAbs for therapy and further in-vivo investigations. More experiments would be necessary for a more systematic understanding of the limitations of each approach.

It should be noted that our analysis in *Figure 4C* only focuses on one aspect of therapy optimization, that is, the suppression of escape. Other factors, including potency (neutralization efficacy) and half-life of the bNAb, or the patient's toleration to bNAbs at different dosages should also be taken into account for therapy design. For example, the bNAb 10E8, which we identified as of the most promising mono-therapy candidates in *Figure 4C*, is shown to be poorly tolerated by patients with short half-life (*Kwon et al., 2016*), making it undesirable for therapy purposes. Thus, the bNAb candidates shown in *Figure 4C* should be taken as a guideline to be complemented with further assessment of efficacy and safety for therapy design.

## Outlook

Our approach showcases that, when feasible, combining high-throughput genetic data with ecological and population genetics models can have surprisingly broad applicability, and their interpretability can shed light into the complex dynamics of pathogens subject to therapy. Application of similar methods to therapy design to curb the escape of cancer tumors against immune- or chemo-therapy, the resistance in bacteria against antibiotics, or the escape of seasonal influenza against vaccination is a promising avenue for future work. However, we expect that more sophisticated methods for inferring fitness from evolutionary trajectories may be necessary to capture the dynamical response of these populations.

## Methods

### Data and code accessibility

The code for the algorithms used in this work and the data are available on GitHub at https://github.com/StatPhysBio/HIVTreatmentOptimization (copy archived at swh:1:rev:0194cf6e554996a066633e99d-d53cd5901da552e, *Chelate, 2022a*) and in the Julia package https://github.com/StatPhysBio/EscapeSimulator (copy archived at swh:1:rev:9a343f598820bafddfc7ea4547cefa90bf96fd6e, *Chelate, 2022b*).

### Description of molecular data

#### Data from bNAb trials

In this study, we considered four clinical trials for passive therapy with bNAbs:

- Monoclonal therapy with 3BNC117 bNAb *Caskey et al., 2015* and sequences reported in *Schoofs et al., 2016*. There were 16 patients enrolled, 13 of whom were off anti-retroviral therapy (ART).
- Monoclonal therapy with 10–1074 bNAb *Caskey et al., 2017*, with 19 patients enrolled, 16 of whom were off ART.
- Combination therapy with 10–1074+3BNC117 bNAb *Baron et al., 2018*, with 7 patients off ART.
- Monoclonal therapy with PGT121 with 17 patients off ART. *Stephenson et al., 2021*. Far fewer viral sequences were available for this trial, and therefore, we could not reliably infer the neural diversity of viruses within patients, as we did for other trials. We only used this trial to compare our predictions based on the DMS-inferred with the trial-inferred escape pathways in Appendix 4.

All sequenced patients across all trials were infected with distinct HIV-1 clade B viral strains. We limited our analyses to those patients not on ART at the time of treatment initiation. In these studies, the injected bNAb level falls off over time within patients and therefore, we only considered dynamics within an 8-week window since infusion. This assures that rebound is not confounded by a drop in bNAb below sensitive-strain neutralizing levels of $IC_{50_S} < 2\mu g/m$.

We used single-genome sequence data of *env* collected from all patients in each trial to characterize the diversity of HIV-1 population within each patient shown in *Figure 2* available from *Zanini et al., 2015* and accessible through European Nucleotide Archive (Accession no: PRJEB9618). The patient sequence data for each trial is available through *Caskey et al., 2015* GeneBank accession number: KX016803, *Caskey et al., 2017* GenBank accession numbers KY323724.1 - KY324834.1, and *Baron et al., 2018*, GenBank accession numbers MH632763 - MH633255.

#### Longitudinal HIV-1 sequence data from untreated patients

Single nucleotide polymorphism (SNP) data was obtained from *Zanini et al., 2015* and aligned to the HXB2 reference using HIV-align tool (*Gaschen et al., 2001*). The dataset includes 11 patients observed for 5–8 years of infection, with HIV-1 sequence data sampled over 6–12 time points per patient (*Figure 2*).

Patient 4 and patient 7 were excluded from the original analysis done in *Zanini et al., 2015* because of suspected superinfection and failure to amplify early samples, respectively. These patients were included in our analysis, since (i) super-infection poses no additional difficulties for our tree-free procedure, and (ii) only time points with measurable viral diversity entered into our selection likelihood, which automatically limits our analysis to samples with high-quality sequences.

All patients were infected with clade B of HIV, except for patient 6 (clade C), and patient 1 (clade 01_AE). We assessed the robustness of our inference to exclusion of these patients from our analysis in Effect of genomic linkage on the inference of selection. Overall, our inference was not strongly affected by this choice (Model robustness *Figure 4—figure supplement 1*), and therefore, we included these patients in our main analyses to enhance the statistics with larger data.

For our analysis we considered only data reported in single nucleotide polymorphism (SNP) counts. The number of raw SNP counts are the result of amplification and must be converted into estimates for the number of pre-amplification template fragments. Zanini et. al. reported that on average about $10^2$ templates of amplicon were associated with the fragments of the envelope (*env*) protein *Zanini*

*et al., 2015*. We converted raw SNP counts into templates by normalizing to 120 counts and rounding the resulting number to the nearest integer.

## Identifying escape-mediating variants against bNAbs

The starting point for our analysis of HIV response to a given a bNAb is the description of the escape-mediating amino acids in the HIV *env* protein. We use a combination of methods to identify the escape variants for a given bNAb. First, we use deep mutational scanning (DMS) data of HIV-1 in the presence of a bNAb from *Dingens et al., 2019* to identify these mutations. These DMS experiments have created libraries of all single mutations from a given genomic background of HIV-1 and tested the fitness of these variants (i.e. growth on T-cell culture) in the absence and presence of nine different bNAbs, including the 10–1074 and 3BNC117 *Dingens et al., 2019*.

Escape variants in DMS data are identified as those which are strongly selected for only in the presence of a bNAb. Specifically, we identify escape variants as those which show 3-logs-change in their frequency in the presence versus absence of a bNAb. DMS data reflects in vitro escape in cell culture. However, some of these variants may not be viable in vivo. To identify the reasonable candidates of escape in vivo, we limit our set to the variants that are also observed in the circulating viral strains of untreated HIV-1 patients from *Zanini et al., 2015*. It should be noted that since we use HIV-1 sequence data from *Zanini et al., 2015* to infer selection on escape mediating variants in the absence of a bNAb, the candidates of escape that are not observed in the dataset *Zanini et al., 2015* would be inferred to be strongly deleterious, and hence, unlikely to contribute to our predictions of viral rebound.

Our analysis of DMS data results in a set of escape mediating amino acids for 10–1074 that is consistent with the escape variants that emerge in response to the bNAb trial (*Caskey et al., 2017*; *Stephenson et al., 2021* Appendix 4). However, the DMS data is very noisy for bNAbs that target CD4 binding site of HIV, that is, 3BNC117 and VRC01 (*Dingens et al., 2019*). One reason for this observation may be that the CD4 binding site is crucial for the entry of HIV to the host's T-cells and mutations in this region are highly deleterious. As a results, only a small number of variants with mutations in this region can survive in the absence of a bNAb in a DMS experiment. Growth in the absence of a bNAb is the first step in the DMS experiments, which is then followed by exposure of the replicated variants to a bNAb. Therefore, a low multiplicity of variants in the absence of a bNAb could result in a noisy pattern of growth of the small subpopulation in the next stage of the experiment, in which growth is subject to a CD4-targeting bNAb.

For the CD4 binding site antibodies 3BNC117 and VRC01, we used additional data to call the escape variants. For 3BNC117, we used a combination of trial-patient sequences (*Caskey et al., 2015*; *Scheid et al., 2016*), that is post-treatment enrichment, along with contact site information compiled in the crystallographic studies to narrow down candidate sites (*Zhou et al., 2015*; *LaBranche et al., 2018*). For VRC01, we assumed a similar escape pattern to 3BNC117 but included sites known from other studies (*Lynch et al., 2015*) and the clear DMS signal at HXB2 site 197. The sites we called were similar to those identified using humanized-mouse models of HIV infection (*Horwitz et al., 2013*), although more complex mutational patterns were seen in the soft-randomization scanning of *Otsuka et al., 2018*. Although the complete list of escape substitutions are unknown and background-dependent (*Otsuka et al., 2018*), the escape profiles which are most important are those that are most likely to be seen consistently in data and to be correctly identified. The list of substitutions are shown in *Appendix 1—table 1*.

## Statistics and dynamics of viral rebound

### Inference of growth parameters from dynamics of viremia

The concentration of viral RNA copies in blood serum is a delayed reflection of the total viral population size $N(t)$, containing a resistant and susceptible subpopulations, with respective sizes $N_r(t)$ and $N_s(t)$. After infusion of bNAbs in a patient, the susceptible sub-population decays due to neutralization by bNAbs and the resistant sub-population grows and approaches the carrying capacity $N_k$, with the dynamics,

$$\frac{dN_r}{dt} = \gamma N_r(1 - N_r/N_k)$$
$$\frac{dN_s}{dt} = -rN_s \tag{4}$$

Here, $\gamma$ is the growth rate of the resistant population, and $r$ is the neutralization rate impacting the susceptible subpopulation. By setting the initial condition for fraction of resistant subpopulation prior to treatment (at time $t = 0$) $x = N_r(0)/(N_r(0) + N_s(0))$, we can characterize the evolution of the total viremia in a patient. This dynamics is governed by the combined processes of neutralization by the infused bNAb and the viral rebound (*Figure 1A*), which entails,

$$N(t) = \begin{cases} N_k & t \leq 0 \\ (1-x)N_k e^{-rt} + \frac{N_k}{1 + \frac{1-x}{x}e^{-\gamma t}} & t > 0 \end{cases} \tag{5}$$

We use *Equation 5* to define the evolution of blood concentration of viral RNA sequences which is observed indirectly via noisy viremia measurement data from *Caskey et al., 2015*; *Caskey et al., 2017*; *Baron et al., 2018*. To connect the data with the simple model of viral dynamics in *Equation 5*, we fit the initial frequency of resistant mutants $x$ for each patient separately, and fit a global estimate for the decay rate of susceptible variants $r$ shared across all patients in a trial, using a joint maximum-likelihood procedure. In addition, we fix the growth rate $\gamma$ to 1/3 days, corresponding to a doubling time of approximately 2 days *García et al., 1999*. Our analyses indicate that the initial viremia decline lagged treatment by about 1 day (*Figure 3—figure supplements 1–3*), consistent with previous findings (*Ioannidis et al., 2000*), and therefore, we included a 1 day lag between the fitted viremia response model and the treatment.

$k = n_p(t)$ The number of viral RNA copies in a blood sample is subject to count fluctuations with respect to the true number of circulating virions in a given volume of the blood. We use a Poisson sampling model to define a likelihood for our model of viral population. The likelihood of observing viral counts in a sample collected from patient at time is given by a Poisson distribution,

$$p(k = n_p(t) | \eta = N_p(t)) = e^{-\eta} \frac{\eta^k}{k!} \tag{6}$$

with rate parameter set by the model value of the viral multiplicity $N_p(t)$ (*Equation 5*). We use the Poisson likelihood in *Equation 6* to characterize an error model to fit the parameters of the viral dynamics in *Equation 5*. However, since the mean and variance of the Poisson distribution are related, combining data with different mean values $N_p(t)$ at different times and from different patients can cause inconsistencies in evaluations of errors in our fits. To overcome this problem, we use a variance stabilizing transformation (*McCullagh and Nelder, 2019*) and define a change in variable $\hat{n}_p(t) = \sqrt{n_p(t)}$. This transformed variable has a constant variance, and in the limit of large-sample size, it is Gaussian distributed with a mean and variance given by, $\hat{n}_p(t) \sim \mathcal{N}(\sqrt{\lambda}, 1/4)$. The constant variance of the transformed variable enables us to combine data from all patients and time points, irrespective of the sample's viral loads, and fit the model parameters $(r, x_p)$ using (non-linear) least-squares fitting of the function

$$R(r, \{x_p\}) = \sum_{p:\ \text{patients},t} \left( \sqrt{N_p(t|r, x_p)} - \sqrt{n_p(t)} \right)^2 \tag{7}$$

Here, $N_p(t|r, x_p)$ is the model estimate of viremia in patient $p$ at time $t$ (*Equation 5*), given the pre-treatment fraction of resistant variants $x_p$, and the decay rate $r$.

Note that the viremia measurements have a minimum sensitivity threshold of 20 RNA copies per ml. We treat the data points below the threshold of detection as missing data and if $n_p(t)$ is below the threshold of detection we impute $n_p(t) = \min(20, N_t)$.

The fitted viremia curves for patients enrolled in the three bNAb trials under consideration are shown in *Figure 3—figure supplements 1–3*, and the respective decay rates $r$ for each experiment are,

| trial | 10-1074 | 3BNC117 | Combination | Avg |
|---|---|---|---|---|
| fitted $r$ (days$^{-1}$) | 0.36 | 0.23 | 0.33 | 0.31 |

$$\tag{8}$$

## Individual-based model for viral population dynamics

To encode for different viral variants, we specify a coarse-grained phenotypic model, where a viral strain of type $a$ is defined by a binary state vector $\vec{\rho}^a = [\rho_1^a, \ldots, \rho_\ell^a]$, with $\ell$ entries for potentially

escape-mediating epitope sites; the binary entry of the state vector at the epitope site represents the presence ($\rho_i^a = 1$) or absence ($\rho_i^a = 0$) of a escape mediating mutation against a specified bNAb at site of variant $a$. We assume that a variant is resistant to a given antibody if at least one of the entries of its corresponding state vector is non-zero.

We define an individual-based stochastic birth-death model (**Wilkinson, 2019**) to capture the competitive dynamics of different HIV variants within a population. This dynamic model will allow us to predict the distribution of rebound times under any combination of antibodies.

We assume that a viral strain of type $a$ can undergo one of three processes: birth, death and mutation to another type $b$ with rates $\beta_a$, $\delta_a$, and $\mu_{a \to b}$, respectively:

$$
\begin{aligned}
\text{birth}: \quad & [a] \xrightarrow{\beta_a} 2[a] \\
\text{death}: \quad & [a] \xrightarrow{\delta_a} * \\
\text{mutation}: \quad & [a] \xrightarrow{\mu_{a \to b}} [b]
\end{aligned}
$$

We specify an intrinsic fitness $f_a$ for a given variant a, defined as the growth rate of the virus in the absence of neutralizing antibody or competition. Since bNAbs target highly vulnerable regions of the virus, we expect that HIV-1 escape mutations to be in trinsically deleterious for the virus and to confer a fitness cost relative to the susceptible viral variants prior to the infusion of bNAbs. Assuming that fitness cost of escape is additive across sites and background-independent, we can express the fitness of a variant as , where $\Delta$ is the cost associated with the presence of an escape mutation at site of variant (i.e., for = 1).

We assume that growth is self-limiting via a competition for host T-cells. This competition enforces a carrying capacity, which sets the steady-state population size $N_k$. Competition is mediated through a competitive pressure term $\phi = \frac{\sum_a N_a f_a}{N_K}$ which attenuates the net growth rate $\gamma_a$ so that $\gamma_a = f_a - \phi$. At the carrying capacity, the competitive pressure equals the mean population fitness $\phi = \bar{f}$, making the net growth rate of the population zero.

The net growth rate of a variant $a$ is given by its birth rate minus the death rate: $\gamma_a = \beta_a - \delta_a$. We assume that the total rate of events (i.e. the sum of birth and death events) is equal for all types, that is, $\lambda = \beta_a + \delta_a$, $\forall a$. Assuming that $\lambda$ is constant is to be agnostic about the mechanism of a fitness decrease, attributing fitness loss equally to (i) an increase in the death rate and (ii) a decrease in the birth rate.

Because the absolute magnitude of $\beta$ and $\delta$ asymptotically converge in the continuum limit for a surviving population, that is, $\lim_{N \to \infty} \beta/\delta = 1$, it is impossible to distinguish between (i) and (ii) in the continuous limit. Choosing constant $\lambda$ simplifies both theoretical calculations and the simulation algorithm.

This leads to the following equations for the birth and the death rates:

$$
\beta_i = \frac{\lambda + (f_i - \phi)}{2} \qquad \delta_i = \frac{\lambda - (f_i - \phi)}{2} \tag{9}
$$

In the presence of an antibody, birth is effectively halted for susceptible variants, resulting in birth and death rate for a susceptible variant $s$,

$$
\beta_s = 0 \qquad \delta_s = r \tag{10}
$$

so that the susceptible phenotype decays at rate $r$.

We assume that mutations occur independently at each site,

$$
\mu_{a \to b} = \begin{cases} \mu_i & \text{if } \rho^a - \rho^b = 1_i \\ \mu_i^\dagger & \text{if } \rho^a - \rho^a = -1_s \\ 0 & \text{otherwise} \end{cases} \tag{11}
$$

where $1_s$ is the vector which has only one non-zero entry at site , and $\mu_i$ and $\mu_i^\dagger$ are the forward the backward mutation rates at site , respectively. We characterize the state of a population by vector $\boldsymbol{n} = (n_1, \ldots n_M)$, where $n_a$ is the number of type $a$ variants within the population. We approximate the evolution of the population state distribution using a Fokker-Planck equation (see Appendix 2).

Of special interest for our analysis is the steady-state density of frequencies, which we use to describe the initial state of the population (before treatment) and to infer selection intensity. In the steady state, the population is fluctuating around carrying capacity $\sum_a n_a \approx N_k$ and we can represent the population state via allele frequencies $x_a = n_a/N_k$. In the simple case of a bi-allelic problem, the equilibrium allele frequency distribution $P_{\text{eq}}(x)$ follows the Wright-equilibrium distribution (**Crow and Kimura, 2010**) with modified rates,

$$P_{\text{eq}}(x) = \frac{1}{Z} \frac{e^{\frac{2N_k}{\lambda}(f_1 - f_0)x}(1-x)^{\frac{2N_k}{\lambda}\mu^\dagger} x^{\frac{2N_k}{\lambda}\mu}}{(1-x)x} \equiv \frac{1}{Z(\sigma,\theta,\theta^\dagger)} \frac{e^{-\sigma x}(1-x)^{\theta^\dagger} x^\theta}{(1-x)x} \tag{12}$$

where $Z$ is the normalization factor, $f_1$ is the intrinsic fitness of the variant of interest, $f_0$ is the fitness of the competing variant, $\mu$ and $\mu^\dagger$ are the forward and backward mutation rates, $N_k$ is the carrying capacity, and $\lambda$ is the total rate of events in the birth-death process, which sets a characteristic time scale over which the impact of selection and mutations can be measured. In this case, we can define an 'effective population size' that sets the effective size of a bottleneck and the natural time scale of evolution as $N_e = N_k/\lambda$, and specify a scaled selection factor $\sigma = N_e s = N_e(f_0 - f_1)$, and scaled forward mutation and backward mutation rates (diversity) $\theta = 2N_e\mu$, and $\theta^\dagger = 2N_e\mu^\dagger$. The normalization factor is given by,

$$Z \equiv Z(\sigma,\theta,\theta^\dagger) = \mathcal{B}(\theta,\theta^\dagger) \, {}_1F_1(\theta, \theta + \theta^\dagger, -\sigma) \tag{13}$$

where ${}_1F_1(\cdot)$ denotes a Kummer confluent hypergeometric function and $\mathcal{B}(\theta,\theta^\dagger) = \frac{\Gamma[\theta]\Gamma[\theta^\dagger]}{\Gamma[\theta+\theta^\dagger]}$ is the Euler beta function.

## Extinction Probability

The logistic dynamics describing a patient's viremia over time in **Equation 5** is the deterministic approximation to the underlying birth-death process. However, the resistant population can also go extinct due stochastic effects, which in turn contribute to the probability of late rebound in a population. To capture this effect, we derive an approximate closed form expression for the probability of extinction.

Using the standard birth-death process generating function theory **Allen, 2010** the probability $P(\text{extinct}|n_i)$ that a population consisting of $n_i$ resistant variants of type go extinct can be expressed as,

$$P(\text{extinct}|n_i) = \left(\frac{\delta_i}{\beta_i}\right)^{n_i}. \tag{14}$$

To characterize the probability of extinction for a population of size $N_k$ with pre-treatment fraction of $i^{\text{th}}$ resistant variants $x_i$, we can convolve the extinction probability in **Equation 14** with a Binomial probability density for sampling $n_i$ resistant variants from $N_k$ trials. Given that the pre-treatment fraction of resistant variants is small $x_i \ll 1$ and $N_k$ is large, this Binomial distribution can be well approximated by a Poisson distribution, $\text{Poiss}(n_i; N_k x_i)$ with rate $N_k x_i$, resulting in an extinction probability,

$$\begin{aligned}
P(\text{extinct}|x_i) &= \sum_{n_i} \text{Poiss}(n_i; N_k x_i) \left(\frac{\delta_i}{\beta_i}\right)^{n_i} \\
&= \exp(-N_k x_i) \sum_{n_i} \frac{(N_k x_i)^{n_i}}{n_i!} \left(\frac{\delta_i}{\beta_i}\right)^{n_i} \\
&= \exp\left(-N_k \frac{\beta_i - \delta_i}{\beta_i} x_i\right)
\end{aligned} \tag{15}$$

Using the expressions for the growth in the absence of competition, $\beta_i - \delta_i = \gamma_i = f_i$ (since $\phi = 0$), and assuming that fitness is small relative to the total rate of birth and death events $f_i \ll \lambda$, we can use the approximation $\beta_i = (\lambda + f_i)/2 \approx \lambda/2$, to arrive at,

$$P(\text{extinct}|x_i) \approx \exp\left(\frac{-2N_k f_i}{\lambda} x_i\right) = \exp\left(-\frac{x_i}{x_{\text{ext}}}\right) \tag{16}$$

$x_{\text{ext}}$ where the characteristic escape threshold can be written in terms of concrete genetic observables,

$$x_{\text{ext}} \equiv \frac{\lambda}{2N_k f_i} = \frac{\mu_{\text{ts}}}{f_i} \theta_{\text{ts}}^{-1}. \tag{17}$$

*Figure 3* shows that this threshold can well separate the fate of stochastic evolutionary trajectories, simulated with relevant parameters for intra-patient HIV evolution.

## Numerical simulations of the birth-death process

To treat the full viral dynamics including mutations, and transient competition effects, we can exactly simulate the viral dynamics defined by our individual based model. Below are the key steps in this simulation.

### Population initialization

At the starting point, we set the population size (i.e. the carrying capacity in the simulations) $N_k$ as a free parameter chosen to be large enough to make discretization effects small. The population is then evolved through time using an exact stochastic sampling procedure (the Gillespie algorithm Gillespie (1977)). Simulating the outcome of this stochastic evolution generates the distribution of rebound times and the probability of late rebound—the key quantities related to treatment efficacy.

The input to our procedure is a list of antibodies for which we specify (i) the escape mediating sites for each antibody, and the (invariant) quantities describing (ii) the site-specific cost of escape $\frac{\sigma}{\theta_{\rm ts}}$, and (iii) the forward and backward mutation rates ($\frac{\theta}{\theta_{\rm ts}}, \frac{\theta^\dagger}{\theta_{\rm ts}}$). To simulate the trial outcome for each patient, we use the neutral population diversity $\theta_{\rm ts}$ directly inferred from the patients; see Inference of mutation rates and the neutral diversity within a population. From this, we construct the list of $L$ site parameters (concatenated across all antibodies) for selection and diversity: $\sigma_{1:L}$, $\theta_{1:L}$, $\theta^\dagger_{1:L}$.

We assume that at the start of the simulation, populations are in the steady state and that the potential escape sites are at linkage equilibrium. The approximate linkage equilibrium assumption is justified since the distance between these escape sites along the HIV-1 genome is greater than the characteristic recombination length scale $\approx 100$bp of the virus (*Zanini et al., 2015*). As a result, we draw an independent frequency $x_i$ from the stationary distribution $P_{\rm eq}(x|\sigma_i, \theta_i, \theta^\dagger_i)$ in *Equation 12* to describe the state of a give site , and use these frequencies to construct the initial viral genotypes $\rho^v$ for each virus $v$ in our initial population; see Algorithm 1 (Appendix 5). In simulations, we show that this assumption does not bias our results even when $\theta_{\rm ts}$ is fluctuating and recombination is absent (Section 6.1 and *Figure 4—figure supplement 2*).

To sample from the stationary distribution itself, we define a novel Gibbs-sampling procedure (*Geman and Geman, 1984*) for generating the allele frequencies of the escape variants for the initial state of the population $x \sim P_{\rm eq}(x|\sigma, \theta, \theta^\dagger)$ (*Equation 12*). To characterize this procedure, we expand the exponential selection factor $e^{\sigma(1-x)}$ in the original distribution, which results in,

$$
\begin{aligned}
P_{\rm eq}(x|\sigma, \theta, \theta^\dagger) &= \frac{e^{-\sigma}}{Z(\sigma, \theta, \theta^\dagger)} e^{\sigma(1-x)} \frac{x^\theta (1-x)^{\theta^\dagger}}{x(1-x)} \\
&= \sum_{k=0}^\infty \frac{e^{-\sigma}}{Z(\sigma, \theta, \theta^\dagger)} \frac{\sigma^k}{k!} x^\theta \frac{(1-x)^{\theta^\dagger + k}}{x(1-x)} \\
&\equiv \sum_{k=0}^\infty Q_{\rm eq}(x, k|\sigma, \theta, \theta^\dagger)
\end{aligned}
\tag{18}
$$

Here, $Q_{\rm eq}(x, k|\sigma, \theta, \theta^\dagger)$ is a joint distribution over $(x, k)$, and the desired distribution over the allele frequency $x$ can be achieved by marginalizing the joint distribution over the discrete variable $k$. We can also express the conditional distributions for $x$ and $k$ as,

$$
Q_{\rm eq}(x|k, \sigma, \theta, \theta^\dagger) = \frac{Q_{\rm eq}(x, k|\sigma, \theta, \theta^\dagger)}{\int {\rm d}x\, Q_{\rm eq}(x, k|\sigma, \theta, \theta^\dagger)} = {\rm Beta}(x; \theta, \theta^\dagger + k)
\tag{19}
$$

$$
Q_{\rm eq}(k|x, \sigma, \theta, \theta^\dagger) = \frac{Q_{\rm eq}(x, k|\sigma, \theta, \theta^\dagger)}{\sum_k Q_{\rm eq}(x, k|\sigma, \theta, \theta^\dagger)} = {\rm Poisson}(k; (1-x)\sigma)
\tag{20}
$$

We use these conditional distributions to define a joint Gibbs sampler for $Q_{\rm eq}$. We summarize the resulting $(x, k) \sim Q_{\rm eq}(x, k|\sigma, \theta, \theta^\dagger)$ in the joint Gibbs sampler in Algorithm 2 (Appendix 5). This chain mixes extremely quickly and avoids calculation of the hypergeometric function for the normalization factor (*Equation 20*), which is computationally costly; see Algorithm 2 (Appendix 5).

## Simulation of the evolutionary process

We use a Gillespie algorithm to simulate the evolutionary process, where we break up the reaction calculation into two parts: randomly choosing a viral strain $\rho_i$ from the population and then determining whether it reproduces or dies based on its fitness $f_i$ and escape status; see Algorithm 3 (Appendix 5).

## Determining the simulation parameters of the birth-death process from genetic data

We set the intrinsic growth rate (fitness) of the wild-type virus, in the absence of competition to be $\gamma = (3\,\text{days})^{-1}$, consistent with intra-patient doubling time of the virus (*García et al., 2001*; *Ioannidis et al., 2000*; *García et al., 1999*). We infer the neutralization rate $r$ by fitting the viremia curves (*Figure 3*) in the trials under study, and use the averaged decay rate $r = 0.31$ for simulations, fitted using *Equation 8*. For the absolute mutation rate $\mu_{\text{ts}}$ (per nucleotide per day), we use $1.1 \times 10^{-5}$ which is the average of the reported values for transitions per site per day from *Zanini et al., 2017*. Using the covariance of neutral diversity in twofold and four-fold synonymous sites, we determine the transition/transversion diversity ratio to be $\theta_{\text{ts}}/\theta_{\text{tv}} = 7.8$ (*Figure 2—figure supplement 1*). We use these values to determine the forward and backward mutation rates $\mu_s$ and $\mu_s^{\dagger}$ for each site (see Inference of mutational target size for each bNAb).

Generally, the trial patients show viral populations with larger neutral diversity at the start of the trial compared to the patients enrolled in the high-throughput study of *Zanini et al., 2015* (*Figure 2*). We account for differences in the genetic makeup of the patients enrolled in the trial by directly estimating the neutral diversity $\theta_{\text{ts}}$ from the synonomous site-frequency data of patients, before the start of the trial. The estimated viral diversity $\theta_{\text{ts}}$, coupled with the mutation rate $\mu = 1.1 \times 10^{-5}$ /day / nt, and the total rate of birth and death events in the viral population $\lambda$, set the carrying capacity $N_k$ for a given individual,

$$N_k = \frac{\lambda}{2\mu_{\text{ts}}}\theta_{\text{ts}}. \tag{21}$$

It should be noted that the value of $N_k$, as the number of viruses in our individual-based simulations, is not related to the maximal viral load in the viremia measurements (i.e. steady state copy number per ml) as this relationship depends on the microscopic details of the population dynamics.

The total rate of birth and death events $\lambda = \beta + \delta$ tunes the amount of stochasticity, that is, more events cause noisier dynamics. Notably, stochasticity can be linked to the size of the population $N_k$, which is directly coupled to $\lambda$ (*Equation 21*). We set the value of $\lambda$ self-consistently by requiring the minimum frequency of a variant in our simulations $x_{\text{min}} = 1/N_k = \frac{2\mu_{\text{ts}}}{\lambda}\theta_{\text{ts}}^{-1}$ to be smaller than the escape threshold $x_{\text{ext}} = \frac{\mu_{\text{ts}}}{\gamma}\theta_{\text{ts}}^{-1}$ due to stochasticity (*Equation 17*). We set $\lambda = 2\,\text{day}^{-1}$ so that $x_{\text{min}} = \frac{1}{3}x_{\text{ext}}$. Increasing $\lambda$ results in an increase in the size of population $N_k$ in our simulations, which is computationally costly, without qualitatively changing the statistics of the rebound trajectories.

## Inference of mutation rates and the neutral diversity within a population

Previous work has indicated an order of magnitude difference between the rate of transitions (mutations within a nucleotide class) and transversions (out-class mutations) in HIV (*Nielsen, 2006*; *Zanini et al., 2017*; *Theys et al., 2018*; *Feder et al., 2017*). Therefore, to infer the neutral diversity parameter $\theta_{\text{ts}}$, we also account for the differences between transition and transversion rates.

Consider the set of sequences sampled from a patient's viral population at a particular time. Two neutral alleles that are linked by a symmetric mutational process $\mu_{1\to 2} = \mu_{2\to 1}$ have a simple count likelihood. The probability to see allele 1 with multiplicity $n$ and allele 2 with multiplicity $m$ is given by a binomial distribution $\text{Binom}(n, m|x)$ with parameter $x$ denoting the probability for occurrence of allele 1, convolved with the neutral biallelic frequency distribution $P_{\text{eq}}(x|\sigma = 0, \theta)$ from *Equation 12*. Using this probability distribution, we can evaluate the log-likelihood $\mathcal{L}(\theta|n, m)$ for the neutral diversity $\theta$ given the observations $(n, m)$ for the multiplicities of the two alleles in the population,

$$\mathcal{L}(\theta|n, m) = \log \int dx\, \text{Binom}(n, m|x) \times P_{\text{eq}}(x|\sigma = 0, \theta) = \log \int dx\, \binom{n+m}{n} x^n (1-x)^m \times \frac{x^{\theta-1}(1-x)^{\theta-1}}{Z(\theta)}$$

$$\tag{22}$$

## Inference of netral diversity, mutational target size, and selection from genetic data

To estimate the transition diversity, we only use twofold synonymous sites, and treat each site independently but with a shared diversity parameter $\theta_{ts}$. For example, consider neutral variations for two amino acids glutamine and phenylalanine. The third position in a codon for both of these amino acids are twofold synonymous, as the two possible codons for glutamine are CAG and CAA, and for phenylalanine are TTT, and TTC. Now consider that in the data a conserved glutamine has $n = 3$ G's and $m = 97$ A's in the third codon position, and a conserved phenylalanine has $n = 10$ T's and $m = 90$ C's at its third codon position. In this case, the combined log-likelihood for the shared diversity parameter is $\mathcal{L}(\theta|\text{data}) = \mathcal{L}(\theta|3, 97) + \mathcal{L}(\theta|10, 90)$. Extending to all of sites in the *env* protein, the maximum-likelihood estimator for the transition diversity $\theta_{ts}$ can be evaluated by maximizing the likelihood summed over all conserved twofold synonymous sites,

$$\theta_{ts}^* = \arg\max_{\theta_{ts}} \sum_{\text{two-fold sites}} \mathcal{L}(\theta_{ts}|n = n_A + n_T; m = n_G + n_C) \tag{23}$$

In *Figure 2—figure supplement 1* (panel A), we show that the maximum likelihood estimation method described above has better properties than the more commonly used estimator of the variance $\overline{x(1-x)}$ *Stoddart and Taylor, 1988*.

In a similar way, the likelihood for the transversion $\theta_{tv}$ is determined from polymorphic data at all conserved four-fold synonymous sites. One such example is the third position in a glycine codon, where (GGT, GGC, GGA, GGG) translate to the same amino acid. The maximum-likelihood estimator for the transversions is

$$\theta_{tv}^* = \arg\max_{\theta_{tv}} \sum_{\text{four-fold sites}} \mathcal{L}(2\theta_{tv}|n = n_G + n_T; m = n_C + n_A) \tag{24}$$

The factor of 2 in the argument of the likelihood accounts for the multiplicity of mutational pathways, e.g. from a $G$ nucleotide there are two transversion possibilities, $G \to C$ and $G \to A$ for moving from one allele to the other *Kimura, 1981*.

Using this likelihood approach, we can infer the neutral diversities $\theta_{ts}^*$ and $\theta_{tv}^*$ for each patient at each time point from the polymorphism in twofold and fourfold synonymous sites. To characterize the ratio of transition to transversion rates, we use linear regression on the entire patient population and sample history and infer a constant ratio $\mu_{ts}/\mu_{tv} = \theta_{ts}/\theta_{tv} = 7.8$ (*Figure 2—figure supplement 1B*). In *Figure 2—figure supplement 1*, we also show that the estimate for this ratio is relatively consistent across different data sources, produced even by different sequencing technologies. The previously reported relative rate of transitions to transversions, based on the estimates of sequence divergence along phylogenetic trees of HIV-1 is $\mu_{ts}/\mu_{tv} = 5.6$ (*Zanini et al., 2017*), which is similar to our maximum likelihood estimate.

## Inference of mutational target size for each bNAb

The nucleotide triplets which encode for amino acids at an escape site undergo substitutions which can change the amino acid type and create an escape variant. The changes in the state of an amino acid codon can be modeled as a Markov jump process and can be visualized as a weighted graph where the nodes represent codon states, and edges represent single nucleotide substitutions linking two codon states (*Figure 1D*). In our mutational model, these edges have weights associated with either the mutation rates for transitions $\mu_{ts}$ or transversions $\mu_{tv}$. We call this the *codon substitution graph*.

The codon states can be clustered into three distinct classes: (i) codons which are fatal $F$, (ii) wild-type (i.e. susceptible to neutralization by the bNAb) $W$, and escape mutants (i.e. resistant to the bNAb) $M$. We expect the escape mutants to be at a selective disadvantage compared to the resistant wild-type, and that the most common escape codons to be those which are adjacent to wild-type states.

The mutational target size is determined by the density of paths from the wild-type $W$ to the escape mutants $M$.

$$\mu = \frac{1}{|W|} \sum_{c \in W; d \in M} [c - d = \text{ts}]\mu_{ts} + [c - d = \text{tv}]\mu_{tv} \tag{25}$$

$$\mu^{\dagger} = \frac{1}{|M|} \sum_{c \in W; d \in M} [c - d = \text{ts}]\mu_{\text{ts}} + [c - d = \text{tv}]\mu_{\text{tv}} \tag{26}$$

The functions $[c - d = \text{ts}]$ or $[c - d = \text{tv}]$ are 1 when the two codons are separated by a transition or transversion, and are zero otherwise. Note that since we only have an estimate for the ratio of the transition to transversion rates $\mu_{\text{tv}}/\mu_{\text{ts}}$, we can only determine the scaled mutational target sizes, $\hat{\mu} = \mu/\mu_{\text{ts}}$ and $\hat{\mu}^{\dagger} = \mu^{\dagger}/\mu_{\text{ts}}$, which are sufficient for inference of selection in the next section. The full list of mutational target sizes inferred for the bNAbs in this study are shown in *Appendix 1—table 1*.

When discussing the mutational target size of escape from a given bNAb, we refer to the total mutation rate from the susceptible (wild type) to the escape variant as,

$$\mu = \sum_{i \in \text{ esc. sites}} \mu_{s \to i} \tag{27}$$

where the sum runs over all the mediating escape sites , and can be interpreted as the average number of accessible escape variants. In the strong selection regime, we can write the frequency of escape mutants as,

$$x_{\text{mut}} \approx \sum_i x_i = \sum_i \frac{\mu_{s \to i}}{\Delta_i} \approx \frac{\mu}{\Delta_{\text{hm}}} \tag{28}$$

## Inference of selection for escape mutations against each bNAb

Here, we develop an approximate likelihood approach to infer the selection ratio $\hat{\sigma} = \frac{\sigma}{\theta_{\text{ts}}}$, using the high-throughput sequence data from bNAb-naive HIV-1 patients from *Zanini et al., 2017*. The quantity $\hat{\sigma}$ is a dimensionless ratio which is independent of the coalescence timescale $N_e$, and therefore, represents a stable target for inference.

We assume that the probability to sample $m$ escape mutants and $s$ susceptible (wild-type) alleles at a given site in the genome of HIV in a population sampled from a patient at a given time point follows a binomial distribution $\text{Binom}(m, s|x)$, governed by the underlying frequency $x$ of the mutant allele. In addition, the frequency $x$ of the allele of interest itself is drawn from the equilibrium distribution $P_{\text{eq}}(x|\sigma, \theta, \theta^{\dagger})$ (*Equation 12*), governed by the diversity $\theta_{\text{ts}}$ inferred from the neutral sites, the estimated mutational target sizes $\hat{\mu} = \mu/\mu_{\text{ts}}, \hat{\mu}^{\dagger} = \mu^{\dagger}/\mu_{\text{ts}}$, and the unknown selection ratio $\hat{\sigma} = \sigma/\theta_{\text{ts}}$. As a result, we can characterize the probability $P(m, s|\theta_{\text{ts}}, \hat{\mu}, \hat{\mu}^{\dagger}, \hat{\sigma})$ to sample $m$ escape mutants and $s$ susceptible-type alleles, given the scaled selection and diversity parameters as,

$$
\begin{aligned}
P(m, s|\theta_{\text{ts}}, \hat{\sigma}) &= P(m, s|\sigma = \hat{\sigma}\theta_{\text{ts}}, \theta = \hat{\mu}\theta_{\text{ts}}, \theta^{\dagger} = \hat{\mu}^{\dagger}\theta_{\text{ts}}) \\
&= \int \mathrm{d}x\, \text{Binom}(m, s|x) P_{\text{eq}}(x|\sigma, \theta, \theta^{\dagger}) \\
&= \frac{1}{Z(\sigma, \theta, \theta^{\dagger})} \binom{s+m}{s} \int \mathrm{d}x \frac{e^{-\sigma x} x^{\theta+m}(1-x)^{\theta^{\dagger}+s}}{x(1-x)}
\end{aligned}
\tag{29}
$$

Here, $Z(\sigma, \theta, \theta^{\dagger})$ is a confluent hypergeometric function of the model parameters that sets the normalization factor for the allele frequency distribution $P_{\text{eq}}(x)$ (*Equation 13*). It should be noted that the viral population is in fact out of equilibrium, due to constant changes in immune pressure evolution from the B-cell and T-cell populations (*Nourmohammad et al., 2016*). Although we are ignoring these significant complications, we later use the same equilibrium distribution in a consistent way to generate standing variation in simulations. For the model to make accurate predictions, it is not necessary that the equilibrium model be exactly correct, but only that it is rich enough to provide a consistent description for the distribution of mutant frequencies observed across viral populations.

We will use the probability density in *Equation 29* to define a log-likelihood function in order to infer the scaled selection $\hat{\sigma} = \sigma/\theta_{\text{ts}}$ from data. To do so, we first express the logarithm of this probability density as,

$$
\begin{aligned}
\log P(m, s|\theta_{\text{ts}}, \hat{\sigma}) &= \log P(m, s|\sigma = \hat{\sigma}\theta_{\text{ts}}, \theta = \hat{\mu}\theta_{\text{ts}}, \theta^{\dagger} = \hat{\mu}^{\dagger}\theta_{\text{ts}}) \\
&= \log Z(\sigma, \theta + m, \theta^{\dagger} + s) - \log Z(\sigma, \theta, \theta^{\dagger}) + \text{const.} \\
&= \log \mathbb{E}\left[e^{-\sigma x}\right]_{\text{Beta}(\theta+m, \theta^{\dagger}+s)} - \log \mathbb{E}\left[e^{-\sigma x}\right]_{\text{Beta}(\theta, \theta^{\dagger})} + \text{const.}
\end{aligned}
\tag{30}
$$

where the constant factors (const.) are independent of selection, and $\mathbb{E}[\cdot]_{\text{Beta}(\cdot)}$ denotes the expectation of the argument over a Beta distribution with parameters specified in the subscript. The expression in *Equation 30* implies that we can evaluate the likelihood of selection strength by computing the difference between the logarithms of the expectation for $e^{-\sigma x}$ over allele frequencies drawn from two neutral distributions (Beta distributions), with parameters $(\theta, \theta^{\dagger})$ and $(\theta + m, \theta^{\dagger} + s)$, respectively. This approach is more attractive as it would not require direct evaluation of the confluent hypergeometric functions for the normalization factors in *Equation 29*. Estimating these normalization factors is computationally intensive for large values of $\sigma$, since many terms in the underlying hypergeometric series should be taken into account to stably compute them. However, evaluating the expectations via sampling from these two neutral distributions has the disadvantage that it is subject to variations across simulations. We reduce the variance of our estimate of $\log P(m, s | \sigma, \theta, \theta^{\dagger})$ in *Equation 30* by using a mixture-importance sampling scheme (*Owen and Zhou, 2000*) with the details shown in Algorithm 4 (Appendix 5).

We use data collected across all time points and from all patients to infer reliable estimates for selection strengths. However, allele frequencies are correlated across time points within patients (*Figure 2B*), and thus, sequential measurements are not independent data points. Our estimates indicate a coalescence time of about $N_e \sim 10^3$ days based on the estimates for the mutation rate $\mu_{\text{ts}} = 10^{-5}$ /nt/day, and the neutral diversity $\theta_{\text{ts}} = 2N_e \mu_{\text{ts}} = 0.01$. This coalescence time is much longer that the typical separation between sampled time points within a patient ($\sim 10^2$ days), suggesting that sequential samples collected from each individual in this data are correlated. Therefore, we treat each patient as effectively a single observation, using the time-averaged likelihood for the (scaled) selection factor $\hat{\sigma}$:

$$\mathcal{L}(\hat{\sigma}) = \sum_p \frac{1}{T_p} \sum_t \log P(m_t, s_t | \theta_{\text{ts}}(t), \hat{\sigma}) \tag{31}$$

where $p$ and $t$ denote patient identity and sampled time points, respectively, and $T_p$ is the total number of time points sampled in patient $p$. We use the likelihood in *Equation 31* to generate samples from the posterior distribution for selection strengths under a flat prior, with a standard Metropolis-Hastings algorithm *Hastings, 1970*. Since the prior is constant, this procedure amounts to simply accepting or rejecting samples based on the likelihood ratio of *Equation 31*. We used the centered-normal distribution with standard deviation of 50 ($\times \mu_{\text{ts}}$ in absolute units) as the proposal density for the jumps in the Markov chain.

Prior work has also inferred the fitness effect of mutations in HIV, but our approach differs in important aspects. For instance, maximum entropy models have been used to infer the preference of different amino acids in the *Gag* and the *env* proteins of HIV-1 from their prevalences across sequences sampled from different patients( *Louie et al., 2018*; *Ferguson et al., 2013*). The inferred preference values can explain the in-vitro growth rate (fitness) of the associated viral strains, especially for sites that are relatively conserved and are not the drivers of antigenic evolution in HIV-1. However, further modeling would be needed to quantitively map these inferred amino acid preferences onto population genetics measures that can be used to characterize the evolutionary dynamics of an HIV population. Other work has used longitudinal HIV-1 sequence data to infer selection and to characterize the role of selective sweeps due to immune pressure, genetic hitchhiking and recombination in the turnover of HIV-1 populations within patients (*Zanini et al., 2017*; *Neher and Leitner, 2010*; *Illingworth et al., 2020*; *Haddox et al., 2018*). In contrast, our work focuses on the expected composition of the population in a viremic patient prior to the start of treatment as opposed to the history of a viral population. Our approach uses a self-consistent formalism for inference of the population genetics parameters (e.g. population diversity and selection strength) and for the evolutionary simulations used to predict outcomes. Therefore, we can directly interpret the fitted parameters in terms of both the viral dynamics and the pre-treatment state of HIV-1 populations within patients.

## Predicting trial outcomes from genetically informed evolutionary models

### Predicting rebound times

We expect different distributions of patient outcomes depending on whether they have been recently infected and thus have relatively low viral diversity, or whether their infection is longstanding with

a diverse viral population. To construct the distribution of initial population diversities $\theta_{ts}$ for simulating trial outcomes, we apply the $\theta_{ts}$ inference procedure (**Equation 22**) to pre-treatment sequence datasets available from the clinical trials under consideration. In **Figure 2D** this set of pre-trial $\theta_{ts}$ is compared to the longitudinal in-patient $\theta_{ts}$ from **Zanini et al., 2017**. We used random draws from the inferred $\theta_{ts}$ values for patients to generate $\theta_{ts}$ for simulations.

We found that there was considerably more viral escape and non-responders in our simulations than in the observed data as shown in **Figure 4—figure supplement 3A**. This is *in addition* to the fact that the patients were screened to have only susceptible variants to the antibodies used in trials (**Caskey et al., 2015**; **Caskey et al., 2017**; **Baron et al., 2018**). In theory, there should be zero non-responders, as such patients should have been excluded by screening. The over-prediction of *both* non-responders and late rebounds is a signature of undercounting the effective diversity of the viral populations.

The failure of both screening and our naive prediction in undercounting the diversity in the viral population can be explained by an effective viral reservoir. Viral variants which mediate rebound can come from compartments such as including bone marrow, lymph nodes, and organ tissues, and can be genetically distinct from those sample from the plasma T-cells during screening (**Chaillon et al., 2020**; **Wong and Yukl, 2016**; **Chun et al., 2005**). This reservoir of viral diversity can reappear in plasma after infusion of a bNAb and could in part contribute to treatment failure (**Avettand-Fenoel et al., 2007**; **Shan and Siliciano, 2013**; **Sharkey et al., 2011**; **Tagarro et al., 2018**).

## Determining patient diversity enhancement due to latent reservoirs

We model the effect of reservoirs as a simple inflation of the diversity observed by a multiplicative factor $\xi$. We fit $\xi$ directly to trial observations, using a disparity-based approach by minimizing an empirical divergence estimator **Ekström, 2008** between the observed and simulated data. To do so, we characterize the Hellinger distance **Lindsay, 1994**; **Simpson, 1987** between the true distribution of rebound times $P(t)$ and the rebound times $Q(t|\xi)$ generated by simulations with a given reservoir factor $\xi$,

$$D_H(P(t)\|Q(t|\xi)) = \int dt \, (Q(t|\xi)^{\frac{1}{2}} - P(t)^{\frac{1}{2}})^2 \approx \sum_{(i)}^{n_q} (Q_{(i)}(\xi)^{\frac{1}{2}} - P_{(i)}^{\frac{1}{2}})^2 \tag{32}$$

Algorithm 5 (Appendix 5) defines the procedure that we use to estimate the Hellinger distance $\mathbf{D}_H(\mathbf{Q}(t)\|\mathbf{P}(t|\xi))$. Specifically, we use $n_q$ quantiles of the observed data $x_{(i)} \sim \mathbf{Q}$ to partition the space of observations into discrete outcomes

$$Q_{(i)}(\xi) = \int dt \, P(t|\xi) \left[ t_{(i)} \leq t < t_{(i+1)} \right] P_{(i)} = \frac{1}{n_q}. \tag{33}$$

where $P_{(i)}$ is a constant by construction, and $P_{(i)}(\xi)$ is estimated by simulations, and $[\cdot]$ is the Iverson bracket **Graham et al., 1989**; see Algorithm 5 (Appendix 5).

$$[B] = \begin{cases} 1 \text{ if } B = \text{true} \\ 0 \text{ if } B = \text{false} \end{cases} \tag{34}$$

To simulate data for this analysis, we generate $S$ rebound times $(T_{1:S})$ by simulations, given the scaled diversity values $\xi\theta_{ts}$. We then find the optimal value $\xi^*$ by minimizing the disparity with the observed rebound times $t_{1:p}$ by brute-force search,

$$R(\xi|t_{1:p}) = \sum_{\text{trials}} \text{ReboundDisparity}(t_{1:p}, T_{1:S}|\xi) \tag{35}$$

$$\xi^* = \arg\min_{\xi} R(\xi|t_{1:p}) \tag{36}$$

Here, ReboundDisparity is the function defined by Algorithm 5 (Appendix 5); see **Ekström, 2008** for details. We find the optimal reservoir factor to be $\xi^* = 2.1$, which we use in subsequent therapy prediction. The disparity over various values of $\xi$ for different trials is shown in **Figure 4—figure supplement 3**.

## Simulating outcomes of clinal trials

Given the reservoir-corrected estimate of the diversity $\xi\theta$ and the posterior samples for selection factors $\hat{\sigma}$, we now summarize how we simulated the outcome of clinical trials.

For a given bNAb, we draw the selection factor at each of the escape mediating sites from the corresponding Bayesian posterior on $\hat{\sigma}$; the posterior distributions are shown in *Figure 4A*, and summarized in *Appendix 1—table 1*. We also use the mutational target size (forward $\mu$ and backward $\mu^{\dagger}$ rates) associated with each of the escape mediating sites of a given bNAb; see *Appendix 1—table 1*. The result can be summarized in a mutation / selection matrix $\widehat{M}_a$ for a given bNAb $a$, where each column corresponds to an escape mediating site against the bNAb,

$$\widehat{M}_a = \begin{pmatrix} \hat{\mu}_1 & \dots & \hat{\mu}_i \\ \hat{\mu}_1^{\dagger} & \dots & \hat{\mu}_i^{\dagger} \\ \hat{\sigma}_1 & \dots & \hat{\sigma}_i \end{pmatrix} \tag{37}$$

The elements of the matrix $\widehat{M}_a$ are the scaled mutation and selection factors, i.e., $\hat{\mu}_i = \mu_i/\mu_{\text{ts}}$, $\hat{\mu}_i^{\dagger} = \mu_i/\mu_{\text{ts}}$, and $\hat{\sigma}_i = \sigma_i/\theta_{\text{ts}}$, where the absolute value of mutation rate is set to $\mu_{\text{ts}} = 1.11 \times 10^{-5}$ /nt/day from *Zanini et al., 2017*.

For each patient in our simulated trial, we then draw diversity $\theta_{\text{ts}}$ from the patient pool, and scale it by our fitted $\xi = 2.1$, resulting in patient-specific selection and mutation factors,

$$\xi \times \theta_{\text{ts}} \times \hat{M}_a = \begin{pmatrix} \theta_1 \dots \theta_i \\ \theta_1^{\dagger} \dots \theta_i^{\dagger} \\ \sigma_1 \dots \sigma_i \end{pmatrix} \quad \mu_{\text{ts}} \times \hat{M}_a = \begin{pmatrix} \mu_1 \dots \mu_i \\ \mu_1^{\dagger} \dots \mu_i^{\dagger} \\ \Delta_1 \dots \Delta_i \end{pmatrix} \tag{38}$$

These parameters are then used to initialize the state of an HIV population within a patient according to (Appendix 5), and to determine the absolute rates in Algorithm 3 (Appendix 5) for the population evolution according to *Equation 38*. The decay rate is set to the fitted trial average of $r = 0.31\,\text{days}^{-1}$ (*Equation 8*). The carrying capacity $N_k$ is set according to *Equation 21*. This determines all parameters of the birth-death process simulating the intra-patient evolution of HIV, which are used in Algorithm 3 (Appendix 5).

We evolve a population through time until 56 days have elapsed since treatment, or until the escape fraction relative to the carrying capacity $x_t$ is above 0.8; see Algorithm 3 (Appendix 5). After $x_t > .8$ the evolution is governed by the deterministic equations, and the stochastic simulation ends. The rebound time $T$, defined as the intersection of the exponential envelope and the carrying capacity, can then be calculated analytically as,

$$T = \frac{1}{\gamma} \log \left( 1 + \exp(\gamma t) \frac{1 - x_t}{x_t} \right). \tag{39}$$

The resulting distribution for rebound times are shown as model predictions in *Figure 3B–D*.

Rebound times generated in this fashion were also used to estimate the probability of late rebound to characterize the efficacy of a given bNAb in curbing viral rebound. The probability of late rebound was estimated from $10^4$ simulated patients. The interdecile quantiles (0.1–0.9) of early rebound ($lt_{56}$ days) probability over 200 values of scaled selection coefficients $\hat{\sigma}$ drawn from the posteriors in *Figure 4A* are shown in *Figure 4C*.

## Model robustness

### Effect of genomic linkage on the inference of selection

In our inference of selection (*Equation 31*, *Figure 4A*), we assume that the escape-mediating sites are at linkage equilibrium and that the distribution of allele frequencies can be approximated by a skewed Beta distribution (*Equation 12*), reflecting the equilibrium of allele frequencies. In reality, despite recombination, the HIV genome exhibits linkage effects, especially at nearby sites *Zanini et al., 2015*, and the viral populations experience changing selective pressures by the immune system *Feder et al., 2021*; *Theys et al., 2018*; *Nourmohammad et al., 2019*, and the transient population bottlenecks during therapy *Feder et al., 2016*.

To test the limits on the validity of our inference procedure, we applied it to in silico populations generated by full-genome forward-time simulations (Appendix 5, Algorithm 3) in the presence and

absence of recombination. To do so, we considered an ensemble of ten patients with 100 genomes sampled at 10 time points, and used two diversity parameters $\theta_{ts} = 0.01$ and $\theta_{ts} = 0.1$, to cover the range reflected in patient data (*Figure 2D*, *Figure 4—figure supplement 2*).

One relevant scenario to consider is the impact of other selected sites in the genome on the distribution of alleles at the escape mediating sites against bNAbs. The sites under a strong constant selection are likely to be already fixed (or at a high a frequency) at their favorable state in the population. However, the strong selection on a large fraction of antigenic sites can be thought as time-varying, due to the changing pressure imposed by the immune system or therapy. To capture this effect, we simulated whole genome evolution in which linked sites were under strong selection ($0.1 \times$ growth rate), and where the sign of selection changed after exponentially distributed waiting times (i.e. as a Poisson process); this model of fluctuating selection has been used in the context of influenza evolution *Strelkowa and Lässig, 2012*, and for somatic evolution of B-cell repertoires in HIV patients *Nourmohammad et al., 2019*. The resulted evolutionary dynamics in this case can involve strong selective sweeps and clonal interference due to the continuous rise of beneficial mutations (in the linked sites) within a population.

To test the robustness of our selection inference, we evaluated the distribution of maximum likelihood estimates (MLEs) for the selection values $\hat{\sigma} = \sigma/\theta_t s$ at the escape mediating sites, inferred from the ensemble of sequences obtain from simulated data with linkage. *Figure 4—figure supplement 2* shows that even for fully linked genomes (zero recombination) our MLE estimate of selection has little bias relative to the true values used in the simulations. Adding recombination into the simulations only further attenuates the effect of linkage (*Figure 4—figure supplement 2*), making the estimates more accurate.

The reason that selective sweeps of linked beneficial mutations have only minor effects on our inference of selection for the escape mediating sites is two-fold: First, the primary mechanism by which selective sweeps change the strength of selection at linked sites is via a reduction in the effective population size *Hill and Robertson, 2009*; *Comeron et al., 2008*. However, variations in the effective population size are already accounted for in our inference procedure: The selection likelihood in *Equation 31* is conditioned on the measured neutral-site diversity, $\theta = 2N_e\mu$. The change in the effective population size impacts the selection coefficient $\sigma = 2N_e\Delta$ and the diversity $\theta = 2N_e\mu$ in the same way, and therefore, the (scaled) selection parameter $\hat{\sigma} = \sigma/\theta$ that we infer from data remains insensitive to changes in the effective population size.

The second reason for the robustness of our selection inference to linkage is due to the fact that a beneficial allele in a linked locus can appear on a genetic background with or without a susceptible variant, leading to the rise of either variants in the population. As a results, the impact of such hitch-hiking remains a secondary issue in inference of selection at the escape sites, for which an ensemble of populations from different patients with distinct evolutionary histories of HIV-1are used.

## Robustness of selection inference to intra-patient temporal correlations of HIV alleles

To infer the selection effect of mutations from the longitudinal deep sequencing data of *Zanini et al., 2015*, we use time averaging of the likelihood (*Equation 31*) to avoid conflating our results due to temporal correlations between the circulating alleles within patients (*Figure 2B*). We can view this choice as being one choice among two extremes: (i) to treat each patient as effectively one independent data point so that all patients are given the same weight or (ii) to treat each time point as independent, giving patients with more time points a higher weight. These two choices correspond to different log-likelihood functions for the (scaled) selection factor $\hat{\sigma}$:

$$\mathcal{L}(\hat{\sigma}) = \begin{cases} \sum_{p,t} \log P_p(m_t, s_t | \theta_{ts}(t), \hat{\sigma}) & (t\text{-independent}) \\ \\ \sum_p \frac{1}{T_p} \sum_t \log P_p(m_t, s_t | \theta_{ts}(t), \hat{\sigma}) & (t\text{-averaged}) \end{cases} \tag{40}$$

where $T_p$ is the total number of time points from patient $p$, and $P_p(m_t, s_t | \theta_{ts}(t), \hat{\sigma})$ is the probability to observe $m$ escape mutants, and $s$ susceptible variants at time $t$ in patient $p$, given the neutral diversity $\theta_{ts}(t)$ and the scaled selection factor $\hat{\sigma}$.

We find that both of these approaches result in similar posteriors for selection $\hat{\sigma}$ (Model robustness *Figure 4—figure supplement 1A* and C), although the $t$-averaged likelihood has a higher uncertainty due to fewer independent time points. Thus, our inference of selection is insensitive to the exact choice of the likelihood function given in *Equation 40*, yet our time-averaged approach remains the more conservative choice between the two.

## Statistical significance of the reservoir-corrected diversity

In *Equation 36*, we introduced the reservoir factor $\xi^* = 2.1$ to account for the diversity of HIV-1 that is not sampled from a patient's plasma prior to therapy, which resulted in a better fit of the rebound time distributions (*Figure 3*) compared to a reservoir-free model (*Figure 4—figure supplement 3A*). Here, we quantify the importance of the reservoir factor with a statistical test on the null hypothesis, $\xi_0 = 1$. Specifically, we perform a hypothesis test to test the necessity of using an inflated diversity $\xi \theta_{ts}$ relative to using the bare diversity observed in pre-trial sequence data $\theta_{ts}$. To do so, we construct a disparity-based test statistic *Ekström, 2008*, which is analogous to the likelihood ratio test statistic.

Recall that the optimal reservoir factor $\xi^* = \arg\min_\xi R(\xi|t_{1:p})$ was obtained by minimizing the disparity function $R(\xi|t_{1:p})$ across measurements of rebound times $t_{1:p}$ from all the $p$ patients in data (*Equation 36*). We can estimate the test statistic for the reduction in disparity between the null hypothesis, $\xi_0 = 1$ and the fitted reservoir factor $\xi^*$ as,

$$\Delta_R(t_{1:p}) = R(\xi_0|t_{1:p}) - \min_\xi R(\xi|t_{1:p}). \tag{41}$$

We can then determine the p-value by estimating the quantile of the observed test statistic $\Delta_R(t_{1:p})$ relative to that inferred from the distribution of $\Delta_R(T_{1:p})$ obtained from simulations under null hypothesis $\xi_0 = 1$ (*Fisher, 1956*). Specifically,

$$\text{p-value} = \mathop{\mathbb{E}}_{T_{1:p}|\xi_0} ([\Delta_R(T_{1:p}|\xi_0) > \Delta_R(t_{1:p})]) \tag{42}$$

where $\mathop{\mathbb{E}}_{T_{1:p}|\xi_0} (\cdot)$ denotes the expectation over the rebound times $T_{1:p}$ obtained from 1000 realizations of simulated populations each with $p$ patients, and under the null hypothesis $\xi_0 = 1$ is the Iverson bracket that takes value 1 when its argument is true and 0, otherwise (*Equation 34*). The observed $\Delta_R$ (*Equation 41*), the distribution of simulated values of $\Delta_R(T_{1:p}|\xi_0)$ under the null hypothesis, and the resulting p-value = 0.004 are shown in *Figure 4—figure supplement 3B, C*.

It should be noted that here we use the disparity measure because the corresponding likelihood function for the reservoir factor is inaccessible through forward simulations of populations. However, a general analogy exists between our approach and the more commonly used likelihood approach. Specifically, in an analogous likelihood-ratio test, the test statistic $\Delta_L = \max_\xi \log p(\xi|t_{1:p}) - \log p(\xi_0|t_{1:p})$ would be asymptotically $\chi^2$-distributed with one degree of freedom under the null hypothesis *Fisher, 1954*, and the quantile under the null-hypothesis (p-value) would be estimated by inverting the $\chi^2$ cumulative distribution function (i.e. a $\chi^2$ test).

## Robustness of selection inference to strains from different clades of HIV

The longitudinal deep sequencing data of *Zanini et al., 2015* is collected from 11 patients, 9 of whom are infected with clade B strains of HIV-1, which is the dominant clade circulating in Europe *Spira et al., 2003*. All of the clinical trials we considered *Caskey et al., 2015*; *Caskey et al., 2017*; *Baron et al., 2018* are from patients carrying clade B strains. For the results presented in the main text, we included all the 11 patients in our analysis. Here, we test wether our inference of selection is sensitive to the choice of including or excluding non-clade B patients in our analysis. We therefore repeated our inference procedure for selection by excluding the two non-clade B patients. Model robustness *Figure 4—figure supplement 1A*,B shows a strong agreement between the Bayesian posterior for selection factors in the two cases, with a slight increase in uncertainty for the case with only clade B patients. This increased uncertainty is related to the reduction in sample size by excluding the non-clade B patients from data. Nonetheless, the richness of the intra-patient diversity makes the inference robust to the exclusion of one or two patients.

## Robustness of predictions for trial efficacy to the inferred values of selection strength

How sensitive are the outcomes of our predictions for the rebound time distributions (*Figure 1*) to the exact values of inferred selection strengths we used for our simulations? We addressed this question by performing a disparity analysis similar to that for the diversity $\theta$ in *Equation 41*. Specifically, we assessed whether we might need to rescale our inferred selection strength $\sigma/\theta_{ts}$ by a multiplicative factor $\xi_s$ (*Figure 4—figure supplement 4*).

In contrast to diversity, the reduction in disparity for adjustment of selection with a factor $\xi_s$ is small (*Figure 4—figure supplement 4A*) and not statistically significant (p-value = 0.49; *Figure 4—figure supplement 4B*), and could be attributed to count noise. Still, we cannot discount the possibility that selection was slightly overestimated, possibly due to the effect of compensatory mutations in linked genome, the interplay between the reservoir and the inference of selection, or other biological factors. Nonetheless, in absolute terms, the null hypothesis (i.e. $\xi_s = 1$) cannot be rejected and we have no statistical justification for adding an adjustment factor for selection inferred from untreated patients.

## Robustness of rebound time predictions to methods of identifying escape mutations

Our predictions rely on identifying escape variants against each bNAb, either based on in vivo trial data and patient surveillance, or in vitro DMS assays. To test the sensitivity of our results to these methods, we compare the predictions of rebound time distributions when identifying escape sites from the DMS data versus the trial data. In Appendix 4, we perform this comparison for the 10–1074 and the PGT121 bNAbs; we do not include the 3BNC117 bNAb in this analysis since it targets the CD4 binding site of HIV-1and the DMS data is unreliable for identifying escape variants against it. As shown in *Appendix 4—figure 1*, both trial-inferred and DMS-inferred escape sites result in good predictions for rebound time distributions. However, its appears that using the DMS-inferred escape sites could lead to a more optimistic prediction for treatment success (i.e. a later rebound). This inconsistency may be due to the fact that DMS data is collected in vitro, and other biological factors could be influencing the in vivo escape patterns. Moreover, the differences in the genetic composition of the HIV-1strains circulating in patients enrolled in trials and the strains used for the DMS experiments could lead to different epistatic interactions that can enhance or reduce the chances of escape. For a systematic understanding of these differences and limitations, more experiments would be necessary. Nonetheless, DMS data can provides baseline to gauge the efficacy of different bNAbs for therapy and further in vivo investigation.

## Acknowledgements

We are thankful to Jesse Bloom and Adam Dingens for their input on the Deep Mutational Scanning data of HIV, and Michael Lässig and Matthijs Meijers for helpful discussions. This work has been supported by the NSF CAREER award (grant No: 2045054), DFG grant (SFB1310) for Predictability in Evolution, and the MPRG funding through the Max Planck Society.

## Additional information

### Competing interests

Armita Nourmohammad: Reviewing editor, eLife. The other authors declare that no competing interests exist.

### Funding

| Funder | Grant reference number | Author |
| --- | --- | --- |
| Deutsche Forschungsgemeinschaft | 1310 | Armita Nourmohammad |
| National Science Foundation | 2045054 | Armita Nourmohammad |

| Funder | Grant reference number | Author |
|---|---|---|
| Max Planck Institute for Dynamics and Self-organization | Open access funding | Colin LaMont Armita Nourmohammad |

The funders had no role in study design, data collection and interpretation, or the decision to submit the work for publication.

## Author contributions

Colin LaMont, Conceptualization, Data curation, Formal analysis, Investigation, Methodology, Software, Validation, Visualization, Writing – original draft, Writing – review and editing; Jakub Otwinowski, Conceptualization, Methodology, Writing – review and editing; Kanika Vanshylla, Conceptualization, Data curation, Writing – review and editing; Henning Gruell, Florian Klein, Conceptualization, Writing – review and editing; Armita Nourmohammad, Conceptualization, Formal analysis, Funding acquisition, Investigation, Methodology, Project administration, Resources, Supervision, Writing – original draft, Writing – review and editing

## Author ORCIDs

Henning Gruell http://orcid.org/0000-0002-0725-7138
Armita Nourmohammad http://orcid.org/0000-0002-6245-3553

## Decision letter and Author response

Decision letter https://doi.org/10.7554/eLife.76004.sa1
Author response https://doi.org/10.7554/eLife.76004.sa2

# Additional files

## Supplementary files

- MDAR checklist

## Data availability

The current manuscript is a computational study, so no data have been generated for this manuscript. Reference to the previously published data used in this manuscript is provided. Modelling code is uploaded on GitHub at https://github.com/StatPhysBio/HIVTreatment Optimization, (copy archived at swh:1:rev:0194cf6e554996a066633e99dd53cd5901da552e) and in the Julia package https://github.com/StatPhysBio/EscapeSimulator (copy archived at swh:1:rev:9a343f598820bafddfc7ea4547cefa90bf96fd6e).

The following previously published datasets were used:

| Author(s) | Year | Dataset title | Dataset URL | Database and Identifier |
|---|---|---|---|---|
| Zanini et al | 2015 | Project: PRJEB9618 | https://www.ebi.ac.uk/ena/browser/view/PRJEB9618?show=reads | European Nucleotide Archive, PRJEB9618 |
| Caskey et al | 2015 | HIV-1 isolate 2A1_DB7_02 from USA envelope glycoprotein (env) gene, partial cds | https://www.ncbi.nlm.nih.gov/nuccore/KX016803 | NCBI GenBank, KX016803 |
| Caskey et al | 2017 | Antibody 10-1074 suppresses viremia in HIV-1-infected individuals | https://www.ncbi.nlm.nih.gov/nuccore/KY323724 | NCBI GenBank, KY323724-KY324834 |
| Bar-On et al | 2018 | Safety and antiviral activity of combination HIV-1 broadly neutralizing antibodies in viremic individuals | https://www.ncbi.nlm.nih.gov/nuccore/MH632763 | NCBI GenBank, MH632763 |

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

# Appendix 1

**Appendix 1—table 1.** Selection and mutational target size for escape-mediating sites against each bNAb.

Shown are the sites (column 1) and the susceptible and the escape amino acids (column 2) for each bNAb. We called patterns for 10–1074 and CD4 binding site targetting antibodies VRC01 and 3BNC117 using genetic trial data and the remainder using DMS data. The inferred mutational target size of escape at each site (forward $\mu$ and backward $\mu^\dagger$ mutation rates) is shown in column 3. The major quantiles (10%, 50% (median), 90%) associated with the inferred site-specific Bayesian posterior of the scaled selection strength $\hat\sigma = \sigma/\theta_{\mathrm{ts}}$ are shown in column 4. The corresponding quantiles for the strength of selection, $\sigma = \hat\sigma\theta_{\mathrm{ts}}$ after convolving the posterior for the scaled selection $\hat\sigma$ with the reservoir-corrected intra-patient diversity of HIV-1 $\xi\theta$ are shown in column 5.

| bNAb | site | susceptible AA | escape AA | $\mu$ | $\mu^\dagger$ | $\sigma/\theta$ quantiles 0.1 | 0.5 | 0.9 | $\sigma$ quantiles 0.1 | 0.5 | 0.9 |
|---|---|---|---|---|---|---|---|---|---|---|---|
| 10–1074 | 325 | DN | EGK | 0.76 | 0.51 | 560 | 1,100 | 2,900 | 8.8 | 29 | 93 |
| | 332 | N | DHIKSTY | 2.8 | 0.4 | 190 | 290 | 450 | 2.6 | 7.1 | 16 |
| | 334 | S | AFGINRY | 1.3 | 0.44 | 100 | 170 | 250 | 1.4 | 4 | 9.5 |
| 10E8 | 671 | KNS | RT | 0.41 | 0.51 | 38 | 110 | 220 | 0.68 | 2.5 | 7.3 |
| | 673 | F | LV | 1.4 | 0.47 | 150 | 290 | 510 | 2.4 | 7 | 18 |
| 3BNC117 | 279 | DN | AHK | 0.33 | 0.22 | 190 | 470 | 930 | 3.4 | 11 | 31 |
| | 281 | AP | TV | 1.1 | 1.1 | 62 | 140 | 250 | 0.97 | 3.2 | 8.6 |
| | 282 | KY | ENR | 1.2 | 0.8 | 120 | 230 | 400 | 1.9 | 5.6 | 14 |
| | 459 | G | DN | 0.5 | 1 | 54 | 140 | 270 | 0.91 | 3.3 | 9.5 |
| PG9 | 160 | N | KY | 0.4 | 0.2 | 150 | 280 | 480 | 2.3 | 6.7 | 17 |
| | 162 | ST | AIP | 1.2 | 1.1 | 540 | 1600 | 2600 | 9.5 | 34 | 98 |
| | 169 | GIKMRVW | ELT | 0.53 | 0.92 | 240 | 590 | 1200 | 4.1 | 14 | 41 |
| | 171 | KPR | AEGHNQST | 1.4 | 0.62 | 100 | 190 | 310 | 1.5 | 4.5 | 11 |
| PGT121 | 330 | FHLRSY | Q | 0.12 | 1.4 | 7.4 | 160 | 530 | 0.15 | 3.6 | 16 |
| | 332 | AENV | DIKT | 1.1 | 1.2 | 80 | 180 | 320 | 1.3 | 4.2 | 11 |
| | 334 | DS | GN | 1 | 2 | 15 | 80 | 170 | 0.26 | 1.8 | 5.9 |
| PGT145 | 121 | K | E | 1 | 1 | 34 | 92 | 170 | 0.58 | 2.1 | 5.8 |
| | 160 | N | KY | 0.4 | 0.2 | 140 | 250 | 440 | 2.1 | 6.2 | 15 |
| | 162 | ST | AIP | 1.2 | 1.1 | 680 | 1200 | 2000 | 11 | 29 | 73 |
| | 166 | KR | AEGST | 0.73 | 0.58 | 58 | 120 | 200 | 0.9 | 2.8 | 7.2 |
| | 169 | GIKLTV | EMRW | 0.59 | 1.4 | 8.8 | 80 | 190 | 0.16 | 1.8 | 6.1 |
| PGT151 | 512 | A | GT | 1.1 | 0.57 | 860 | 2,700 | 4100 | 16 | 59 | 160 |
| | 611 | GN | DS | 1.3 | 2 | 360 | 890 | 1300 | 6.5 | 20 | 50 |
| | 613 | ST | CN | 0.28 | 0.7 | 1200 | 2,800 | 3500 | 21 | 58 | 140 |
| | 637 | DN | EKST | 0.83 | 0.41 | 120 | 210 | 350 | 1.8 | 5.1 | 13 |
| | 639 | T | IM | 1 | 1 | 1100 | 1600 | 2600 | 16 | 42 | 96 |
| VRC01 | 197 | DN | S | 0.5 | 1 | 730 | 1600 | 2300 | 12 | 35 | 88 |
| | 279 | DN | AHK | 0.33 | 0.22 | 200 | 420 | 940 | 3.2 | 10 | 30 |
| | 280 | N | D | 1 | 1 | 440 | 1000 | 1800 | 7.3 | 24 | 63 |
| | 281 | AP | TV | 1.1 | 1.1 | 52 | 130 | 250 | 0.9 | 3.1 | 8.6 |
| | 458 | G | D | 0.5 | 1 | 290 | 1200 | 2200 | 6 | 26 | 75 |

*Appendix 1—table 1 Continued on next page*

Appendix 1—table 1 Continued

| bNAb | site | susceptible AA | escape AA | $\mu$ | $\mu^{\dagger}$ | $\sigma/\theta$ quantiles | | | $\sigma$ quantiles | | |
|---|---|---|---|---|---|---|---|---|---|---|---|
| | | | | | | 0.1 | 0.5 | 0.9 | 0.1 | 0.5 | 0.9 |
| | 512 | A | GT | 1.1 | 0.57 | 720 | 1400 | 2900 | 11 | 34 | 92 |
| | 524 | GP | AERS | 1.5 | 0.67 | 45 | 95 | 150 | 0.71 | 2.2 | 5.6 |
| VRC34 | 88 | N | K | 0.26 | 0.26 | 570 | 1400 | 1900 | 9.8 | 30 | 74 |
| | 90 | ST | AEK | 0.48 | 0.8 | 1400 | 2500 | 3300 | 19 | 57 | 130 |

## Appendix 2

### Fokker-Planck description of the birth-death process

The individual-based birth-death model introduced above specifies the stochastic dynamics of a population state over time. We characterize the state of a population by vector $\boldsymbol{n} = (n_1, \ldots n_M)$, where $n_a$ is the number of type $a$ variants within the population. Using the concept of chemical reactions, suitable for Gillespie algorithm *Wilkinson, 2019*; *Gillespie, 2002*, we can determine the propensity $a_r(\boldsymbol{n})$ for a given reaction $r$ (i.e., birth, death, or mutation) in a population of state $\boldsymbol{n}$, which in turn determines the rate at which the reactions occur (*Equation 9*). We denote the resulting change in the state of a population due to reaction $r$ by $\boldsymbol{\nu}_r$. Taken together, the impact of the reactions in the birth-death model can be summarized as, Reaction Rate parameter Propensity State change

| Reaction | Rate parameter | Propensity $a_r(\mathbf{n})$ | State change $v_r$ |
|---|---|---|---|
| Birth | $\beta_a = \frac{\lambda + (f_i - \phi)}{2}$ | $n_a \beta_a$ | $+\hat{e}_i$ |
| Death | $\delta_a = \frac{\lambda - (f_i - \phi)}{2}$ | $n_a \delta_a$ | $-\hat{e}_i$ |
| Mutation | $\mu = \mu_{a \to b}$ ($\mu^\dagger = \mu_{b \to a}$) | $n_a \mu_{a \to b}$ | $+\hat{e}_b - \hat{e}_a$ |

$$(43)$$

where $\hat{e}_i$ is a vector of size $M$ equal to size of the population state vector, in which the $i^{th}$ element equal to one and the rest are zero. For example, a mutation reaction $\mu(a \to b)$ destroys a variant $a$ and creates a variant $b$, resulting in the following change in the state vector,

$$\boldsymbol{v}_{\mu(i \to j)} = -\hat{e}_i + -\hat{e}_j \tag{44}$$

The reactions in *Equation 43* specify a Master equation for the change in the probability of the population state $p(\boldsymbol{n})$,

$$\dot{p}(\boldsymbol{n}) = \sum_r a_r(\boldsymbol{n} - \boldsymbol{\nu}_r) p(\boldsymbol{n} - \boldsymbol{\nu}_r) - a_r(\boldsymbol{n}) p(\boldsymbol{n}) \tag{45}$$

where $\boldsymbol{n} = \sum_i n_i \hat{e}_i$ is the state vector. Using a Kramers-Moyal expansion *Risken, 1989*, we arrive at a Fokker-Planck approximation for the change in the probability distribution of the population state $p(\boldsymbol{n})$,

$$\frac{\mathrm{d}}{\mathrm{d}t} p(\boldsymbol{n}) = \left[ \sum_r \left( \frac{1}{2} \sum_{i,j} \frac{\partial}{\partial n_i} \frac{\partial}{\partial n_j} \nu_r^i \nu_r^j a_r(\boldsymbol{n}) - \sum_i \frac{\partial}{\partial n_i} \nu_r^i a_r(\boldsymbol{n}) \right) \right] p(\boldsymbol{n}) \tag{46}$$

We can identify the drift (i.e. the deterministic force) and diffusion tensors of the Fokker-Planck operator:

$$\boldsymbol{b}(\boldsymbol{n}) = \sum_r \boldsymbol{\nu}_r a_r(\boldsymbol{n}) \qquad \boldsymbol{\Sigma}(\boldsymbol{n}) = \sum_r \boldsymbol{\nu}_r^2 a_r(\boldsymbol{n}) \tag{47}$$

To better demonstrate the structure of this birth-death operator, consider a bi-allelic case (e.g. susceptible and resistant) with a two-dimensional state vector, $\boldsymbol{n} = \begin{pmatrix} n_0 \\ n_1 \end{pmatrix}$. The drift and the diffusion tensors associated with this process follow,

$$\boldsymbol{b}(\boldsymbol{n}) = \begin{pmatrix} f_0 - \frac{n_0 f_0 + n_1 f_1}{N_k} - \mu & \mu^\dagger \\ \mu & f_1 - \frac{n_0 f_0 + n_1 f_1}{N_k} - \mu^\dagger \end{pmatrix} \cdot \boldsymbol{n} \tag{48}$$

$$\boldsymbol{\Sigma}(\boldsymbol{n}) = \lambda \begin{pmatrix} n_0 & 0 \\ 0 & n_1 \end{pmatrix} + \mathcal{O}(\mu) \tag{49}$$

Note that in *Equation 49* we neglect the stochasticity due to mutations since the magnitude of the associated noise is much smaller than the noise due to the birth and death (i.e., genetic drift). From this Fokker-planck equation we directly recover the equilibrium distribution which links the parameters of our model to the genetic observables.

In the steady state, the population is fluctuating around carrying capacity $\sum_a n_a \approx N_k$ and we can represent the population state via allele frequencies $x_a = n_a/N_k$. In the simple case of a bi-allelic problem, the equilibrium allele frequency distribution $P_{\mathrm{eq}}(x)$ follows the Wright-equilibrium distribution *Crow and Kimura, 2010* with modified rates,

$$P_{\mathrm{eq}}(x) = \frac{1}{Z} \frac{e^{\frac{2N_k}{\lambda}(f_1 - f_0)x} (1-x)^{\frac{2N_k}{\lambda}\mu^{\dagger}} x^{\frac{2N_k}{\lambda}\mu}}{(1-x)x} \equiv \frac{1}{Z(\sigma,\theta,\theta^{\dagger})} \frac{e^{-\sigma x}(1-x)^{\theta^{\dagger}} x^{\theta}}{(1-x)x} \tag{50}$$

## Appendix 3

### Incomplete escape of HIV-1 from bNAbs

In our model, we assume that during therapy the susceptible HIV-1 sub-population is completely neutralized by bNAbs and is removed at an exponential rate from the viral population. On the other hand, the resistant variants grow in accordance with their intrinsic fitness in the presence of bNAbs. However, this binary categorization is an approximation, and binding kinetics between viral epitopes and bNAbs implies that neutralization is a probabilistic process and could be incomplete, resulting in the removal of only a fraction of the susceptible variants. The degree of this neutralization depends on the concentration of the infused bNAbs in a patient's serum, and the binding affinity between bNAbs and the susceptible viral epitopes. In a simple model of binding kinetics, the binding affinity is characterized by the half maximal inhibitory concentration, or $IC_{50}$. The probability that a (susceptible) viral variant ' ' does not bind to a neutralizing bNAb with serum concentration [A] is given by,

$$u_i([A]) = \frac{1}{1 + \frac{[A]}{IC_{50}^i}}.$$

(51)

where $IC_{50}^i$ is the half maximal inhibitory concentration of the bNAb against viral variant  . We model the growth rate $\gamma_i([A])$ of the susceptible variant   in the presence of a neutralizing bNAb with concentration [A] as a weighted sum of the viral growth rate in the absence of bNAbs $\gamma_i^0$, and the decay rate $r$ of the virus due to neutralization; the respective weights are set according to the probability that a viral variant is bound (or unbound) to a bNAb (*Equation 51*), which results in,

$$\gamma_i([A]) = u_i([A])\gamma_i^0 + (1 - u_i([A]))r.$$

(52)

In the course of a trial, the concentration of an infused bNAb decays exponentially $[A](t) = [A]_0 e^{-\frac{t}{\tau_a}}$, with a characteristic time of about $\tau_a \simeq 20$ days *Stephenson et al., 2021*; here $[A]_0$ is the concentration of the bNAb upon infusion. By combining *Equation 52* with the bNAb's exponential decay over time, we find that the growth rate of variant   over time is given by

$$\gamma_i(t) = \frac{1}{1 + \frac{[A]_0}{IC_{50}^i} e^{-\frac{t}{\tau_a}}} (\gamma_i^0 - r) + r.$$

(53)

The growth rate of the susceptible variant is determined by the ratio $[A]_0/IC_{50}^i$ between the initial antibody concentration and the half maximal inhibitory concentration of the bNAb.

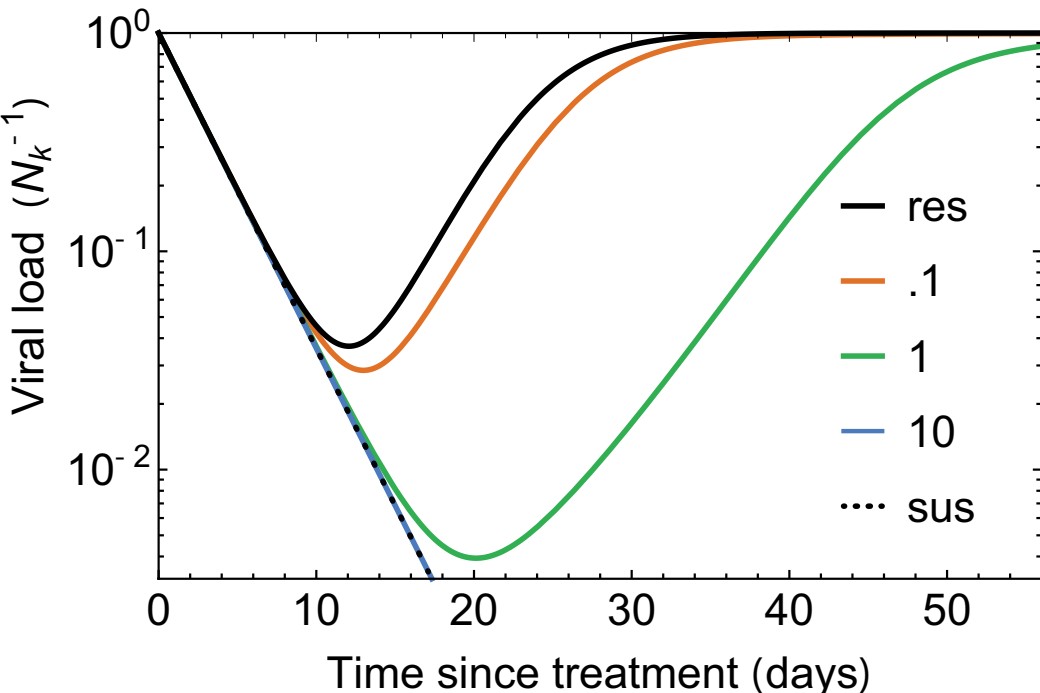

**Appendix 3—figure 1.** The effect of incomplete neutralization. Viral load curves, simulated under the noise-averaged model (deterministic limit) *Equation 4*, are plotted under the assumption of incomplete neutralization for different values of $[A]_0/IC_{50}$ (colors); incomplete neutralization is defined by a changing growth rate given by *Equation 53*. Small values of $[A]_0/IC_{50} \ll 1$ (red) lead to similar trajectories to the perfectly resistant variant (solid black line), while large $[A]_0/IC_{50} \gg 1$ (green line) is indistinguishable from the fully susceptible variants (dotted black line). For all curves, the escaping virus has an initial frequency $x_{res} = 10^{-3}$, with decay rate $r$ and free growth rate $\gamma^0$, $r = \gamma^0 = 1/3$ days.

As shown in *Appendix 3—figure 1*, if the initial antibody concentration is an order of magnitude below the $IC_{50}$(orange line), the dynamics of a patient's viremia closely resembles that of a fully resistant population (solid black line). On the other hand, if the initial bNAb dosage is an order of magnitude above the $IC_{50}$(blue line), the dynamics behaves as though the viral population is completely susceptible (dotted line), resulting in a late viral rebound (more than 8 weeks). We find that incomplete neutralization is most relevant when the initial bNAb concentration is roughly similar to $IC_{50}$, that is, $[A]_0/IC_{50} \approx 1$ (green line). Therefore, our binary categorization is applicable so long as the distribution of $IC_{50}^i$ has low density around initial serum concentrations $[A]_0$. The likelihood of $IC_{50}^i$ matching the antibody dose determines the rate at which our results will be biased by incomplete neutralization.

To compare $IC_{50}$with the initial bNAb concentration, we use the neutralization data from the 10–1074 bNAb trial *Caskey et al., 2017*. For this trial, TZM-bl neutralization assays against different bNAbs were obtained on 114 pseudoviruses expressing envelope proteins derived from circulating viruses in patients on day 0 (55 pseudoviruses) and week 4 (59 pseudoviruses) after 10–1074 infusion. Expectantly, a fraction of viruses from week 4 were resistant (i.e. large $IC_{50}$) to the 10–1074 bNAb used in the trial (orange histogram in *Appendix 3—figure 2A*). However, almost all viruses were susceptible to the (control) 3BNC117 bNAb, which the viruses had not been previously exposed to (orange histogram in *Appendix 3—figure 2B*).

For comparison, the distribution for the initial concentration $[A]_0$ (or equivalently the maximum concentration) of the infused 10–1074 bNAb in patients enrolled in this trial is shown in *Appendix 3—figure 2* (blue histograms), and is peaked around $200 \mu g/ml$. The $IC_{50}$values in this trial are much lower (higher) for susceptible (resistant) variants compared to the initial bNAb concentrations. Therefore, our simplified model assuming that a viral variant is either fully resistant or susceptible to a bNAb (i.e. no incomplete escape) is a reasonable approach for capturing the statistics of treatment failure at the concentrations tested in these trials. Nonetheless, developing a genotype-to-neutralization

model such as the ones developed by *Wagh et al., 2016*; *Yu et al., 2019*, may allow for a more nuanced approach to modeling of neutralization in future work.

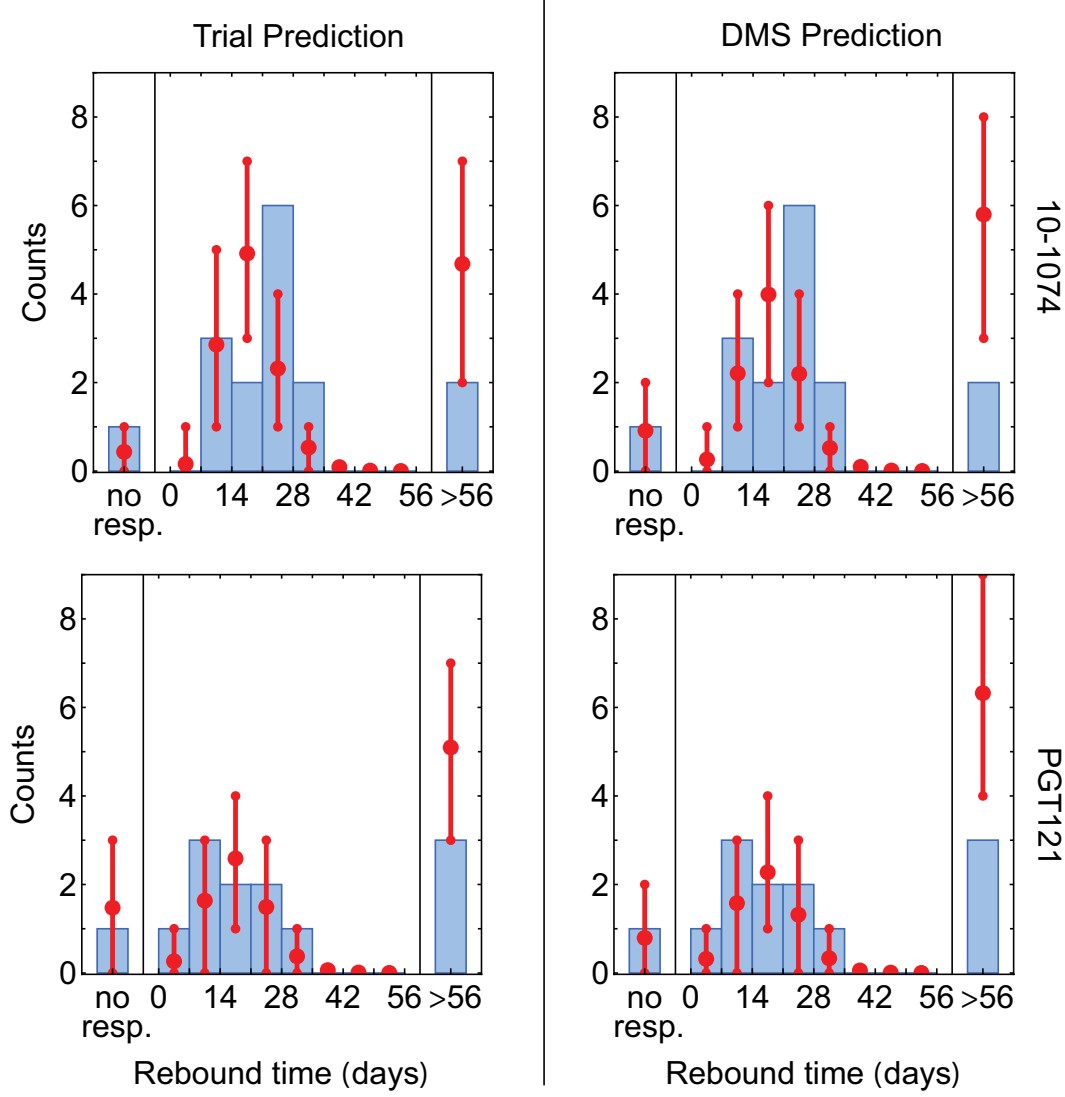

**Appendix 3—figure 2.** The distribution of $IC_{50}$ and $[A]_0$. The distribution of $IC_{50}$ for envelope proteins derived from circulating viruses in patients after 10–1074 infusion is shown against (**A**) 10–1074, (**B**) 3BNC117 as a control. For comparison, the distribution of the maximum (initial) 10–1074 bNAb concentration used in the trial is shown in blue. The maximum measurable $IC_{50} = 50\,\mu g\,$/ml associated with the TZM-bl neutralization assays used in these experiments is indicated in each panel.

## Appendix 4

### Comparison of DMS-inferred and trial-inferred escape pathways

In this appendix we compare the sensitivity of our rebound time predictions to the methodology used to identify sites of escape (i.e., data from DMS experiments versus therapy trials). To do so, we focus on two bNAbs, 10–1074 and PGT121 for which we have access to both the trial *Caskey et al., 2017*; *Stephenson et al., 2021* and the DMS data *Dingens et al., 2019*; *Dingens et al., 2017*; *Schommers et al., 2020*. The inferred resistant and susceptible variants at the identified sites for both of these bNAbs based on their respective trials and the DMS data are reported in *Appendix 4—table 1*.

Although there is a lot of commonalities among the two approaches, the two methods can be sensitive to different mutational pathways in certain sites. This is in part expected as the DMS procedure can generate many rare variants, but all on a common genetic background background; in this case the experiments scan only single point mutations away from the HIV-1 strain BF520. W14M.C2 *Dingens et al., 2017*. On the other hand, the clinical trial data shows escape variants on the genetic background of a diverse viral population circulating in a patient, and the fate of a mutation can be strongly determined by the epistatic interactions with the background genome that it appears on.

**Appendix 4—table 1.** Escape variants inferred from trial and DMS data.

Tables show the sites mediating escape of HIV-1 from the 10–1074 and the PGT121 bNAbs, using the respective trial data *Caskey et al., 2017*; *Stephenson et al., 2021* (left) and the DMS data *Dingens et al., 2017* (right). The susceptible (sus) and resistant (res) variants at each site are identified according to the procedure detailed in the Methods. Sites that are identified as escape-mediating by only one of the methods are marked with (*). While substitutions at site 325 were called as escape mediating based on the PGT121 trial data *Stephenson et al., 2021* but are missed by the DMS data, this site was only inconsistently associated with escape, and its associated escape variants usually co-occur with other escape substitutions; see the Extended Data Figure 3 in *Stephenson et al., 2021*. Sites are numbered according to HXB2.

| | 10–1074 trial | | | 10–1074 trial | |
|---|---|---|---|---|---|
| HXB2 site | sus | res | HXB2 site | sus | res |
| 325 | DN | EGK | 325 | DN | EGK |
| 330* | FHLQSYR | *1,731 | 330 | FHLQSYR | R |
| 332 | N | DHIKSTY | 332 | N | ADEIKTV |
| 334 | S | AFGINRY | 334 | S | DGN |

| | PGT121 trial | | | PGT121 DMS | |
|---|---|---|---|---|---|
| HXB2 site | sus | res | HXB2 site | sus | res |
| 325 | D | NKT | 325* | DNKT | * |
| 330* | FHLQSYR | *1,733 | 330 | FHLRSY | Q |
| 332 | NV | DSRTI | 332 | AENV | DIKT |
| 334 | TS | RND | 334 | DS | GN |

To assess the sensitivity of our analyses, we make predictions for the distribution of rebound times, using the escape pathways identified based on the DMS versus the trial data in *Appendix 4—table 1*. Overall, we see good agreements between the two approaches, but DMS-inferred escape pathways predict slightly too many patients with late rebound ($T_p > 56$ days) compared to the trial-based inference, i.e., DMS-based predictions are more optimistic (*Appendix 4—figure 1*). This deviation is likely to be related to the diversity of viral genetic backgrounds in clinical trials, since more escape pathways could be realized through positive epistasis, leading to a faster viral rebound.

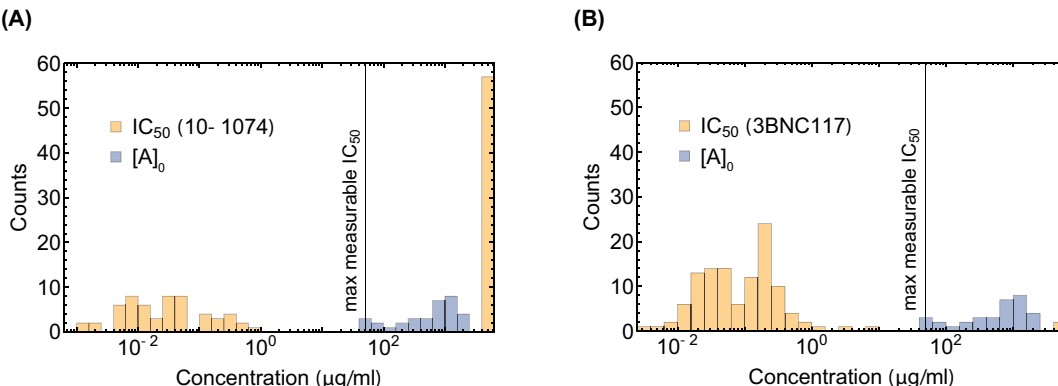

**Appendix 4—figure 1.** Statistics of viral rebound with using escape pathways inferred from the DMS and the trial data. We compare the distribution of rebound times in patients from the clinical trials with 10–1074 *Caskey et al., 2017* (top) and PGT121 *Stephenson et al., 2021* (bottom) bNAbs, using escape pathways inferred from the respective trial data (left) and the DMS data *Dingens et al., 2017* (right). The error bars show the inter decile range (0.1–0.9 quantiles) generated by the simulations for the corresponding trial. The PGT121 trial predictions relied on the neutral diversity estimates $\theta_{ts}$ from the other three trials *Caskey et al., 2015*; *Caskey et al., 2017*; *Baron et al., 2018* due to the relatively limited genetic data available from this study. The fitted reservoir value of $r_{resv.} = 2.07$ is used for all predictions.

# Appendix 5

# Simulation algorithms

---

**Algorithm 1. Population initialization**

---

**procedure** Population Initialization $(N_k, \sigma_{1:L}, \theta_{1:L}, \theta_{1:L}^\dagger)$

▷ Generates a population of size of $N_k$ from
equilibrium. $\theta$, and $\sigma$ are the parameters defining the
equilibrium values of the population state. Returns the
initial vector of genotypes
 **for** $i \in 1 : L$ **do**
 $x_i \sim P_{\text{eq}}(x|\sigma_i, \theta_i, \theta_i^\dagger)$
 **end for**
 **for** $v \in 1 : N_k$ **do** $\rho_i^v \sim \text{Bernoulli}(x_i)$
 **end for**
**return** $\rho^{1:N_k}$
**end procedure**

---

**Algorithm 2. Gibbs Sampler for Allele Frequencies**

---

**procedure** Equlibrium Sampler $(\sigma, \theta, \theta^\dagger | \text{Samples})$

▷Generates a stream of non-independent but rapidly mixing samples $X \sim p(x|\sigma, \theta, \theta^\dagger)$. Default BurnIn is 10.

 $N \leftarrow \text{Samples} + \text{BurnIn}$
 $K_0 \leftarrow \text{Round}(\sigma)$
 **for** $n \in 1 : N$ **do**
 $X_{n+1} \sim \text{Beta}(\theta, \theta^\dagger + K_n)$ ▷Sample the mutant fraction
 $K_{n+1} \sim \text{Poisson}(\sigma(1X_{n+1}))$ ▷Sample the auxiliary parameter
 **end for**
**return** $X_{\text{BurnIn}:N}$ ▷Return only the mutant frequency $X$ part of the chain (marginalize over $k$)
**end procedure**

---

**Algorithm 3. Population time step**

---

**procedure** EvolvePopulation $(t, \rho^{1:N}|\lambda, \gamma, r,)$
▷ Acts on a time $t$ and a list of $N$ genotypes $\rho^{1:N}$. Inherits dependency on other parameters from the fitness
function $F(G)$ and the **Mutate**$(G)$ operator which depend on $\Delta_{1:L}, \mu_{1:L}, \mu_{1:L}^\dagger$ and $\gamma$ and the population diversity
measure $\theta_{ts}$.
 $\phi \leftarrow \frac{1}{N} \sum_i F(g_i)$

 $t' \leftarrow t + \frac{\text{RandExp}()}{\lambda N}$

 $i \sim \text{Rand}(1 : N)$

 $G \leftarrow \rho^i$
 **if** IsEscaped$(G)$ **then** ▷ If the virus is escaped
 $D \sim \text{Bernoulli}(\frac{\lambda - F(G) + \phi}{2\lambda})$ ▷ Determine if the virus dies ($D = \textbf{true}$) or lives
$(D = \textbf{false})$.
 **if** $D$ **then**
 $N' \leftarrow N - 1$ ▷ Delete genotype at position
 $\rho^{1:N'} \leftarrow \rho^{1:\tilde{i}:N}$
 **else**
 $N' \leftarrow N + 1$ ▷ Duplicate genotype at position
 $\rho^{1:N'} \leftarrow \text{Append}(\rho^{1:N}, G)$
 **end if**
 **else** ▷ If the virus is neutralized
 $D \sim \text{Bernoulli}(\frac{r}{\lambda})$ ▷ remove it at the appropriate rate
 **if** $D$ **then**
 $N' \leftarrow N - 1$ ▷ Delete genotype at position
 $\rho^{1:N'} \leftarrow \rho^{1:\tilde{i}:N}$
 **end if**
 **end if**
 $j \sim \text{Rand}(1 : N')$ ▷ Choose a random virus to mutate
 $\rho^j \leftarrow \text{Mutate}(\rho^j)$ ▷ Apply mutation operator with intensity $\mu/\lambda$
**return** $(t', \rho^{1:N'})$ ▷ Return the new time and the new population.
**end procedure**

---

## Inference algorithms

---

**Algorithm 4. Importance sampled log-likelihood given a single datapoint**

---

**procedure** SigmaLikelihood $(\hat{\mu}, \hat{\mu}^\dagger, m, s, \theta_{ts}, N)$ ▷ Takes mutant $m$ and susceptible $s$ counts for a particular site at a single timepoint. Returns an approximate log-likelihood function $l(\hat{\sigma}) = \log P(m, s|\theta_{ts}, \hat{\sigma}) + c$, up to an additive constant. SigmaLikelihood is a *closure* that returns a one-parameter function. We used N =103 samples.

$\quad \theta \leftarrow \hat{\mu}\theta_{ts}$

$\quad \theta^\dagger \leftarrow \hat{\mu}^\dagger \theta_{ts}$

$\quad$ **for** $i \in 1 : N$ **do**

$\quad\quad x_i \sim \text{Beta}(\theta, \theta^\dagger)$ ▷ Sample from the neutral distribution

$\quad\quad w_i \leftarrow \frac{\mathcal{B}(\theta+m,\theta^\dagger+s)}{\mathcal{B}(\theta,\theta^\dagger)} \frac{1}{x_i^m(1-x_i)^w}$ ▷ Importance weight ratio

$\quad\quad y_i \sim \text{Beta}(\theta + m, \theta^\dagger + s)$ ▷ sample from the neutral distribution,

conditioned on the observations

$\quad\quad v_i \leftarrow \frac{\mathcal{B}(\theta+m,\theta^\dagger+s)}{\mathcal{B}(\theta,\theta^\dagger)} \frac{1}{y_i^m(1-y_i)^w}$ ▷ Importance weight ratio

$\quad$ **end for**

$\quad Z_0(\hat{\sigma}) := \frac{1}{N}\sum_i e^{\hat{\sigma}\theta_{ts}x_i}\frac{1}{1+w_i} + \frac{1}{N}\sum_i e^{-\hat{\sigma}\theta_{ts}y_i}\frac{1}{1+v_i}$ ▷ importance sampling mean

$\quad Z_1(\hat{\sigma}) := \frac{1}{N}\sum_i e^{\hat{\sigma}\theta_{ts}x_i}\frac{1}{1+w_i^{-1}} + \frac{1}{N}\sum_i e^{-\hat{\sigma}\theta_{ts}y_i}\frac{1}{1+v_i^{-1}}$ ▷ Note the inversion in the weighting

factor compared to $Z_0$.

**return** $l(\hat{\sigma}) := \log Z_1(\hat{\sigma}) - \log Z_0(\hat{\sigma})$ ▷ Return a log-likelihood function. The same random variable realizations $x_{1:N}$ and $y_{1:N}$ are cached in memory and used for each function evaluation, making $l(\hat{\sigma})$ continuous and differentiable.

**end procedure**

---

**Algorithm 5. Rebound-time Disparity**

---

**procedure** ReboundDisparity $(t_{1:P}, T_{1:S})$

▷ Takes the observed rebound times $t_{1:P} \sim Q$ from a set of trial patients and simulated late rebound times $T_{1:S} \sim P$ and returns a disparity estimator.

$\quad$ ▷ First estimate the probabilities in the truncated observation categories

$\quad P_{(NR)} \leftarrow \frac{1}{P}\sum_p [t_p < 1\text{day}]$ ▷ Count the fraction of non-responders in trial

$\quad Q_{(NR)} \leftarrow \frac{1}{S}\sum_s [T_s < 1\text{day}]$ ▷ ... and in simulation

$\quad P_{(LR)} \leftarrow \frac{1}{P}\sum_p [t_p \geq 56\text{ days}]$ ▷ Count the fraction of late rebounds in trial

$\quad Q_{(LR)} \leftarrow \frac{1}{S}\sum_s [T_s \geq 56\text{ days}]$ ▷ ... and in simulation

$\quad$ ▷ Then construct a histogram over the continuous data-points (i.e. $t \in [0, 56]$) and estimate the probability in each bin

$\quad t_{1:P'} = \text{SortAscending}(Filter[1 \leq t < 56](t_{1:p}))$ ▷ Select only the observed rebound times

$\quad t_0 \leftarrow -\infty$ and $t_{P'+1} \leftarrow \infty$

$\quad$ **for** $p \in 1 : P'$ **do**

$\quad\quad Q_{(p)} \leftarrow \frac{1}{P}$ ▷ By design, each histogram bin contains one observed data point, and gets 1 /P mass

$\quad\quad P_{(p)} \leftarrow \frac{1}{S}\sum_s [\max(1, \frac{1}{2}(t_{p-1} + t_p)) \leq T_s < \min(56, \frac{1}{2}(t_{p+1} + t_p))\text{ days}]$ ▷ Use the midpoints of

adjacent points to construct the boundaries of histogram bins, and determine probability mass in each bin.

$\quad$ **end for**

**return** $P \sum_{i\in NR,LR,1:P'}(Q_{(i)}^{1/2} - P_{(i)}^{1/2})^2$ ▷ Return the discretized estimate of the Hellinger distance, scaled by the number of patients

**end procedure**

---

