## [Editor Report]

This paper will be of interest to scientists within the fields of statistical and biological physics, immunology, and vaccinology. The mathematical/statistical framework is rigorously constructed based on key concepts from population genetics and high-throughput viral genetic sequence data. The results provide important insights into the failures of past treatment regimens with broadly neutralizing antibodies to suppress viral escape in clinical trial participants. The results also present exciting and highly testable predictions of improved treatment strategies for combatting HIV through passive bnAb immunization.

---

## [Decision Letter]

**Decision letter after peer review:**

Thank you for submitting your article "Design of an optimal combination therapy with broadly neutralizing antibodies to suppress HIV-1" for consideration by *eLife*. Your article has been reviewed by 3 peer reviewers, including Kayla Sprenger as Reviewing Editor and Reviewer #1, and the evaluation has been overseen by Miles Davenport as the Senior Editor.

Essential revisions:

1) The text should be modified to more clearly convey the novelty of the current work, and to better place the current work within the context of the existing literature on bnAb therapy design both from an experimental and computational perspective. Specifically, the conceptual advances that this work offers in terms of methodology and/or success in therapy design over past studies should be made more clear, and additional citations/discussion of existing literature is warranted.

For example, combination therapy with more than two bnAbs vs. fewer has long been shown to be more effective in suppressing early rebound. A suggestion is to note this fact and discuss the benefits of clinical corroboration of the model's results – i.e., that this corroboration provides a basis for using the model to interrogate new bnAbs (for which DMS data is available), and/or the combinatorial explosion of higher order cocktails for which we cannot possibly test all combinations.

2) Related to the above note, the non-independence of bnAbs is an important point that should be discussed/acknowledged as a potential shortcoming of the model.

3) As presented, the deep mutational scanning data is not convincing in its ability to fully characterize the spectrum of escape mutations, given the typical inability of DMS data to probe the low-frequency variants that primarily mediate rebound. More work needs to be done here to ensure/make clear that the predictive power of the approach is strong in this regard.

4) More discussion should be focused around specific use cases of the work. Particularly, since the framework is most predictive for early rebound times, it should be emphasized that the results from this work would best translate to, e.g., the design of short term suppressive treatments, rather than therapy efficacy over longer timescales, which would require a more complete characterization.

5) More discussion of how the predicted fitness costs compare to the older fitness landscape work would be useful context in the discussion.

*Reviewer #1 (Recommendations for the authors):*

General comments/questions:

In the Abstract (and Discussion), it is stated that bnAbs may serve as an alternative strategy to antiretroviral therapy against HIV, yet in the Introduction it is stated that augmenting antiretroviral therapy with bnAbs may be a fruitful strategy against HIV. Perhaps highlight the more promising approach of the two for consistency?

Line 52 states that prior work on designing bnAb therapies did not consider viral dynamics, yet the following sentence appears to give theoretical examples of just this ("dynamics of viremia"). Was the first sentence referring to prior experimental work?

In general, I find the paragraph in the Introduction between lines 52-72 to be a bit unfocused; what are the authors wishing to communicate here? Specifically, regarding lines 58-72, it seems like one could skip straight from line 57 to 73 and it would flow just as well.

Line 93: are the authors referring to cancer cells, or cancer-causing pathogens? Do such extensive longitudinal sequence datasets also exist for resistant bacteria and cancer-causing pathogens as for HIV, that would enable translation of this approach to those settings?

Line 106-107, Equation 1 appears to model the virus population size as a function of time (N(t)), rather than the rebound time explicitly (perhaps reword slightly?).

Line 119: in defining the actual rebound time, T=-γ^-1*logx, it would seem that this is a constant, since γ is a constant, and x is also an inferred constant. If this is correct, why is equation 1 needed? Likely, I am just missing something here!

Line 135: it might be helpful to talk more physically about what is meant by the 'birth' and 'death' of a variant (variant completely dies out? Or the variant population declines as a result of bnAb infusion?). Does a single amino acid mutation move a variant a to another type b?

Line 155: Is not the mutation rate of HIV well known already? Is the rate inferred by the authors in line with past estimates?

Line 156-157: The authors state that the 'fastest process in the dynamics…is the growth rate of susceptible viruses". Should this not say 'resistant viruses", since γ = 1/3 days^-1^ vs. 3 days^-1^ for susceptible viruses, or "slowest process"?

Line 174: "mutational target size" is not very intuitive for me. I know it is defined later on lines 211-213, but perhaps this can be summarized or stated more intuitively here, as well as in the Discussion?

Lines 178-182: the authors propose to infer mutation and fitness characteristics of escape-mediating HIV variants via sequence data. Work out of Arup Chakraborty's lab at MIT comes to mind on developing HIV fitness landscapes. Would this not offer the same information for parameterizing the birth-death model? Perhaps the potential relevance/connection between those past works and the current work could be briefly discussed.

In the section on 'Diversity of the viral population', it seems counterintuitive to me that characterizing only synonymous mutations as a measure of neutral genetic diversity would allow one to determine the 'chance to observe a rare (e.g., resistant) mutation' (line 185), since this implies a change in the amino acid sequence of the protein. I must not be thinking about this correctly; can the authors please clarify this for me?

Lines 222-223: The footprint of a bnAb is often larger than the actual epitope, which may contain both variable and conserved sites (e.g., the CD4 binding site). Thus, bnAbs must bind to both residue types. Given this, why do the authors expect HIV escape mutations to be intrinsically deleterious for the virus (implying they occur mostly at conserved sites), vs. occurring at the nearby variable sites that bnAbs also contact? For example, the following paper, https://www.pnas.org/content/115/4/E564, states that "Even for the relatively "conserved" CD4 binding-site region, only mutations at some residues are predicted to incur large fitness costs upon mutations."

Line 314: what do the authors classify as 'early rebound'? Below 56 days?

Line 330-332: the first part of this statement makes sense to me (that fitness-limiting bnAbs like PGT151 are best against high-diversity viral populations), but I am not certain about the second part. Wouldn't fitness-limiting and mutation-limiting bnAbs be equally good against low-diversity viral populations?

Lines 378-379: Might it be possible for the authors to speculate on how the exclusion of the dynamics of Ab concentration and IC50 neutralization during treatments could affect their results/conclusions?

*Reviewer #2 (Recommendations for the authors):*

In Appendix 1, the authors might consider coloring the resistant and susceptible amino acid identities based on the data source (DMS, patient-derived, crystallography).

---

## [Author Response]

Essential revisions:1) The text should be modified to more clearly convey the novelty of the current work, and to better place the current work within the context of the existing literature on bnAb therapy design both from an experimental and computational perspective. Specifically, the conceptual advances that this work offers in terms of methodology and/or success in therapy design over past studies should be made more clear, and additional citations/discussion of existing literature is warranted.For example, combination therapy with more than two bnAbs vs. fewer has long been shown to be more effective in suppressing early rebound. A suggestion is to note this fact and discuss the benefits of clinical corroboration of the model's results – i.e., that this corroboration provides a basis for using the model to interrogate new bnAbs (for which DMS data is available), and/or the combinatorial explosion of higher order cocktails for which we cannot possibly test all combinations.

We thank the reviewers for this comment. We have now reorganized and extended the introduction and the discussion to better reflect prior work in both fitness inference and therapy optimization.

In the introduction, we elaborate on various modeling and experimental approaches for bNAb mono and combo therapy design, and describe how our approach is distinguished from the existing work. We have made the following changes in the introduction:

“We discuss the modeling and machine learning techniques trained on experimental data from neutralization assays against pseudo-viruses to characterize the efficacy of bNAbs and their combinations against different variants of HIV [R1–R3]. These modeling approaches to optimization view the infection as a static collection of viral strains to be neutralized as opposed to an actively evolving population. We then discuss the mechanistic models that have been developed to explain the dynamics of viremia in patients following passive infusion of bNAbs [R4–R9]. These detailed models use trial data to fit parameters in relation to a bNAb’s efficacy in clearing virions, reducing viral load, etc. These detailed mechanistic models cannot easily generalize from one trial to another in order to predict the efficacy of a new bNAb mono- or combination therapy. We describe how evolution of the HIV population is another key factor to consider in modeling the dynamics of viremia in response to therapy with ART or bNAb. We then present our approach as a coarse-grained evolutionary model of viral response to bNAb infusion that uses genetic data of HIV-1 in untreated patients to predict bNAb therapy outcome by characterizing the chances of viral escape from a given bNAb. Although our model does not accurately reproduce the detailed dynamics of viremia in each patient and lacks the mechanistic insight of richer models proposed previously, it can accurately predict the distribution of viral rebound times in response to passive bNAb infusions – a key measure of efficacy for a bNAb therapy trial. We then emphasize that our prediction for the viral rebound time in response to a bNAb relies on only a few patient-specific parameters (i.e., the genetic diversity of patients prior to treatment), and is primarily done based on the inferred genetic parameters from the deep sequencing of HIV-1 populations in a separate cohort of ART-naive patient. Therefore, we argue that our model could be used to guide therapy trial design by identifying optimal combinations of bNAbs to suppress evolutionary escape of HIV-1 in patients.”

In the Discussion we have added the reviewer’s suggestion to the text to argue how our approach can be used to identify combo therapies. From the Discussion section (lines 419-426):

“Combination therapy with more than two bNAbs (or drugs in ART) has long been shown to be more effective in suppressing early HIV rebound, both in theory and practice [R1,R2,R4,R10–R12]. In addition to corroborating this conclusion quantitatively, we provide a method for assessing new bNAbs for which escape mutations are known. Our method can be understood as a tool to navigate the combinatorial explosion of higher order cocktails for which we cannot possibly test all combinations. By assessing the evolvability of resistance against different combinations we can identify the best therapies to target for clinical trial. Specifically, we show that to suppress the chance of viral rebound to below 1%, we show that a combo-therapy with 3 bNAbs with a mixture of mutation- and selection-limited strategies that target different regions of the viral envelope is necessary. Such combination can counter the full variation of viral diversity observed in patients. We found that PG9, PG151, and VRC01, which respectively target V2 loop, Interface, and CD4 binding site of HIV envelope, form an optimal combination for a 3-bNAb therapy to limit HIV-1 escape in patients infected with clade B of the virus.”

2) Related to the above note, the non-independence of bnAbs is an important point that should be discussed/acknowledged as a potential shortcoming of the model.

We thank the reviewers for pointing this out. Indeed, using bNAbs that target the same (or structurally adjacent) epitopes in a therapy could lead to physical interference during neutralization, and in those cases the assumption of independence can well be violated. The sharing of mutational pathways invalidates the assumption of independence and may make our predictions too optimistic. Still, our model correctly finds that escape from combinations of bNAbs that target the same epitope is easier due to the fewer necessary mutations. We have now expanded our discussions on this matter in the manuscript:

From section Devising optimal bNAb therapy cocktails in the main text (lines 375-386):

“Escape from bNAbs in multivalent therapy is also influenced by the non-independence of escape pathways. Several of the bNAbs (e.g. 10-1074 and PG121) in this study target the same (or structurally adjacent) epitopes and escape from them is mediated by the same mutations. The sharing of mutational pathways invalidates the assumption of independence and may make our predictions too optimistic. However, the best performing antibody combinations in Figure 4C target multiple epitopes to reduce the chances of collective escape. Therefore, the assumptions of independent fitness effects and independent mutational pathways in this case are not consequential for the main predictions of our model (i.e., the choice of bNAb combinations). in vitro data shown in [R13] suggest that additive and independent effects are the norm, but that small but consistent synergistic effects may imply that our assumption of site-wise independence is conservative. More data on HIV escape from combinations of bNAbs would be informative for further modeling efforts and relevant for long-term therapy design.”

3) As presented, the deep mutational scanning data is not convincing in its ability to fully characterize the spectrum of escape mutations, given the typical inability of DMS data to probe the low-frequency variants that primarily mediate rebound. More work needs to be done here to ensure/make clear that the predictive power of the approach is strong in this regard.

We agree with the reviewers that the utility and limitations of the DMS data should be more clearly demonstrated. We have included a new Appendix (Appendix 4; Appendix 4 Figure 1; Appendix 4 Table 1) to compare predictions of rebound time distributions when identifying escape sites from the DMS data versus the trial data. We performed this comparison for the 10-1074 bNAb for which we have access to both the DMS and the trial data. We also included an analysis for a new therapy trial dataset with the PGT121 bNAb [R9]; in the previous version of this manuscript we only had access to the DMS data to characterize escape against PGT121. We should note that two other trial datasets are available for the 3BNC117 and VRC01 bNAbs. However, both of these bNAbs target the CD4 binding site of the env protein and, as noted in the manuscript, the DMS experiments in the presence of these bNAbs are very noisy and cannot be used to reliably identify the escape sites; see Appendix 4 for details.

In addition, we have included a discussion on this matter in the Methods section Model robustness, subsection Robustness of rebound time predictions to methods of identifying escape mutations (lines 1104-1121):

“Our predictions rely on identifying escape variants against each bNAb, either based on in-vivo trial data and patient surveillance, or in-vitro DMS assays. To test the sensitivity of our results to these methods, we compare the predictions of rebound time distributions when identifying escape sites from the DMS data versus the trial data. In Appendix 4 we perform this comparison for the 10-1074 and the PGT121 bNAbs; we do not include the 3BNC117 bNAb in this analysis since it targets the CD4 binding site of HIV-1and the DMS data is unreliable for identifying escape variants against it. As shown in Appendix 4-Figure 1, both trial-inferred and DMS-inferred escape sites result in good predictions for rebound time distributions. However, its appears that using the DMS-inferred escape sites could lead to a more optimistic prediction for treatment success (i.e., a later rebound). This inconsistency may be due to the fact that DMS data is collected in-vitro, and other biological factors could be influencing the in-vivo escape patterns. Moreover, the differences in the genetic composition of the HIV-1strains circulating in patients enrolled in trials and the strains used for the DMS experiments could lead to different epistatic interactions that can enhance or reduce the chances of escape. For a systematic understanding of these differences and limitations, more experiments would be necessary. Nonetheless, DMS data can provides baseline to gauge the efficacy of different bNAbs for therapy and further in-vivo investigation.”

We have also included a paragraph in the Discussion section, Limitations subsection on this point (lines 472490):

“Our predictions are limited by our ability to identify the escape variants for each bNAb, either based on the in-vivo trial data and patient surveillance, or in-vitro assays such as DMS experiments. In Appendix 4 we compare the accuracy of our predictions for the rebound time distributions of HIV using DMS-inferred versus trial-inferred escape variants against the 10-1074 and the PGT121 bNAbs, and we find good agreements between the two approaches in both cases. However, it should be noted that identifying escape variants from DMS experiments lead to a more optimistic prediction for treatment success during the first 8 weeks of trials (i.e., they suggest a later rebound). One reason for this discrepancy may be related to the distinct genetic composition of viruses in the two approaches. DMS experiments characterize HIV escape by introducing mutations on a single genetic background [R14–R16] (e.g. the HIV-1 strain BF520.W14M.C2 in [R14]), whereas clinical data contain diverse populations of viruses between and within individuals. It is more likely for diverse viral populations to contain variants in which positive epistasis between escape mutations and the background genome is present. Therefore, it is reasonable to expect that genetic data originating in clinical trials show more pathways of escape, resulting in a faster viral rebound following bNAb infusion. Nevertheless, DMS experiments are less costly for identifying escape variants compared to trials, and they provide a baseline to assess the efficacy of different bNAbs for therapy and further in-vivo investigations. More experiments would be necessary for a more systematic understanding of the limitations of each approach.”

4) More discussion should be focused around specific use cases of the work. Particularly, since the framework is most predictive for early rebound times, it should be emphasized that the results from this work would best translate to, e.g., the design of short term suppressive treatments, rather than therapy efficacy over longer timescales, which would require a more complete characterization.

We agree with the reviewers that we should emphasize the utility of our approach in designing short-term treatments, which we have included in the Discussion section. We have also added a discussion about combining ART and bNAb therapy for longer term treatments.

From the Discussion section (lines 433-455):

“The statistical agreement between our coarse-grained model and the observed distribution of the rebound times (Figure 3-B) implies that many of the mechanistic details are of secondary importance in predicting viral escape. Nonetheless, our approach falls short of reproducing the detailed characteristics of viremia traces in patients, especially at very short or very long times, during which the dynamics of T-cell response or the decay of bNAbs could play a role [R17–R19]. The relationship between the short-term suppression of the virus, which is the focus of this analysis, and the long-term treatment success is complicated by reestablishment of HIV from the latent reservoirs and different modes of intra-host HIV evolution [R20,R21].

One strategy to achieve a longer term treatment success is by combining bNAb therapy with ART. One main advantage of bNAb therapy is the fact that it can be administered once every few months, in contrast to ART, which should be taken daily and missing a dose could lead to viral rebound. Although multivalent bNAb therapy reduces the chances of short-term viral escape, viral escape remains a real obstacle for longer term success of a treatment with bNAbs. Alternatively, (fewer) bNAbs can be administered in combination with ART [R22,R23], whereby ART could lower the replication rate of the HIV population, reducing the viral diversity and the chances of viral escape. Specifically, we can expect that emergence and establishment of rare (i.e., strongly deleterious) escape variants against bNAbs to be less likely in ART+ patients, which suggests that fitness-limited bNAbs should be more effective in conjunction with ART. Still, more data would be necessary to understand the longterm efficacy of such augmented therapy, and specifically the role of viral reservoirs in this context. A modeling approach could then shed light on how ART administration and bNAb therapy could be combined to efficiently achieve viral suppression.”

Moreover, we also discuss aspects of therapy optimization (e.g. safety) that must be considered when identifying the most promising therapy.

From the Discussion section (lines 491-498):

“It should be noted that our analysis in [figure 4c (antibody ranking)] only focuses on one aspect of therapy optimization, i.e., the suppression of escape. Other factors, including potency (neutralization efficacy) and half-life of the bNAb, or the patient’s tolerance of bNAbs at different dosage should also be taken into account for therapy design. For example, the bNAb 10E8, which we identified as of the most promising mono-therapy candidates in Figure 4, is shown to be poorly tolerated by patients with short half-life [R24], making it undesirable for therapy purposes. Thus, the bNAb candidates shown in Figure 4C should be taken as a guideline to be complemented with further assessment of efficacy and safety for therapy design.”

5) More discussion of how the predicted fitness costs compare to the older fitness landscape work would be useful context in the discussion.

We thank the reviewers for pointing this out to us. We have included a comparison of our approach to other fitness inference work in the Methods section, Inference of selection for escape mutations against each bNAb (lines 902-919):

“Prior work has also inferred the fitness effect of mutations in HIV, but our approach differs in important aspects. For instance, maximum entropy models have been used to infer the preference of different amino acids in the Gag and the env proteins of HIV-1 from their prevalences across sequences sampled from different patients [R25,R26]. The inferred preference values can explain the in-vitro growth rate (fitness) of the associated viral strains, especially for sites that are relatively conserved and are not the drivers of antigenic evolution in HIV-1. However, further modeling would be needed to quantitively map these inferred amino acid preferences onto population genetics measures that can be used to characterize the evolutionary dynamics of an HIV population. Other work has used longitudinal HIV-1 sequence data to infer selection and to characterize the role of selective sweeps due to immune pressure, genetic hitchhiking and recombination in the turnover of HIV-1 populations within patients [R27–R30]. In contrast, our work focuses on the expected composition of the population in a viremic patient prior to the start of treatment as opposed to the history of a viral population. Our approach uses a self-consistent formalism for inference of the population genetics parameters (e.g. population diversity and selection strength) and for the evolutionary simulations used to predict outcomes. Therefore, we can directly interpret the fitted parameters in terms of both the viral dynamics and the pre-treatment state of HIV-1 populations within patients.”

Reviewer #1 (Recommendations for the authors):General comments/questions:In the Abstract (and Discussion), it is stated that bnAbs may serve as an alternative strategy to antiretroviral therapy against HIV, yet in the Introduction it is stated that augmenting antiretroviral therapy with bnAbs may be a fruitful strategy against HIV. Perhaps highlight the more promising approach of the two for consistency?

We thank the reviewer for pointing out this inconsistency to us. We have now included a paragraph in the Discussion section about augmenting ART with bNAb Therapy (lines 442-455):

“One strategy to achieve a longer term treatment success is by combining bNAb therapy with ART. One main advantage of bNAb therapy is the fact that it can be administered once every few months, in contrast to ART, which should be taken daily and missing a dose could lead to viral rebound. Although multivalent bNAb therapy reduces the chances of short-term viral escape, viral escape remains a real obstacle for longer term success of a treatment with bNAbs. Alternatively, (fewer) bNAbs can be administered in combination with ART [R22,R23], whereby ART could lower the replication rate of the HIV population, reducing the viral diversity and the chances of viral escape. Specifically, we can expect that emergence and establishment of rare (i.e., strongly deleterious) escape variants against bNAbs to be less likely in ART+ patients, which suggests that fitness-limited bNAbs should be more effective in conjunction with ART. Still, more data would be necessary to understand the longterm efficacy of such augmented therapy, and specifically the role of viral reservoirs in this context. A modeling approach could then shed light on how ART administration and bNAb therapy could be combined to efficiently achieve viral suppression.”

Line 52 states that prior work on designing bnAb therapies did not consider viral dynamics, yet the following sentence appears to give theoretical examples of just this ("dynamics of viremia"). Was the first sentence referring to prior experimental work?

We have fully restructured and extended the introduction in response to the Q #1, from essential revisions. We have also extended the paragraph that the reviewer is referring to: we have now separate discussions on the experimental work to measure the breadth of bNAbs for optimal therapy design without modeling the dynamics of viremia, and the mechanistic modeling and theoretical work to describe the details of the viremia dynamics in trial data. We believe these modifications have clarified the points conveyed in the introduction; see the response to Q#1 from essential revisions and the marked changes in the introduction.

In general, I find the paragraph in the Introduction between lines 52-72 to be a bit unfocused; what are the authors wishing to communicate here? Specifically, regarding lines 58-72, it seems like one could skip straight from line 57 to 73 and it would flow just as well.

We thank the reviewer for pointing this out and apologize for the confusing text. As noted in response Q #1 from essential revisions, we have restructured the Introduction to address this point.

Line 93: are the authors referring to cancer cells, or cancer-causing pathogens? Do such extensive longitudinal sequence datasets also exist for resistant bacteria and cancer-causing pathogens as for HIV, that would enable translation of this approach to those settings?

We clarified the language and explicitly mention “cancer tumor cells” in the manuscript.

Still, the reviewer’s question about cancer-causing pathogens is very interesting. There are various longitudinal studies for hepatitis C virus (HCV). The evolutionary rate of HCV is also comparable to that of HIV [R31]. Moreover, there is growing evidence that (naturally emerging) broadly neutralizing antibodies play an important role in immune-mediated control of hepatitis C virus (HCV) infection [R32]. Therefore it is likely that our modeling approach would be of relevance for therapy design against HCV. We have now added the example of HCV in the introduction.

Longitudinal data for HBV and HPV (two other cancer-causing pathogens) are more limited, but it would be interesting to collect and study such data.

Line 106-107, Equation 1 appears to model the virus population size as a function of time (N(t)), rather than the rebound time explicitly (perhaps reword slightly?).Line 119: in defining the actual rebound time, T=-γ^-1*logx, it would seem that this is a constant, since γ is a constant, and x is also an inferred constant. If this is correct, why is equation 1 needed? Likely, I am just missing something here!

The reviewer is correct that equation 1 is the viral population size over time N(t). Specifically, this equation refers to the total viral population without reference to type, in the continuous deterministic limit (i.e. ignoring the noise terms in equation 2.) of the stochastic individual-based model. We simulate treatment outcomes using the full stochastic birth-death model, but we use this deterministic limit to analyze observed trajectories of patient viremia. Fitting N(t) is the basis for measuring the characteristic rebound time T and then for connecting the rebound time to the initial frequency of resistant variants x. This quantity is patient-specific—it is a single number characterizing the patient’s viremic response to treatment, but it varies from patient to patient. We have added clarifying wording and added patient-indicating subscripts to the parameters to make this clear (lines145-147):

“The maximum-likelihood fits of N(t) to the viremia measurements in Figure 3—figure supplement 1-3 specifies the initial resistant fraction xp and thus the rebound time Tp in each patient p, which in this simple model, is given by Tp = −γ−1 logxp.”

Line 135: it might be helpful to talk more physically about what is meant by the 'birth' and 'death' of a variant (variant completely dies out? Or the variant population declines as a result of bnAb infusion?). Does a single amino acid mutation move a variant a to another type b?

As an individual-based model, it is not birth and death of a variant, but rather birth and death of an individual member of a variant class (or generally an individual within a population). We have modified the language in the section Model: Stochastic evolutionary dynamics of HIV subject to bNAb therapy to clarify when we are describing the dynamics for individual virions (lines 164-169):

“At each generation, a virus with phenotype a can undergo one of three processes: birth, death and mutation to another type b, with rates βa, δa, and µa→b, respectively (Figure 1-B). The net growth rate of the viral subpopulation with phenotype a is the birth rate minus the death rate, γa = βa − δa (Figure 1-C). The total rate of events (birth and death) per virion λ = βi +δi modulates the amount of stochasticity in this birth-death process, which we assume to be constant across phenotypic variants.”

Regarding “Does a single amino acid mutation move a variant a to another type b?”: The amino acids at each site are grouped into two categories: susceptible and resistant. Replacing a susceptible amino acid with a resistant amino acid changes the type and fitness of the virus, but if the substitution replace an amino acid with another from the same category, the type of the virus is unchanged. Note that furthermore, a change of type does not necessarily move a virus between the susceptible and resistant classes. A virus is resistant to a given therapy if it has at least one escape mutation against each antibody, for all antibodies in a given therapy; see figure 1E for the details regarding the mutational model.

Line 155: Is not the mutation rate of HIV well known already? Is the rate inferred by the authors in line with past estimates?

The reviewer is correct that the mutation rates are known, both in-vitro and in-vivo. We were interested in the transition/transversion ratio as it manifests in formulation of the frequency spectrum (equation 12 in Methods, page 17). This ratio, rather than mutation rate itself, is necessary to model the frequency distribution of resistant viruses. We compare the result of our analysis to the estimate from [R33] in Methods (Page 21, as well as Figure 2-Supplement 1B-E) and find it in agreement. Nonetheless, for consistency, we used our likelihood estimates for the transition/transversion ratio in our further analysis, since we measured this ratio using the standing variation in untreated patients, and that our model is concerned with how mutational processes maintain resistant variants in steady state (standing variation) of a population. In [R33] and elsewhere, mutation rates are often measured using the divergence along a lineage over time.

Line 156-157: The authors state that the 'fastest process in the dynamics…is the growth rate of susceptible viruses". Should this not say 'resistant viruses", since γ = 1/3 days^-1^ vs. 3 days^-1^ for susceptible viruses, or "slowest process"?

The reviewer is correct, and we have fixed the typo.

Line 174: "mutational target size" is not very intuitive for me. I know it is defined later on lines 211-213, but perhaps this can be summarized or stated more intuitively here, as well as in the Discussion?

We have added a brief intuitive description at this line (now lines 204-205):

“The mutational target size for escape from the bNAb (i.e., the number of paths leading to escape, weighted by their respective probabilities) …”

Lines 178-182: the authors propose to infer mutation and fitness characteristics of escape-mediating HIV variants via sequence data. Work out of Arup Chakraborty's lab at MIT comes to mind on developing HIV fitness landscapes. Would this not offer the same information for parameterizing the birth-death model? Perhaps the potential relevance/connection between those past works and the current work could be briefly discussed.

We thank the reviewer for pointing this out to us. Please see our response to Q#5 from the essential revisions.

In the section on 'Diversity of the viral population', it seems counterintuitive to me that characterizing only synonymous mutations as a measure of neutral genetic diversity would allow one to determine the 'chance to observe a rare (e.g., resistant) mutation' (line 185), since this implies a change in the amino acid sequence of the protein. I must not be thinking about this correctly; can the authors please clarify this for me?

The reviewer is right in that neutral diversity alone is not enough to determine the “chance to observe a rare (e.g. resistant) mutation.” The neutral diversity is only part of our larger model that also includes the mutational targets size and the fitness effects. We have made the wording of this section more precise (lines 216-218):

“The neutral genetic diversity θ = 2Nkµ/λ (i.e., the number of segregating alleles) is an observable that relates to key population genetics parameters, i.e., the per-nucleotide mutation rate µ, the population carrying capacity Nk, and the total number of events per virus in the birth-death process λ, which determines the noise amplitude in the evolutionary dynamics (Methods). Nk and λ together determine the effective population size Ne = Nk/λ.”

To further clarify why neutral diversity is important for the inference of the non-synonomous substitution frequency: Note that the transition-mediated diversity θts = 2Neµts, is proportional to the effective population size Ne = Nk/λ with a fixed, known timescale µts. Because the value of Ne is shared across synonymous and non-synonymous sites, we can achieve higher statistical precision by determining Ne through θts using the synonymous sites, and then independently fitting the fitness differences from the frequency of non-synonymous substitutions.

Lines 222-223: The footprint of a bnAb is often larger than the actual epitope, which may contain both variable and conserved sites (e.g., the CD4 binding site). Thus, bnAbs must bind to both residue types. Given this, why do the authors expect HIV escape mutations to be intrinsically deleterious for the virus (implying they occur mostly at conserved sites), vs. occurring at the nearby variable sites that bnAbs also contact? For example, the following paper, https://www.pnas.org/content/115/4/E564, states that "Even for the relatively "conserved" CD4 binding-site region, only mutations at some residues are predicted to incur large fitness costs upon mutations."

Yes, the variable regions mediate escape. However even the variable regions are not completely free from stabilizing fitness. We see that the distribution of amino acids at the escape sites we identify have much reduced entropy compared to the most diverse regions of env, the so-called hyper-variable regions (see the spreadsheet at https://www.hiv.lanl.gov/components/sequence/HIV/featuredb/search/env_ab_search_pub.comp for entropy estimates along gp160). There must therefore be stabilizing selection at these escape sites, reducing the amino-acid distribution entropy. At the same time, we agree that the strength of this selection need not be large. Note the selection we infer is only strong compared to mutation, and the relative fitness costs on absolute growth rate are generally not observable in the lab (perhaps 1% of the wild-type growth rates.)

The intrinsic fitness cost is not assumed, it is determined from the frequency data. It is possible for the inference process to return zero or even negative fitness costs and to include such costs in the simulation. However, it is certainly true that for a bNAb to be broad, escape must be rare. It is this feature which means that our fitness inference (which depends on observed frequencies) is generally expected to return a positive fitness cost of escape (or else infer an extremely small mutational target size). Otherwise escape alleles would be too common in sera and neutralization panels to be identified as broad in the first place. Therefore, our inferred valued cost values are consistent with this picture.

Line 314: what do the authors classify as 'early rebound'? Below 56 days?

Yes, we have clarified this term in the manuscript now. Early rebound refers to any rebound time occurring within 56 days of the start of therapy.

Line 330-332: the first part of this statement makes sense to me (that fitness-limiting bnAbs like PGT151 are best against high-diversity viral populations), but I am not certain about the second part. Wouldn't fitness-limiting and mutation-limiting bnAbs be equally good against low-diversity viral populations?

In the limit of very low diversity, the effect of selection is very weak since selection can act only on the circulating alleles. Therefore, the frequencies of resistant variants are determined by the mutational target size of each variant, irrespective of the fitness cost. It is therefore more advantageous to use a mutation-limiting antibody against a low-diversity viral population, rather than a selection-limiting antibody.

Lines 378-379: Might it be possible for the authors to speculate on how the exclusion of the dynamics of Ab concentration and IC50 neutralization during treatments could affect their results/conclusions?

We thank the reviewer for bringing up this interesting point. We have now added a new appendix (Appendix 3) discussing the impact of incomplete neutralization on viral rebound. We have also added language in the Discussion section, Limitations (page 12; lines 464-471):

In our model of viral escape, we neglect the possibility of incomplete escape of the virus due to the reduced neutralization efficacy of bNAbs as their concentrations decay during trials. In Appendix 3, we show that this simplifying assumption is valid as long as the IC50 is not the same order of magnitude as the initial dosage concentration of the infused bNAb. Notably, the data from therapy trials used in this study fall into the regime for which we can neglect the impact of incomplete neutralization (Appendix 3-Figure 2). However, taking into account the dependence of viral fitness on bNAb concentration and its neutralization efficacy, as in the model proposed by [R34], could improve the long-term predictive power of our approach.

Moreover, in Appendix 3 we explore the effects of incomplete neutralization on rebound trajectories. As we show in Appendix 3-Figure 1, if an antibody has an IC50 against the viral variant which is an order of magnitude above the initial antibody concentration, the viral dynamics very closely follows the idealized “escaped” trajectory (i.e., with complete neutralization). On the other hand, for an IC50 an order of magnitude below the initial concentration, the viral dynamics behave similarly to a completely neutralized virus, with a late rebound (later than 8 weeks). We found that the most important effect of incomplete neutralization on the dynamics of viremia occurs when the antibody has an IC50 against a resistant variant that is roughly of similar magnitude to the initial bNAb concentration in a patient’s serum; see Appendix 3-Figure 1. In Appendix 3-Figure 2 we show the distribution of IC50 and the initial bNAb concentration from the 10-1074 trial [R35] to see how often we would expect IC50 and initial concentration to be of the same order of magnitude. We find that the IC50 values in this trial are much lower (higher) for susceptible (resistant) variants compared to the initial bNAb concentration in all patients. Therefore, our simplified model assuming that a viral variant is either fully resistant or susceptible to a bNAb (i.e., no incomplete escape) is a reasonable approach for capturing the statistics of treatment failure at the concentrations tested in these trials. Nonetheless, developing a genotype-to-neutralization model such as the ones in [R1,R2] may allow for a more nuanced approach to characterize neutralization in future work.

Reviewer #2 (Recommendations for the authors):In Appendix 1, the authors might consider coloring the resistant and susceptible amino acid identities based on the data source (DMS, patient-derived, crystallography).

We hope the added section on DMS comparison (the new Appendix 4) clarifies this since it covers the cases where sites are called by more than one method.

References:

[R1] Wagh K, Bhattacharya T, Williamson C, Robles A, Bayne M, et al. (2016) Optimal Combinations of Broadly Neutralizing Antibodies for Prevention and Treatment of HIV-1 Clade C Infection. PLOS Pathogens 12: e1005520.

[R2] Yu WH, Su D, Torabi J, Fennessey CM, Shiakolas A, et al. (2019) Predicting the broadly neutralizing antibody susceptibility of the HIV reservoir. JCI Insight 4: e130153.

[R3] Mayer BT, deCamp AC, Huang Y, Schiffer JT, Gottardo R, et al. (2022) Optimizing clinical dosing of combination broadly neutralizing antibodies for HIV prevention. PLOS Computational Biology 18: e1010003.

[R4] Perelson AS, Neumann AU, Markowitz M, Leonard JM, Ho DD (1996) HIV-1 Dynamics in vivo: Virion Clearance Rate, Infected Cell Life-Span, and Viral Generation Time. Science 271: 1582–1586.

[R5] Rong L, Feng Z, Perelson AS (2007) Emergence of HIV-1 Drug Resistance During Antiretroviral Treatment. Bulletin of Mathematical Biology 69: 2027–2060.

[R6] Rong L, Dahari H, Ribeiro RM, Perelson AS (2010) Rapid emergence of protease inhibitor resistance in hepatitis C virus. Science translational medicine 2: 30ra32.

[R7] Tomaras GD, Yates NL, Liu P, Qin L, Genevieve G Fouda, et al. (2008) Initial B-Cell Responses to Transmitted Human Immunodeficiency Virus Type 1: Virion-Binding Immunoglobulin M (IgM) and IgG Antibodies Followed by Plasma Anti-gp41 Antibodies with Ineffective Control of Initial Viremia. Journal of Virology 82: 12449–12463.

[R8] Cardozo-Ojeda EF, Perelson AS (2021) Modeling HIV-1 Within-Host Dynamics After Passive Infusion of the Broadly Neutralizing Antibody VRC01. Frontiers in Immunology 12: 710012.

[R9] Stephenson KE, Julg B, C Sabrina Tan, Zash R, Walsh SR, et al. (2021) Safety, pharmacokinetics and antiviral activity of PGT121, a broadly neutralizing monoclonal antibody against HIV-1: A randomized, placebocontrolled, phase 1 clinical trial. Nature Medicine 27: 1718–1724.

[R10] Feder AF, Rhee SY, Holmes SP, Shafer RW, Petrov DA, et al. (2016) More effective drugs lead to harder selective sweeps in the evolution of drug resistance in HIV-1. *eLife* 5: e10670.

[R11] Klein F, Halper-Stromberg A, Horwitz JA, Gruell H, Scheid JF, et al. (2012) HIV therapy by a combination of broadly neutralizing antibodies in humanized mice. Nature 492: 118–122.

[R12] Mendoza P, Gruell H, Nogueira L, Pai JA, Butler AL, et al. (2018) Combination therapy with anti-HIV-1 antibodies maintains viral suppression. Nature 561: 479–484.

[R13] Kong R, Mark K Louder, Wagh K, Bailer RT, deCamp A, et al. (2014) Improving Neutralization Potency and Breadth by Combining Broadly Reactive HIV-1 Antibodies Targeting Major Neutralization Epitopes. Journal of Virology 89: 2659–2671.

[R14] Dingens AS, Haddox HK, Overbaugh J, Bloom JD (2017) Comprehensive Mapping of HIV-1 Escape from a Broadly Neutralizing Antibody. Cell Host Microbe 21: 777–787.e4.

[R15] Dingens AS, Arenz D, Weight H, Overbaugh J, Bloom JD (2019) An Antigenic Atlas of HIV-1 Escape from Broadly Neutralizing Antibodies Distinguishes Functional and Structural Epitopes. Immunity 50: 520–532.e3.

[R16] Schommers P, Gruell H, Abernathy ME, Tran MK, Dingens AS, et al. (2020) Restriction of HIV-1 Escape by a Highly Broad and Potent Neutralizing Antibody. Cell 180: 471–489.e22.

[R17] Lu CL, Murakowski DK, Bournazos S, Schoofs T, Sarkar D, et al. (2016) Enhanced clearance of HIV-1-infected cells by broadly neutralizing antibodies against HIV-1 in vivo. Science 352: 1001–1004.

[R18] Reeves DB, Huang Y, Duke ER, Mayer BT, Cardozo-Ojeda EF, et al. (2020) Mathematical modeling to reveal breakthrough mechanisms in the HIV Antibody Mediated Prevention (AMP) trials. PLoS Comput Biol 16: e1007626.

[R19] Saha A, Dixit NM (2020) Pre-existing resistance in the latent reservoir can compromise VRC01 therapy during chronic HIV-1 infection. PLoS Comput Biol 16: e1008434.

[R20] Liu B, Zhang W, Zhang H (2019) Development of CAR-T cells for long-term eradication and surveillance of HIV-1 reservoir. Current Opinion in Virology 38: 21–30.

[R21] Margolis DM, Koup RA, Ferrari G (2017) HIV antibodies for treatment of HIV infection. Immunological reviews 275: 313–323.

[R22] Horwitz JA, Halper-Stromberg A, Mouquet H, Gitlin AD, Tretiakova A, et al. (2013) HIV-1 suppression and durable control by combining single broadly neutralizing antibodies and antiretroviral drugs in humanized mice. Proc Natl Acad Sci USA 110: 16538–16543.

[R23] Gruell H, Klein F (2018) Antibody-mediated prevention and treatment of HIV-1 infection. Retrovirology 15: 73.

[R24] Kwon YD, Georgiev IS, Ofek G, Zhang B, Asokan M, et al. (2016) Optimization of the Solubility of HIV-1Neutralizing Antibody 10E8 through Somatic Variation and Structure-Based Design. Journal of Virology 90: 5899–5914.

[R25] Louie RHY, Kaczorowski KJ, Barton JP, Chakraborty AK, Matthew R McKay (2018) Fitness landscape of the human immunodeficiency virus envelope protein that is targeted by antibodies. Proceedings of the National Academy of Sciences 115.

[R26] Ferguson AL, Mann JK, Omarjee S, Ndung’u T, Walker BD, et al. (2013) Translating HIV sequences into quantitative fitness landscapes predicts viral vulnerabilities for rational immunogen design. Immunity 38: 606–617.

[R27] Zanini F, Puller V, Brodin J, Albert J, Neher RA (2017) in vivo mutation rates and the landscape of fitness costs of HIV-1. Virus Evol 3: vex003.

[R28] Neher RA, Leitner T (2010) Recombination rate and selection strength in HIV intra-patient evolution. PLoS Comput Biol 6: e1000660.

[R29] Illingworth CJR, Raghwani J, Serwadda D, Sewankambo NK, Robb ML, et al. (2020) A de novo approach to inferring within-host fitness effects during untreated HIV-1 infection. PLOS Pathogens 16: e1008171.

[R30] Haddox HK, Dingens AS, Hilton SK, Overbaugh J, Bloom JD (2018) Mapping mutational effects along the evolutionary landscape of HIV envelope. *eLife* 7: e34420.

[R31] Tisthammer KH, Dong W, Joy JB, Pennings PS (2021) Comparative Analysis of Within-Host Mutation Patterns and Diversity of Hepatitis C Virus Subtypes 1a, 1b, and 3a. Viruses 13.

[R32] Keck ZY, Pierce BG, Lau P, Lu J, Wang Y, et al. (2019) Broadly neutralizing antibodies from an individual that naturally cleared multiple hepatitis C virus infections uncover molecular determinants for E2 targeting and vaccine design. PLoS Pathog 15: e1007772.

[R33] Zanini F, Puller V, Brodin J, Albert J, Neher RA (2017) in vivo mutation rates and the landscape of fitness costs of HIV-1. Virus evolution 3.

[R34] Meijers M, Vanshylla K, Gruell H, Klein F, L¨assig M (2021) Predicting in vivo escape dynamics of HIV-1 from a broadly neutralizing antibody. Proceedings of the National Academy of Sciences 118: e2104651118.

[R35] Caskey M, Schoofs T, Gruell H, Karagounis T, Kreider EF, et al. (2017) Antibody 10-1074 suppresses viremia in HIV-1-infected individuals. Nature medicine 23: 185–191.

[R36] Caskey M, Klein F, Lorenzi JC, Seaman MS, West AP, et al. (2015) Viraemia suppressed in HIV-1-infected humans by broadly neutralizing antibody 3BNC117. Nature 522: 487–491.

[R37] Schoofs T, Klein F, Braunschweig M, Kreider EF, Feldmann A, et al. (2016) HIV-1 therapy with monoclonal antibody 3BNC117 elicits host immune responses against HIV-1. Science 352: 997–1001.

[R38] Caskey M, Schoofs T, Gruell H, Settler A, Karagounis T, et al. (2017) Antibody 10-1074 suppresses viremia in HIV-1-infected individuals. Nat Med 23: 185–191.

[R39] Bar-On Y, Gruell H, Schoofs T, Pai JA, Nogueira L, et al. (2018) Safety and anti-viral activity of combination HIV-1 broadly neutralizing antibodies in viremic individuals. Nature medicine 24: 1701–1707.